Technical Report

# Joint epigenome profiling reveals cell-type-specific gene regulatory programmes in human cortical organoids

Florian Noack[1,4], Silvia Vangelisti [1,4], Nora Ditzer [2], Faye Chong [1], Mareike Albert [2] & Boyan Bonev [1,3] ✉

Gene expression is regulated by multiple epigenetic mechanisms, which are coordinated in development and disease. However, current multiomics methods are frequently limited to one or two modalities at a time, making it challenging to obtain a comprehensive gene regulatory signature. Here, we describe a method—3D genome, RNA, accessibility and methylation sequencing (3DRAM-seq)—that simultaneously interrogates spatial genome organization, chromatin accessibility and DNA methylation genome-wide and at high resolution. We combine 3DRAM-seq with immunoFACS and RNA sequencing in cortical organoids to map the cell-type-specific regulatory landscape of human neural development across multiple epigenetic layers. Finally, we apply a massively parallel reporter assay to profile cell-type-specific enhancer activity in organoids and to functionally assess the role of key transcription factors for human enhancer activation and function. More broadly, 3DRAM-seq can be used to profile the multimodal epigenetic landscape in rare cell types and different tissues.

Gene expression is regulated by multiple epigenetic mechanisms, which jointly orchestrate cell fate decisions in development and disease. These mechanisms frequently converge on *cis*-regulatory elements (CREs), at which epigenetic marks such as histone marks, DNA methylation and chromatin accessibility can influence the binding of transcription factors (TFs). Physical proximity between CREs and their target genes represents an additional molecular layer that can be established and modulated by cell-type-specific TFs[1–4]. However, the exact relationship between chromatin looping and gene regulation remains unclear.

Methods that profile DNA methylation, chromatin accessibility and three-dimensional (3D) spatial proximity have enabled us to obtain a genome-wide map of the molecular state and connectivity of CREs, revealing high cell-type specificity and rapid dynamics[3,5–8]. Recent multiomics methods are able to map 3D genome architecture together with either DNA methylation[9,10] and accessibility[11] or accessibility together

with transcription[12,13], but still cannot fully capture the complex interplay between multiple epigenetic layers.

Several studies have examined how changes in chromatin accessibility are related to cell fate decisions in either human fetal cortex[8,14,15] or cerebral organoids[16–21], but the importance of other epigenetic modalities such as DNA methylation and chromatin interactions remains unclear. In addition, a promoter-centric assay for 3D genome organization (PLAC-seq) suggested that cell-type-specific regulatory interactions with putative enhancers influence gene expression in the human fetal cortex[7], but did not examine global changes in chromatin organization such as topologically associating domains (TADs) or compartments.

Here we describe the multiomics method 3DRAM-seq, which simultaneously interrogates several epigenetic layers and gene expression. To enable the profiling of specific cell types, we combine 3DRAM-seq with immunoFACS in human cortical organoids and map the epigenome landscape in radial glial cells (RGCs) and intermediate

[1]Helmholtz Pioneer Campus, Helmholtz Zentrum München, Neuherberg, Germany. [2]Center for Regenerative Therapies Dresden, Technische Universität Dresden, Dresden, Germany. [3]Physiological Genomics, Biomedical Center, Ludwig-Maximilians-Universität München, Munich, Germany. [4]These authors contributed equally: Florian Noack, Silvia Vangelisti. ✉e-mail: boyan.bonev@helmholtz-munich.de

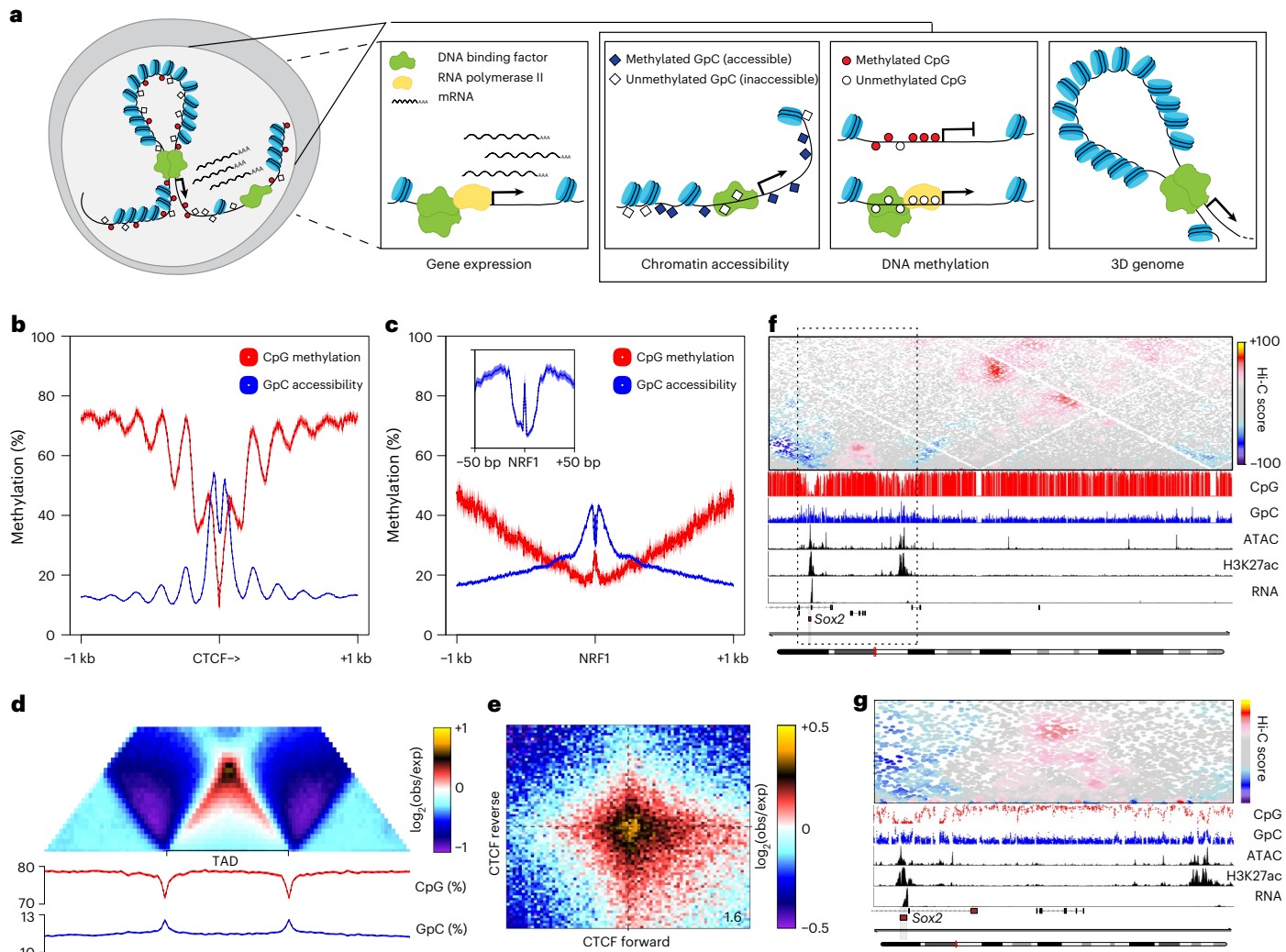

**Fig. 1 | 3DRAM-seq enables joint profiling of 3D genome organization, chromatin accessibility and DNA methylation. a**, Schematic representation of 3DRAM-seq. **b**, Average CpG methylation and GpC accessibility levels at motif-centred and directional CTCF ChIP–seq peaks (5 bp bins). **c**, Same as **b** but for motif-centred NRF1 ChIP–seq peaks (2 bp bins). **d**, Average contact enrichment, DNA methylation and GpC accessibility levels at TADs. exp, expected; obs, observed. **e**, Aggregate contact enrichment between convergent CTCF motifs within ChIP–seq peaks. Number in the bottom-right corner indicates the ratio of the centre enrichment to the mean of the four corners. **f**, Contact map and genomic tracks showing DNA methylation, GpC accessibility, ATAC–seq, H3K27ac ChIP–seq and RNA sequencing across the *Sox2* locus. **g**, Magnified region of the dotted black box indicated in **f**.

progenitor cells (IPCs), identifying TFs associated with widespread epigenetic remodelling. Finally, using a massively parallel reporter assay (MPRA) to profile cell-type-specific enhancer activity in human organoids, we functionally assess the role of key TFs for enhancer function.

## Results

### Development and validation of 3DRAM-seq

We reasoned that chromatin accessibility can be measured alongside DNA methylation and 3D genome organization by incorporating an enzymatic treatment of bulk fixed nuclei with the GpC methyltransferase M.CviPI before Hi-C[22] (Fig. 1a and Extended Fig. 1a).

First, we used bisulfite amplicon sequencing to optimize M.CviPI treatment (Extended Data Fig. 1b and Supplementary Table 1). We then performed 3DRAM-seq in mouse embryonic stem cells in three biological replicates (Supplementary Table 2), which were characterized by high bisulfite conversion efficiency based on spike-in controls (>98%; Extended Data Fig. 1c) and high reproducibility genome-wide (Extended Data Fig. 1d). Furthermore, CTCF and transcriptional start sites (TSSs) had the expected accessibility and DNA methylation pattern[22] (Fig. 1b and Extended Data Fig. 1e,f). Importantly, the single

nucleotide resolution of 3DRAM-seq enabled us to also visualize the motif footprint of TFs such as NRF1, which was consistent with its binding as a homodimer[23] (Fig. 1c and Extended Data Fig. 1g).

To enable profiling of the transcriptome alongside the epigenome, we also optimized the recovery of high-quality RNA from fixed cells. Gene expression was characterized by high reproducibility and uniform coverage (Extended Data Fig. 1h,i). 3D genome organization was highly reproducible across replicates (Extended Data Fig. 1j–l) and was characterized by a distance-dependent decrease in contact probability (Extended Data Fig. 1k), insulation across TAD borders (Fig. 1d) and chromatin loops associated with convergent CTCF sites[5,24] (Fig. 1e). The ability of 3DRAM-seq to profile multiple epigenetic modalities is exemplified at the *Sox2* locus, where we observed increased accessibility and low DNA methylation at its enhancer and promoter, as well as the presence of a chromatin loop connecting these two elements[5] (Fig. 1f,g).

### 3DRAM-seq generates high-quality epigenome data

First, we examined the quality of the transcriptome data and found that 3DRAM-seq was highly correlated with previously published bulk

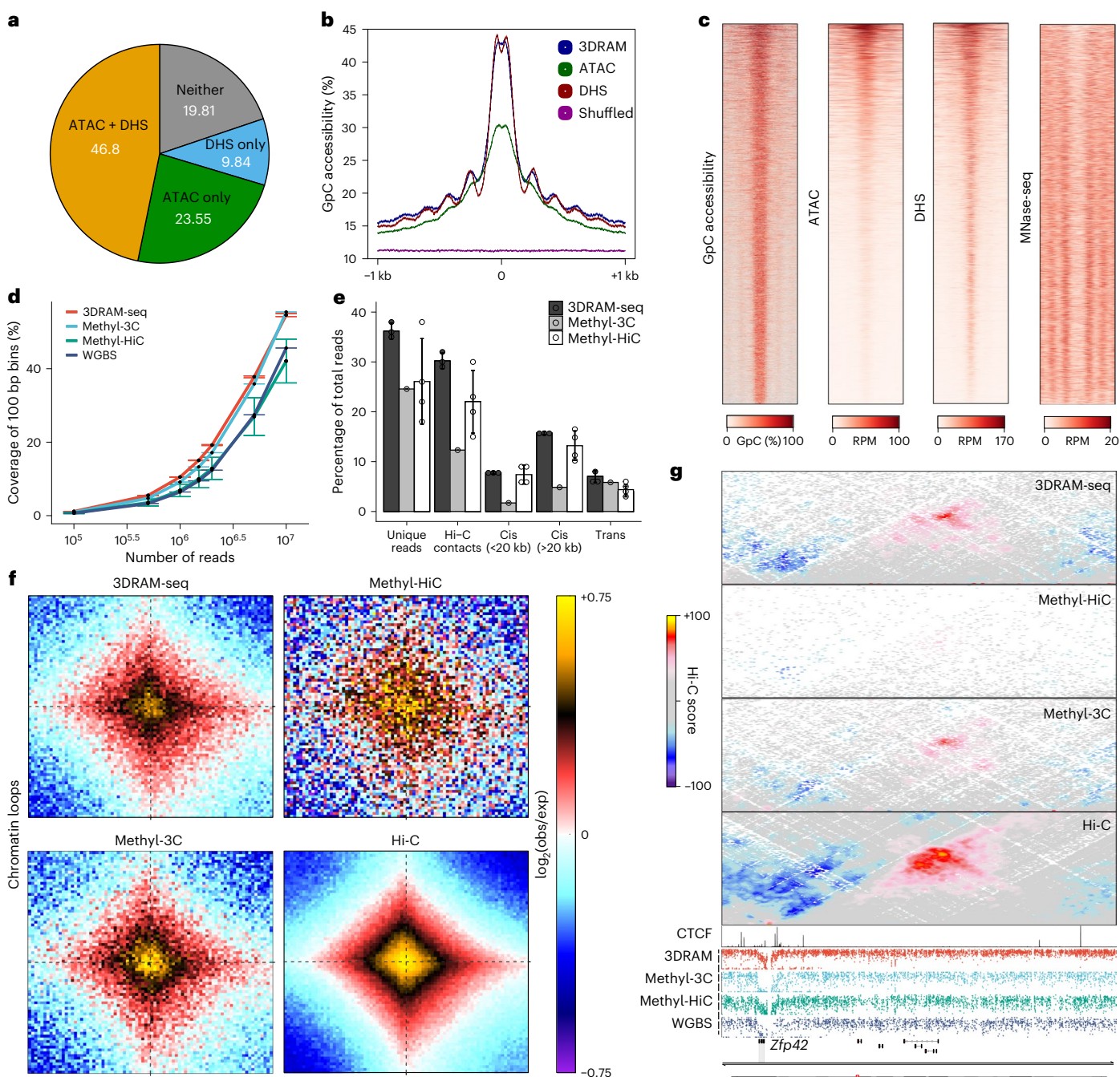

**Fig. 2 | Comparison of 3DRAM-seq with other multiomics methods. a**, Pie chart depicting the percentage of 3DRAM-seq GpC-accessible peaks overlapping with only ATAC–seq peaks, only DHS peaks, both ATAC and DHS, or neither. **b**, Average GpC accessibility levels at 3DRAM-seq GpC-seq, ATAC–seq, DHS-seq or shuffled regions (5 bp bins). **c**, Heatmaps showing accessibility measured by GpC methylation, ATAC–seq (data from the Gene Expression Omnibus (GEO) database, accession number GSE113952), DHS (data from ref. 57), as well as nucleosome occupancy (MNase-seq; data from ref. 58) across GpC peaks. RPM, reads per million mapped reads. **d**, Comparison of coverage across the different methods. Black dot and whiskers indicate the mean ± s.d. (*n* = 1–4 biological replicates). Methyl-3C data from ref. 9; Methyl-HiC data from ref. 10; whole-genome bisulfite

sequencing (WGBS) data from ref. 27. **e**, Sequencing statistics for 3DRAM-seq, Methyl-3C and Methyl-HiC. Bar plot with mean ± s.d.; circles indicate individual data points (*n* = 1–4 biological replicates). **f**, Average contact enrichment from different methods between loops identified in in situ Hi-C data (from ref. 5) at 5 kb resolution using HICCUPs[59]. Please note that the resolution for this and the next panel is highly correlated with the sequencing depth per dataset (contacts: 3DRAM-seq, 311 × 10⁶; Methyl-HiC, 32 × 10⁶, Methyl-3C, 180 × 10⁶; and Hi-C, 2.950 × 10⁹). **g**, Comparison of contact maps and DNA methylation patterns across the different methods, together with a CTCF ChIP–seq track. Each dot in the methylation tracks represents an individual CpG dinucleotide. Source numerical data are available in the source data.

total RNA-seq datasets from the same cell line[5,25], both genome-wide and across specific features (Extended Data Fig. 2a–c).

Next, we focused on chromatin accessibility. We observed high correlation of GpC levels genome-wide compared to assay for transposase-accessible chromatin with sequencing (ATAC–seq) and

DNase I hypersensitivity sites sequencing (DHS) (Extended Data Fig. 2d), and higher levels of GpC methylation at open chromatin regions (Extended Data Fig. 2e). We identified 67,177 accessible regions based on GpC methylation[26] (referred to as GpC peaks), which were largely consistent with the peaks identified by ATAC and DHS (Fig. 2a and

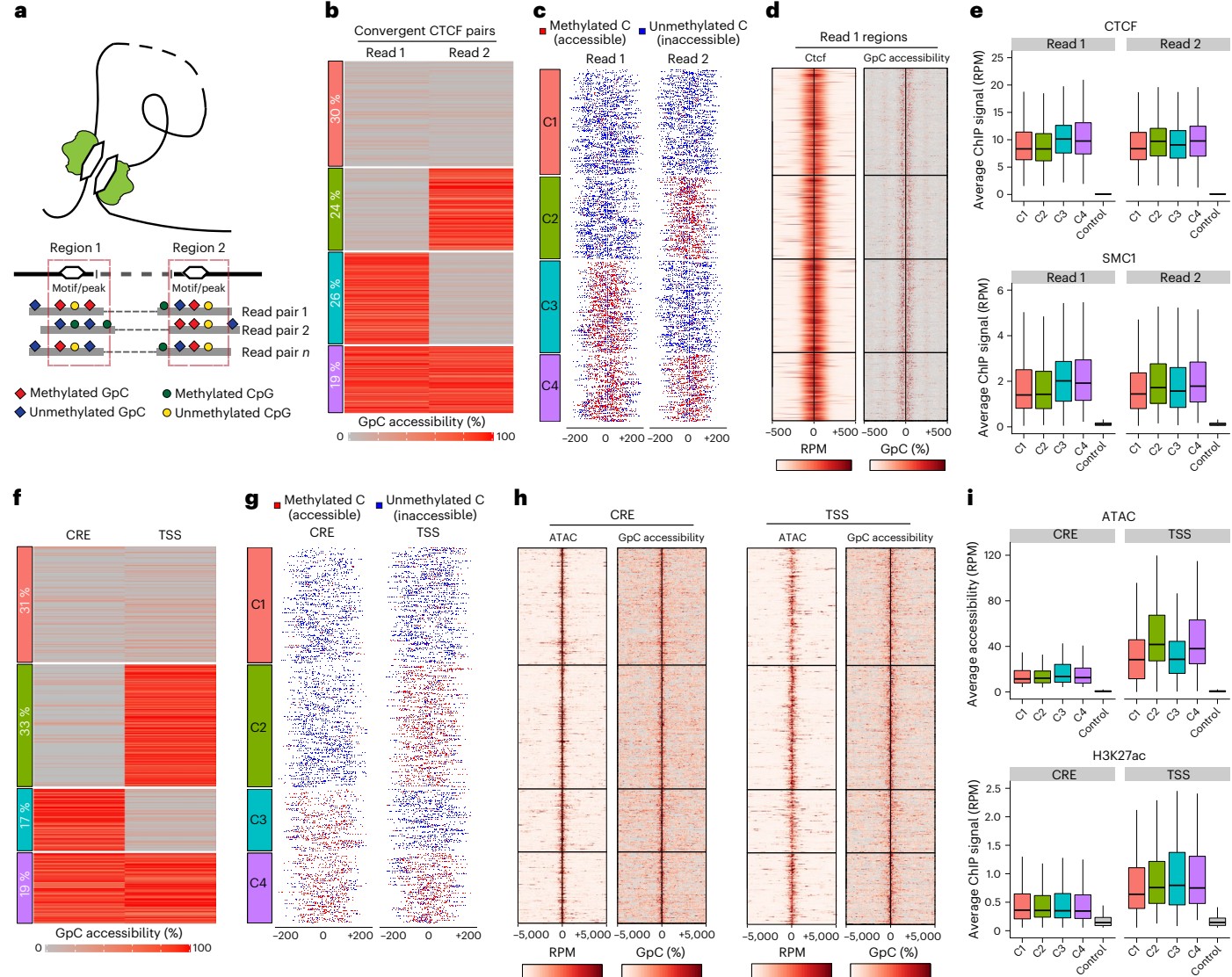

**Fig. 3 | 3DRAM-seq enables paired co-accessibility measurements at single-molecule resolution. a**, Schematic overview of the single-molecule co-accessibility assay. **b**, Clustered average paired co-accessibility levels in a 100 bp windows centred at convergent CTCF motifs separated by at least 1 kb (*k*-means clustering, *k* = 4). **c**, Same as **b** but showing the methylation status of individual GpC dinucleotides in each read. **d**, Average CTCF ChIP–seq signal and GpC accessibility levels in a ±500 bp window for the same read 1 regions containing the CTCF forward motif as in **b**. **e**, Boxplots displaying CTCF or SMC1 (ref. 60) ChIP–seq signals in a ±100 bp window centred at regions identified and clustered in **b** as well as randomized control regions for each (*n* = 483 (C1), 376 (C2), 418

(C3), 302 (C4) and 1,576 (control) regions). **f,g**, Same as **b** and **c** but for read pairs containing a CRE (defined as a distal open chromatic region) on read 1 and a TSS at read 2. Read pairs span at least 5 kb. Odds ratio = 1.07, *P* = 0.6. **h**, ATAC–seq signal and GpC accessibility in a ±5 kb window for the same regions as in **f. i**, Boxplots display ATAC–seq and H3K27ac ChIP–seq signals in a ±250 bp window centred at the CRE or TSS with randomized control regions for each (*n* = 288 (C1), 310 (C2), 155 (C3), 181 (C4) and 928 (control) regions). Clusters as in **f**. All boxplots display median (line), 25th and 75th percentiles (box limits), and 10th and 90th percentiles (whiskers).

Extended Data Fig. 2f) and were characterized by increased accessibility and nucleosome phasing (Fig. 2b,c and Extended Data Fig. 2g,h).

Focusing on DNA methylation and 3D genome organization, we benchmarked 3DRAM-seq against other methods that measure one or both modalities[9,10,27] and observed high genome-wide correlation (Extended Data Fig. 2i). 3DRAM-seq was characterized by high coverage (Fig. 2d) and high resolution at bound CTCF motifs (Extended Data Fig. 2j). 3DRAM-seq also had a high proportion of uniquely mapped reads and total contacts and was characterized by a high *cis*-to-*trans* ratio (Fig. 2e) and a high distance-dependent contact profile, compartments and loops (Fig. 2f,g and Extended Data Fig. 2k,l).

These results suggest that 3DRAM-seq can jointly measure all three epigenetic modalities and gene expression with high reproducibility, coverage and data quality.

## 3DRAM-seq enables single-molecule co-accessibility measurements

We next developed a strategy to quantify the degree of co-accessibility or co-methylation at the single-molecule level. Previous approaches have been limited to a distance of a few hundred base pairs[28] or up to a few thousand base pairs using long-read sequencing[29,30]. By contrast, 3DRAM-seq can be used to interrogate regions that are separated by large distances (Fig. 3a).

We first focused on convergent CTCF motifs overlapping CTCF peaks measured using chromatin immunoprecipitation with sequencing (ChIP–seq) and found that 19% were co-accessible (Fig. 3b,c) compared with 5% in a randomized control, for which only one read was required to overlap a CTCF site (Extended Data Fig. 3a). However, the probability of two CTCF sites to be simultaneously co-accessible

was not higher than expected by chance (Fig. 3b,c; odds ratio = 0.94, $P = 0.52$). Similar results were obtained for convergent CTCF pairs located at the anchors of chromatin loops (Extended Data Fig. 3b) or in non-convergent orientations (Extended Data Fig. 3c). Importantly, all regions were considered open based on average accessibility and were bound by CTCF and SMC1, which indicated rapid turnover of the local epigenetic landscape in individual cells (Fig. 3d,e).

Next, we asked how methylation and accessibility were correlated at individual reads. We observed that most binding sites were characterized by low methylation levels and high accessibility as expected (Extended Data Fig. 3d,e, clusters 1 and 2). However, 12% of the reads were simultaneously methylated and accessible at different positions, which suggested that DNA methylation and CTCF binding are not always mutually exclusive (Extended Data Fig. 3d,e, cluster 4). Finally, 20% of the sites were methylated and inaccessible (Extended Data Fig. 3d,e, cluster 3), which correlated with less bound CTCF and SMC1 (Extended Data Fig. 3f). The proportion of methylated reads was also lower at CTCF sites overlapping TAD boundaries, which potentially indicated a longer residence time of CTCF and SMC1 at these regions (Extended Data Fig. 3g).

Finally, we examined CRE–TSS pairs. Like CTCF, we did not observe any synergistic effect on chromatin accessibility at these regions at either single-molecule (Fig. 3f,g; odds ratio = 1.07, $P = 0.6$) or bulk level (Fig. 3h,i), which suggested that chromatin accessibility is also locally regulated.

Overall, these results showcase the ability of 3DRAM-seq to quantify single-molecule co-accessibility at pairs of regions separated by large genomic distances. Our results suggest that changes in chromatin accessibility and DNA methylation are primarily local events and not typically influenced by the proximity of other genome regions.

## Multimodal epigenetic rewiring in human cortical organoids

To profile the regulatory dynamics in human brain development, we coupled 3DRAM-seq with immunoFACS (Fig. 4a,b, Extended Data Fig. 4a) to purify RGCs and IPCs from human cortical organoids[19,31,32].

After verifying the quality of the data (Extended Data Fig. 4b–g and Supplementary Data 2), we first focused on gene expression. We observed downregulation of RGC-specific genes such as *SOX2*, *HES1* and *PAX6*, and upregulation of genes involved in neuronal differentiation such as *EOMES* and *NEUROG2* in IPCs (Fig. 4c) and Gene Ontology (GO) terms such as neuron differentiation and cell morphogenesis (Fig. 4d). DNA methylation and accessibility at CTCF-accessible motifs or ChIP–seq peaks[33] was characterized by the expected pattern (Fig. 4e,f and Extended Data Fig. 4h,i). Furthermore, GpC peaks identified in RGCs and IPCs significantly overlapped (90.9% and 75.1% respectively) with accessible loci identified in the corresponding cell type in human fetal brain[8].

Focusing on the 3D genome, we found that global chromatin organization was similar between cell types at the level of long-range interactions (Fig. 4g and Extended Data Fig. 4j), TADs (Fig. 4h and Extended Data Fig. 4k) and CTCF loops (Extended Data Fig. 4l). Conversely, specific regulatory interactions, such as at the *SOX2* locus, were more dynamic and correlated with loss of accessibility at its putative enhancers (Fig. 4i).

Next, we examined the dynamics of the epigenetic landscape following RGC-to-IPC transition. We identified 19,316 differentially accessible loci during the RGC-to-IPC transition (Fig. 5a and Extended Data Fig. 5a,b), out of which 12,837 gained and 6,479 lost accessibility (IPC and RGC differential accessible regions (DARs), respectively; Supplementary Table 3). IPC DARs became demethylated after becoming accessible, as expected (Fig. 5b; Pearson's $r = -0.50$), whereas there was no change in DNA methylation at RGC DARs (Fig. 5b; Pearson's $r = -0.05$). Analysis of differential DNA methylation confirmed these conclusions (Extended Data Fig. 5c). Genes interacting with DARs showed enrichment in categories such as neural differentiation, cell morphogenesis and migration (Fig. 5c), were differentially expressed (Extended Data Fig. 5d) and were associated with dynamic interactions (Extended Data Fig. 5e,f).

To identify the mechanism underlying these epigenome changes, we performed a motif enrichment analysis. We found that SOX2, LHX2 and FOS-JUN (also known as AP-1)[3,34] were enriched in RGC DARs (Fig. 5d), whereas EOMES, NFIA and neurogenic bHLH TFs such as NEUROG2 or NEUROD1 were enriched in IPC DARs (Fig. 5f and Extended Data Fig. 5g). Furthermore, NEUROG2 motifs were associated with loss of DNA methylation in IPCs (Extended Data Fig. 5h–j), analogous to our previous results in the mouse cortex[3].

Next, we asked whether TF binding is correlated with dynamic chromatin looping. We identified LHX2, a TF important for forebrain specification and RGC proliferation[35], to be associated with RGC-specific regulatory loops (Fig. 5e), although other factors are probably also involved (Extended Data Fig. 5l). Interestingly, these differences were not accompanied by altered DNA methylation levels at LHX2 motifs, which suggested that changes in these modalities can be at least partially uncoupled (Extended Data Fig. 5h,k). These results were in contrast to NEUROG2-motif-containing IPC DARs, which showed both increased connectivity and decreased DNA methylation levels in IPCs (Fig. 5g and Extended Data Fig. 5m,n).

These TF-associated epigenome dynamics can be exemplified at two loci. At the *GAS1* locus, which is highly expressed in RGCs and has several LHX2-containing distal DARs, loss of accessibility in IPCs was accompanied by weaker interactions with the *GAS1* promoter (Fig. 5h). At the *NFIA* locus, which contains multiple NEUROG2 motifs and is upregulated in IPCs, we observed the opposite pattern: gain of chromatin accessibility and stronger interactions (Fig. 5i). A similar pattern was observed in the mouse *Nfia* locus during cortical development[3], which indicates that there is evolutionary conservation (Extended Data Fig. 5o).

Finally, we asked whether we could identify synergistic effects between TFs using our single-molecule co-accessibility approach. Indeed, we observed that two of the RGC-enriched TFs, SOX2 and LHX2, are characterized by high co-accessibility in RGCs but not in IPCs (Fig. 5j; odds ratio in RGCs of 3.23, $P = 1.51 \times 10^{-15}$; odds ratio in IPCs of 9.3, $P = 2 \times 10^{-12}$). Conversely, NEUROG2 and EOMES motifs were co-accessible primarily in IPCs but not in RGCs (Fig. 5k; odds ratio in RGCs of 11.08, $P < 2.2 \times 10^{-16}$; odds ratio in IPCs of 4.1, $P < 2.2 \times 10^{-16}$). These findings are consistent with previous results showing that SOX2–LHX2 (ref. 36) and NEUROG2–EOMES[37] can either interact directly or co-bind on chromatin in RGCs and IPCs, respectively.

Overall, these results show that 3DRAM-seq can be used to dissect the multilayered epigenome landscape in a heterogeneous system such as human cortical organoids. They also identify distal regulatory regions and TFs that are associated with dynamic remodelling of the epigenetic landscape during RGC-to-IPC transition.

## Epigenome dynamics at transposable elements in organoids

To examine the contribution of transposable elements (TEs) to the epigenetic rewiring in human brain development, we first focused on chromatin looping. We found that loci containing endogenous retroviruses (ERVs) such as LTR24C and HERVE-int interacted strongly in RGCs (Extended Data Fig. 6a,b) and were enriched in TF-binding motifs such as TEAD and FOS-JUN (Extended Data Fig. 6a,b).

Next, we identified two classes of TEs, MER130 and UCON31, which became more accessible in IPCs (Fig. 6a,c,d and Supplementary Table 4). However, both TE classes remained highly methylated overall (Fig. 6b–d), which suggested that there was a partial uncoupling of these two modalities. This pattern was also conserved in mouse corticogenesis[3] (Extended Data Fig. 6c,d). To identify factors that regulate both TE classes, we performed motif analysis and found a strong enrichment of neurogenic TFs such as NEUROG2, NFIA and NFIX, which have been implicated in IPC differentiation[38] (Fig. 6e,f). At least some but not all of these TEs were also bound by NEUROD2 or NEUROG2 in the mouse cortex (Extended Data Fig. 6e,f) based on ChIP–seq data.

Finally, we asked whether the TF binding and accessibility changes at these two TE classes could also affect transcription. We found a

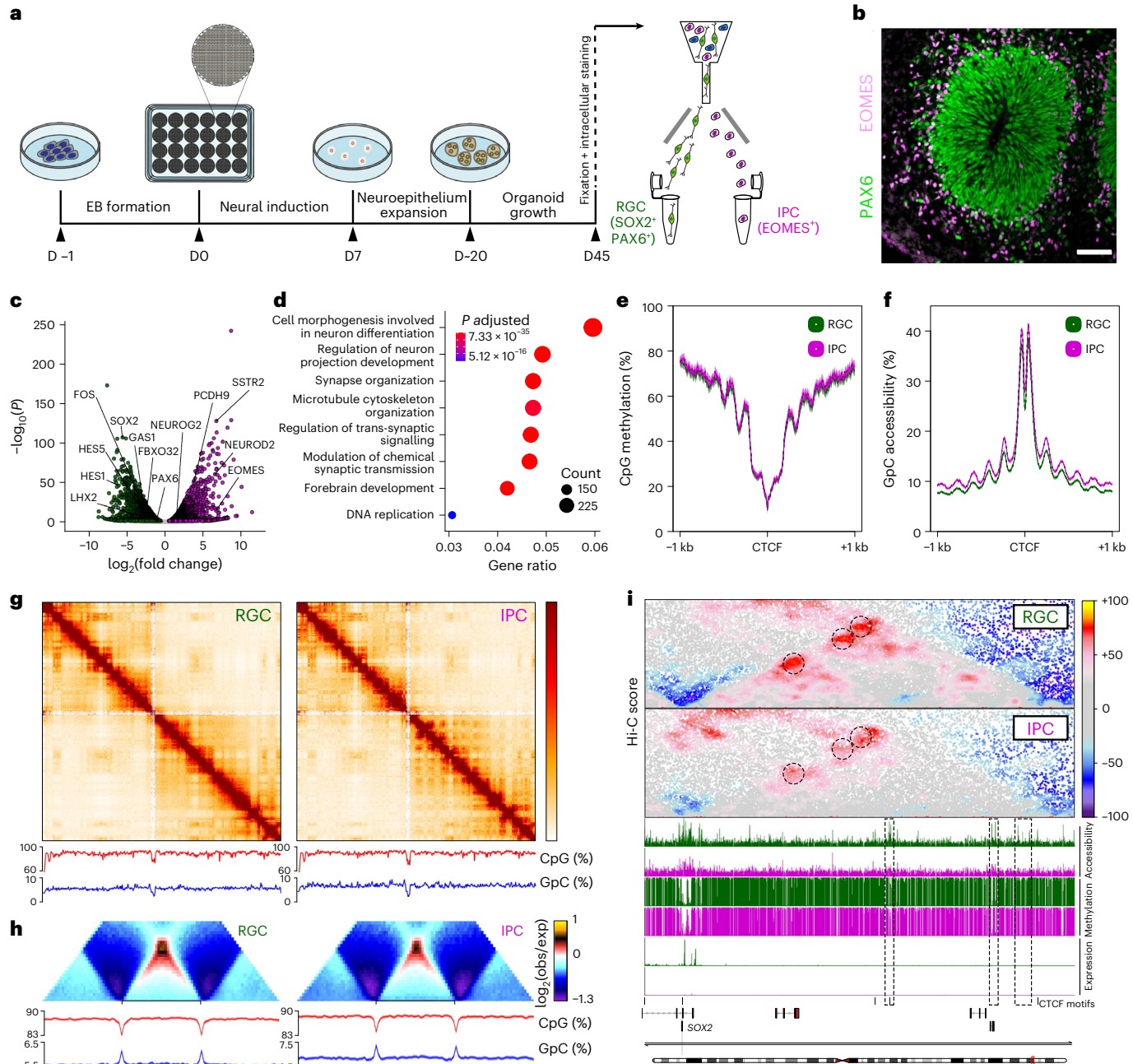

**Fig. 4 | Combining 3DRAM-seq with immunoFACS enables multimodal profiling of the cell-type-specific epigenetic landscape in human cortical organoids. a**, Experimental overview of human cortical organoid generation followed by the isolation of RGCs and IPCs by immunoFACS. D, day; EB, embryoid body. **b**, Representative immunofluorescence image of a neural rosette formed within a cortical organoid at day 45. Scale bar, 10 μm. **c**, Scatterplot depicting significantly (false discovery rate (FDR) < 0.05) upregulated or downregulated genes in RGC-to-IPC differentiation. Purple, genes upregulated in IPC compared to RGC. Green, genes downregulated in IPC. **d**, GO term enrichment (biological processes) of differentially regulated genes. Colour and size of circles indicate Benjamini–Hochberg adjusted *P* value (hypergeometric test) and number of genes, respectively. **e,f**, CpG methylation (**e**) and GpC accessibility (**f**) levels at CTCF-motif-centred GpC peaks for RGCs and IPCs. **g**, Contact maps (top) and CpG methylation and GpC accessibility maps (bottom) for chromosome 3 (200 kb bins). **h**, Average contact enrichment, CpG methylation and GpC accessibility levels at TADs for RGCs and IPCs. **i**, Contact maps, GpC accessibility, DNA methylation and expression levels for RGCs and IPCs at the *SOX2* locus. Black dotted circles and boxes in the genomic tracks indicate putative CREs.

significant enrichment for GO terms related to neuronal differentiation and maturation or for brain development (Fig. 6g and Extended Data Fig. 6g), but no evidence for differential gene expression (Fig. 6h).

**Cell-type-specific MPRA in human cortical organoids**

To dissect whether the identified DARs can drive gene expression, we applied a MPRA to cortical organoids, coupling electroporation with immunoFACS to dissect cell-type-specific regulation (Fig. 7a).

We included 5,876 sequences (500 scrambled controls) in the MPRA pool and recovered >98% (Fig. 7b and Extended Data Fig. 7a,b). We used electroporation coupled with immunoFACS to obtain high-quality, cell-type-specific MPRA libraries from RGCs, IPCs and PAX6⁻EOMES⁻ (N) cells (Extended Data Fig. 7c–f and Supplementary Table 5). Notably, regions overlapping with previously validated VISTA enhancers[39] were more active compared to scrambled controls (Extended Data Fig. 7g), and reporter activity of significantly active

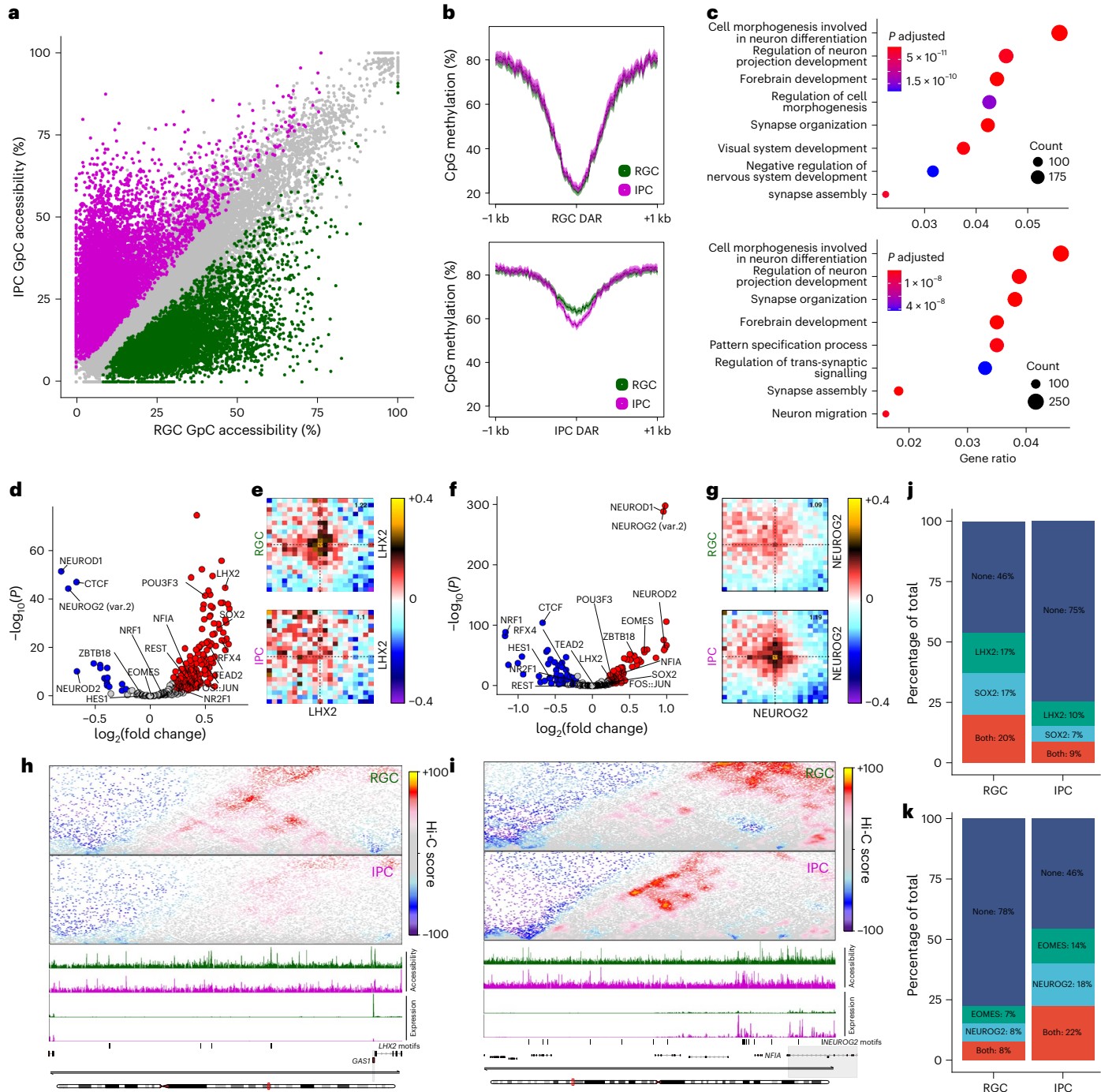

**Fig. 5 | TFs associated with epigenome remodelling in cortical organoids.**
**a**, Scatterplot depicting GpC accessibility levels for individual GpC peaks in RGCs and IPCs. Peaks that lose (RGC DAR; $n = 6479$) or gain accessibility (IPC DAR; $n = 12,837$) are coloured, respectively. Grey dots indicate GpC peaks that do not change ($n = 46,964$). **b**, Average CpG methylation levels of RGCs or IPCs at RGC (top) or IPC (bottom) DARs (10 bp bins). **c**, GO term enrichment analysis of genes associated with RGC (top) or IPC (bottom) DARs based on chromatin contacts (Methods). **d**, Volcano plot showing the enrichment of TF motifs within RGC DARs. Red and blue dots indicate significantly ($P \leq 0.01$; log(absolute fold change) $\geq 0.25$) enriched or depleted motifs, respectively. **e**, Aggregated contact enrichment for pairs of RGC DARs containing LHX2 motifs ($n = 3459$). Number in the top-right corner indicates the ratio of the centre enrichment to the mean of the four corners. **f,g**, Same as **d** and **e** but for IPC DARs and IPC DARs with the NEUROG2 motif ($n = 9,116$), respectively. **h,i**, Contact maps, GpC accessibility levels and gene expression for RGCs and IPC at the *GAS1* (**h**) and *NFIA* (**i**) locus. **j**, Single-molecule co-accessibility levels of paired reads (separated by 100–300 bp) containing LHX2 and/or SOX2 motifs overlapping RGC peaks. **k**, Single-molecule co-accessibility levels of paired reads (separated by 100–300 bp) containing EOMES and/or NEUROG2 motifs overlapping IPC peaks.

CREs was correlated with their classification based on cell-type-specific accessibility (Fig. 7c).

To identify which TFs govern this cell-type specificity, we used two parallel approaches. First, a regression-based approach[40] identified neurogenic bHLH TFs such as NEUROG2, T-box TFs such as EOMES, and the RGC repressor HES1 (ref. 41) as correlated with increased reporter activity in IPCs (Extended Data Fig. 7h). RGC activity was associated with multiple TF families with a known role in cortical development

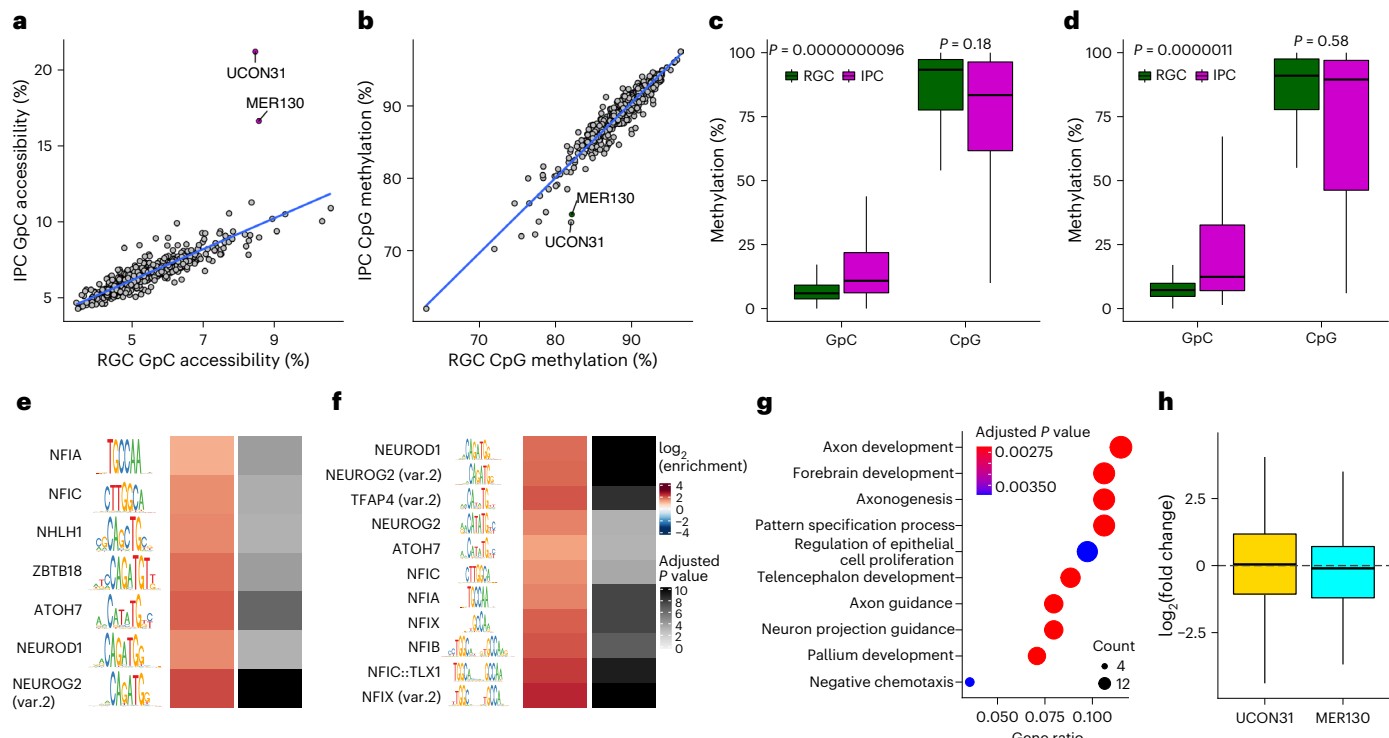

**Fig. 6 | MER130 and UCON31 repetitive elements are associated with changes in accessibility and enriched for neurogenic TF motifs. a,b,** Scatterplot depicting the median accessibility (**a**) or DNA methylation (**b**) levels of different classes of repetitive genomic elements. **c,d,** Boxplots representing GpC accessibility and DNA methylation levels for MER130 (**c**; $n = 181$) and UCON31 repeats (**d**; $n = 117$). Statistical significance was calculated using paired two-sided Wilcoxon rank-sum test. **e,f,** TF motif enrichment for MER130 (**e**) and UCON31

(**f**) repetitive elements. **g,** GO term enrichment analysis of genes associated with UCON31 repetitive elements (Methods). **h,** Boxplots depicting gene expression changes (IPC versus RGC) in genes interacting with either UCON31 ($n = 121$) or MER130 ($n = 174$). Statistical significance was calculated using a two-sided Wilcoxon rank-sum test. All boxplots display median (line), 25th and 75th percentiles (box limits), and 10th and 90th percentiles (whiskers).

such as SOX[42] and NR2F2 (also known as COUP-TF1)[43]. Second, we clustered the significantly active enhancers (Fig. 7d,e) and found that SOX and neuronal bHLH motifs were strongly enriched in RGC or IPC clusters, respectively. Notably, the POU3F3 motif was strongly enriched in cluster 5 that had MPRA activity in both IPCs and N cells, consistent with its function in neuronal migration[44].

To test whether TF binding can directly affect activity, we mutated selected TF motifs. As predicted, this resulted in a strong reduction of reporter activity (Fig. 7f,g and Extended Data Fig. 7i,j). Furthermore, although only 30% of MER130 and 25% of UCON31 sequences had significant enhancer activity in either RGCs or IPCs, mutating NEUROG2 motifs within those led to a significant reduction in their activity, most prominently in IPCs (Extended Data Fig. 7k,l).

Next, we asked whether TFs can act synergistically to increase the transcriptional output, as previously proposed[45,46]. To do so, we focused on TF pairs such as SOX2–LHX2 (ref. 36), NEUROG2–EOMES[37] and NEUROD1–POU3F3 (ref. 47), which have been shown to either interact or co-bind on chromatin in RGCs, IPCs and neurons, respectively. Importantly, enhancers with both motifs had higher activity in a cell-type-specific way (Fig. 7h,i and Extended Data Fig. 7m–p), which indicated synergistic effects.

These complex regulatory effects can be exemplified at the *PCHD9* locus, which was strongly upregulated in IPCs (Fig. 4c). This increase in expression was accompanied by the formation of a chromatin loop between its promoter and an intragenic enhancer that becomes accessible in IPCs (Fig. 7j). Although mutation of the EOMES motif within this enhancer strongly reduced MPRA activity, mutation of the NEUROG2 motif completely abolished it (Fig. 7k), which indicated a potential hierarchy within TF function.

Finally, we focused on the *FBXO32* locus, which is not expressed in the mouse cortex at embryonic day 14 (ref. 3) but is present in human ventricular RGCs[14] and in organoids (Figs. 4c and 7l). We observed multiple putative enhancer elements (E1–E4), which were accessible and engaged in chromatin looping with the *FBXO32* promoter in RGCs but not in IPCs (Fig. 7l). Two enhancer elements (E2 and E4) were considered differentially accessible and were therefore included in our MPRA. Both were significantly active in RGCs but not in IPCs (Extended Data Fig. 7q), which was in agreement with the predictions based on accessibility. However, when we examined the mouse *Fbxo32* locus, we found no evidence for RGC-specific enhancers or chromatin loops despite an overall high degree of synteny (Extended Data Fig. 7r).

Finally, we focused on the E2 FBXO32 enhancer based on its high MPRA activity (Extended Data Fig. 7q). Comparative genomic analysis showed that the human and the orthologous mouse sequence were moderately conserved (64.7%) but had different predicted TF-binding motifs (Extended Data Fig. 7s). Importantly, only the human but not the orthologous mouse sequence was able to drive expression (Fig. 7m,n and Extended Data Fig. 7t).

Overall, the application of MPRA to cortical organoids enabled us to directly quantify cell-type-specific enhancer activity, dissect the importance of key TFs to their regulation in human neurogenesis and validate a human enhancer for RGCs.

## Discussion

To obtain a comprehensive map of the genome-wide regulatory landscape, we developed a method, 3DRAM-seq, which can simultaneously profile 3D genome organization, chromatin accessibility and DNA methylation together with gene expression.

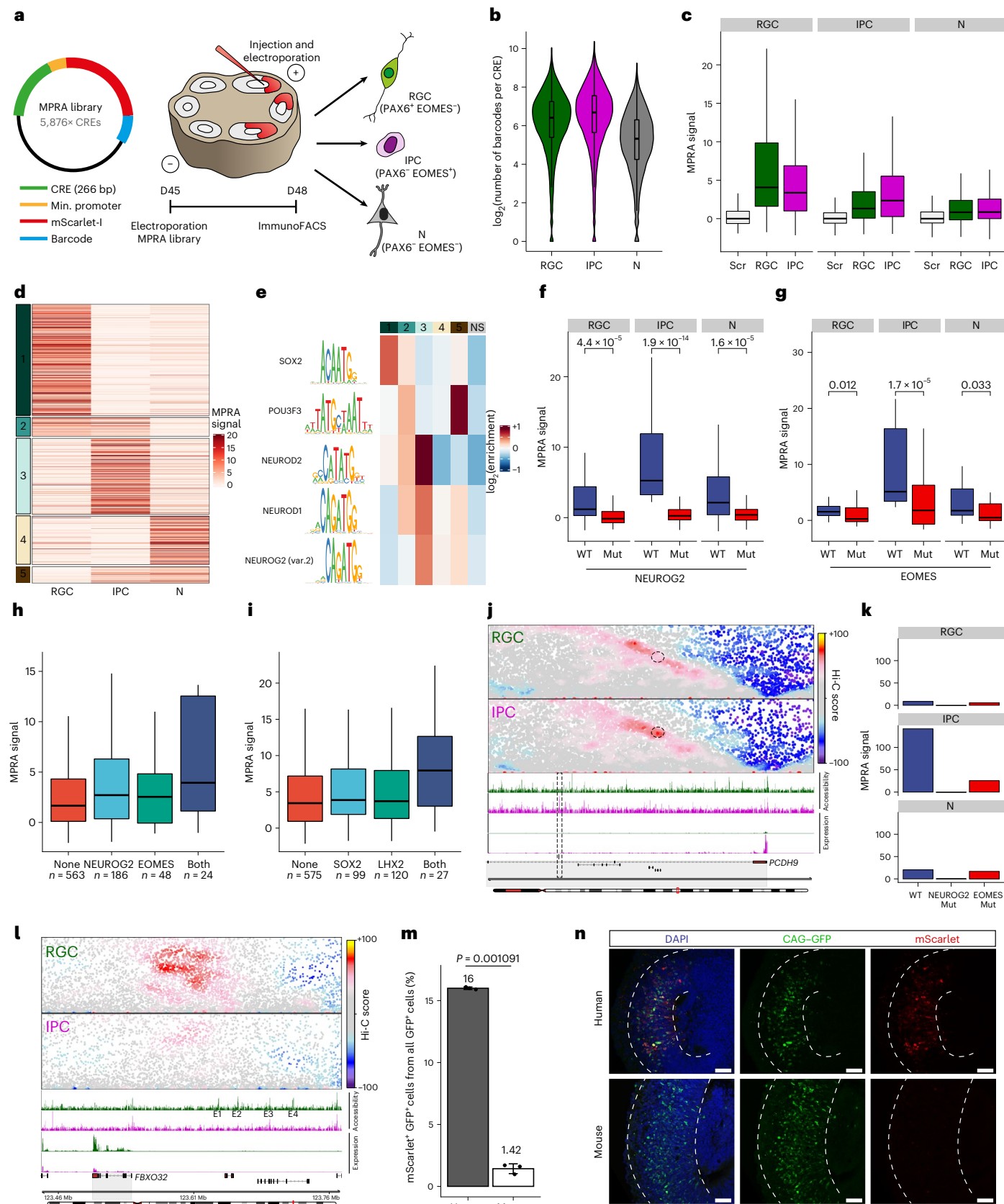

Exploiting the inherent single-molecule resolution of the assay, we were able to measure co-accessibility at pairs of regions that are physically proximal but separated linearly by large distances. Applying this to pairs of convergent CTCF sites and enhancers and promoters, we found no evidence for synergistic effects. These results are consistent with the low CTCF residence time, as previously observed[48]. Furthermore the results suggest that the increased accessibility associated with CTCF binding is highly dynamic and occurs independently at each anchor, which is in agreement with recent findings based on microscopy that CTCF loops are relatively rare and occur between 3–6% (ref. 49) and

**Fig. 7 | Cell-type-specific MPRA in human cortical organoids. a**, Experimental overview of the immunoMPRA in human cortical organoids. **b**, Violin and boxplots displaying the number of unique barcodes obtained per CRE ($n$ = 5,822, 5,831 and 5,801 for RGCs, IPCs and N cells, respectively). **c**, Boxplots depicting MPRA signals of scrambled controls (Scr; $n$ = 492) and significantly active CREs associated with RGC ($n$ = 273) or IPC ($n$ = 548) enhancers. **d**, $K$-means clustering ($k$ = 5) of significantly active CREs in all cell types. Each row represents a single CRE and its MPRA signal in RGCs, IPCs and N cells. **e**, Heatmap depicting motif enrichment for the five significant CRE clusters from **d**, and CREs not significant in any condition (NS). **f,g**, Boxplots depicting cell-type-specific MPRA signals for significantly active CREs in IPCs containing either wild-type (WT) or mutated (Mut) NEUROG2 ($n$ = 83) or EOMES ($n$ = 23) TF motif. **h,i**, Boxplots showing MPRA signal in IPCs (**h**)

and RGCs (**i**) for significantly active CREs containing the indicated motifs. **j**, Contact maps, GpC accessibility and gene expression for RGCs and IPCs at the *PCDH9* locus. Dotted circle indicates an IPC DAR containing both NEUROG2 and EOMES motifs. **k**, Bar plots showing MPRA activity of the CREs indicated in **j** with or without motif mutations. **l**, Same as **j** but for the *FBXO32* locus. **m,n**, Data relate to the E2 FBXO32 enhancer. **m**, Bar chart displaying the percentage of mScarlet⁺ cells among the GFP⁺ cell populations ($n$ = 3) in quantified by FACS. Statistical significance was calculated using a two-sided unpaired $t$-test. **n**, Representative immunofluorescence images of co-electroporated human cortical organoids (from $n$ = 3 independent experiments). Scale bars, 50 μm. All boxplots display the median (line), 25th and 75th percentiles (box limits), and the 10th and 90th percentiles (whiskers). Source numerical data are available in the source data.

20–30% (ref. [50]) of the time in single cells. They are also consistent with the lack of stable enhancer–promoter loops[51] and point to dynamic but independent accessibility changes at these regions.

To further improve 3DRAM-seq and to make it applicable to rare cell types, we coupled it with immunoFACS-based purification to dissect the regulatory landscape in human cortical organoids. We identified multiple TFs associated with epigenome rewiring, including NEUROG2, which we have previously shown to have a similar role in mouse cortical development[3], as well as the TF LHX2 in RGCs. Although LHX2 has previously been shown to be required for olfactory receptor choice by mediating *trans* interactions[52], its importance for enhancer–promoter rewiring in the cortex has not yet been demonstrated.

In addition to TFs, we identified TEs associated with dynamic 3D chromatin looping (such as LTR24C and HERVE-int) and chromatin accessibility (UCON31 and MER130). These TEs harboured distinct TF motifs, and in the case of UCON31 and MER130, have been proposed to be co-opted as enhancers in the mouse cortex[53]. However, increased chromatin accessibility at these two TE classes did not lead to an overall change in gene expression of their target genes, which suggests that other mechanisms (such as DNA methylation) were able to counteract the changes in accessibility.

Finally, to complement our 3DRAM-seq data and to directly measure cell-type-specific enhancer activity, we applied an electroporation-based MPRA to human cortical organoids. We showed that some TF pairs, such as LHX2–SOX2 and NEUROG2–EOMES, act synergistically at enhancers, as has been proposed based on ChIP–seq data[36,37].

While this manuscript was in revision, another group reported combining Hi-C with nucleosome occupancy and methylome sequencing (NOMe–HiC)[54]. Although conceptually similar, this approach requires significantly more cells and results in a lower number of informative (>20 kb) contacts compared to 3DRAM-seq. Interestingly, the authors reported that reads located at chromatin loops anchors are more likely to be co-accessible than by chance. However, a potential reason for the differences with our results is that a chromatin loop anchor size of 25 kb was used in that study[54], whereas we used a window of 100 bp centred around a genomic feature (such as CTCF motif or CRE–TSS) to determine co-accessibility patterns.

The advantages of 3DRAM-seq include low cost, reduced input requirements and ability to perform joint co-accessibility analysis. Furthermore, it can be expanded in several ways. First, adapting the method to be compatible with Micro-C[55] or Hi-C3.0 (ref. [56]) will further enhance the identification of regulatory chromatin loops. Second, combining 3DRAM-seq with region-specific capture methods will facilitate more cost-effective analysis of the multimodal epigenome reorganization at specific loci of interest and enable paired single-molecule TF footprinting[28].

## Online content

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

## Methods

### Cell culture and organoids generation

**Mouse embryonic stem cell culture.** The mouse embryonic stem (mES) cell E14TG2a line was obtained from the American Type Culture Collection (ATCC, CRL-1821). Cells were cultured at 37 °C (5% $CO_2$) on 0.1% gelatin (Millipore, ES-006-B) coated 10-cm dishes (Falcon, 35300) in sterile-filtered (Millipore, SCGPS05RE) Glutamax DMEM medium (Gibco, 31966047) supplemented with 15% heat-inactivated FBS (ThermoFisher, 16141079), 1% penicillin–streptomycin (Gibco, 15140122), 1% MEM (Gibco, 11140035), 0.2% β-mercaptoethanol (Gibco, 31350010) and 0.1% LIF (Merck, ESG1106). Medium was changed every day and cells were split using Accutase (ThermoFisher, A1110501) every second day to a density of $3 \times 10^5$ cells.

**Human induced pluripotent stem cell maintenance.** The human induced pluripotent stem (hiPS) cells used for the generation of cortical organoids and 3DRAM-seq were provided by the Helmholtz Zentrum München iPSC core facility (ISFi001-A hiPS cell line)[61]. For the MPRA in cortical organoids, we used the previously generated hiPS cell line CRTDi004-A (https://hpscreg.eu/cell-line/CRTDi004-A), which was derived from a healthy donor[62]. The cells were cultured at 37 °C (5% $CO_2$) on plates (StemCell Technologies, 38016) coated with Matrigel (Corning, 354277) in mTeSR plus medium (StemCell Technologies, 100-0276), and passaged as colonies using ReLeSR (StemCell Technologies, 05872) or Gentle Cell Dissociation reagent (StemCell Technologies, 07174). Before each round of human cortical organoid production, hiPS cell cultures were tested for mycoplasma contamination using a LookOut Mycoplasma PCR Detection kit (Sigma-Aldrich, MP0035) and validated for pluripotency markers by immunohistochemical staining using a Human Pluripotent Stem Cell 3-Color Immunocytochemistry kit (R&D Systems, SC021). Chromosome analysis of the HMGU1 fixed cell suspension was performed by the Cytogenetics Laboratory of the Cell Guidance Systems, and 20 metaphases were analysed.

**3D human cortical organoid generation.** For the generation of 3D human cortical organoids (3D-hCOs), hiPS cells were cultured at 37 °C (5% $CO_2$) on 10-cm dishes (Corning, 353803) coated with Matrigel (Corning, 354277) in mTeSR plus medium (StemCell Technologies, 100-0276) to 80–90% confluency. The day before the culture reached the correct confluency, hiPS cells were pre-treated with 1% dimethyl sulfoxide (Sigma-Aldrich, D2650), and the mTeSR plus medium was switched to complete Essential 8 medium (Life Technologies, A1517001). After 24 h, the hiPS cell colonies were dissociated to single cells using Gentle Cell Dissociation reagent (StemCell Technologies, 07174), and resuspended in complete Essential 8 medium supplemented with 10 μM of the ROCK inhibitor Y-27632 (Sigma-Aldrich, SCM075) to prevent cell death. In total, $1 \times 10^4$ single cells were seeded in one well of an AggreWell 800 plate (StemCell Technologies, 34815) pre-treated with 500 μl of Anti-Adherence Rinsing Solution (StemCell Technologies, 07010). The plate was then centrifuged at 100$g$ for 3 min at room temperature to distribute the cells into the microwells. hiPS-cell-derived embryo bodies were formed within the microwells after 24 h and were collected by firmly pipetting the medium up and down with a cut 1 ml pipetting tip and collected on a 40 μm cell strainer (VWR, 734-2760). The embryo bodies were transferred to ultra-low attachment 10-cm dishes (Corning, 3262) and cultured in Essential 6 medium (Life Technologies, A1516401) supplemented with 2.5 μM dorsomorphin (StemCell Technologies, 72102), 10 μM SB-431542 (StemCell Technologies, 72232) and 2.5 μM XAV-939 (Tocris, 3748) for the first 5 days of culture. Medium was changed daily, except for day 1. At day 7, embryo bodies were embedded in a drop of Matrigel (Corning, 354234) using Organoid Embedding Sheet (StemCell Technologies, 08579). Matrigel-embedded embryo bodies were cultured in differentiation medium (without vitamin A) containing a 1:1 mixture of DMEM/F-12 (Gibco, 11330-032) and neurobasal medium (Gibco, 21103-049) supplemented with

0.5% N2 supplement (Life Technologies, 17502-048), 0.025% Insulin (Sigma-Aldrich, I9278), 1% B-27 Supplement minus vitamin A (Life Technologies, 12587010), 1% GlutaMAX supplement (Life Technologies, 35050-061), 0.5% MEM-NEAA (Life Technologies, 1140-050), 1% penicillin–streptomycin (Gibco, 15140122) and 0.1% β-mercaptoethanol (Gibco, 31350010) for 4 days with one change of medium 2 days after embedding. After day 4, Matrigel-embedded embryo bodies were cultured in differentiation medium (with vitamin A) containing a 1:1 mixture of DMEM/F-12 and neurobasal medium supplemented with 0.5% N2 supplement, 0.025% insulin, 1% B-27 supplement (Life Technologies, 17504044), 1% GlutaMAX supplement, 0.5% MEM-NEAA, 1% penicillin–streptomycin and 0.1% β-mercaptoethanol with changes of medium every 3–4 days. 3D-hCOs were collected after 45 days and dissociated using a Papain-based Neural Tissue Dissociation kit (Miltenyi Biotec, 130-092-628) following the manufacturer's dissociation protocol.

**Sectioning and immunohistology.** Cortical organoids were washed twice in PBS for 5 min, fixed in freshly prepared 4% formaldehyde in PBS at room temperature for 20 min, washed twice in PBS for 5 min and cryoprotected in 30% sucrose in PBS at 4 °C until they sank. Subsequently, cortical organoids were embedded in Tissue Tek OCT compound (Science Services, SA62550-01), snap frozen on dry ice and finally cryosectioned (-16 μm) using a CryoStar NX70 (ThermoFisher). Sections were collected on Superfrost Plus adhesive microscope slides (ThermoFisher, J1800AMNZ) and stored at −80 °C until further use. For immunohistochemistry, the sections were hydrated in PBS and incubated for 1 h at room temperature in PBS blocking buffer containing 5% horse serum (Sigma-Aldrich, H0146), 1% BSA (Thermo Scientific, 15260-037) and 0.3% Triton X-100 (Sigma Aldrich, X100). Staining was performed overnight at 4 °C either with anti-PAX6 (1:100 dilution; BioLegend, 901301) and anti-EOMES (1:150 dilution; R&D Systems, AF6166) antibodies for wild-type organoids or with anti-RFP (1:1,000 dilution; Rockland, 200-101-379) and anti-GFP (1:1,000; Abcam, ab13970) for co-electroporated organoids. All antibodies were diluted in PBS blocking buffer. Sections were washed three times for 10 min with 0.1% Triton X-100 in PBS followed by secondary staining with either donkey anti-sheep-A488 (Thermo Scientific, A32794) and donkey anti-rabbit-A555 (Thermo Scientific, A32794) or goat anti-chicken Alexa Fluor 488 (Thermo Scientific, A-11039) and donkey anti-goat Alexa Fluor Plus 555 (Thermo Scientific, A32816) all diluted in blocking buffer (1:1,000). Sections were washed, stained with 4′,6-diamidino-2-phenylindole (DAPI) and finally mounted using Fluoromount-G (Invitrogen, 00-4958-02). All images were acquired using a Zeiss LSM 710 confocal microscope.

**Cell fixation.** Cells from dissociated 3D-hCOs, electroporated organoids or mES cells were fixed at a concentration of $1 \times 10^6$ cells per ml with 1% formaldehyde (ThermoFisher, 28906) in PBS for 10 min at room temperature with slow rotation. To quench the reaction, glycine (ThermoFisher, 15527-013) was added to a final concentration of 0.2 M followed by incubation for 5 min at room temperature with slow rotation. Thereafter, cells were spun down at 500$g$ for 5 min at 4 °C and washed once with PBS containing 1% BSA (Sigma-Aldrich, B6917) and 0.1% RNAsin plus RNase inhibitor (Promega, N261A). Fixed cells from mES cells were directly used for 3DRAM-seq, whereas fixed cells from 3D-hCOs were first subjected to immunoFACS before use.

**ImmunoFACS.** ImmunoFACS was performed as previously described (https://www.protocols.io/view/immunofacs-b2a2qage/)[3] with minor modifications, which included the addition of 0.5× complete, EDTA-free protease inhibitor cocktail (Roche, 11873580001) to all buffers. SOX2-PE (1:20; BD Biosciences, 562195), PAX6-Alexa Fluor 488 (1:40; BD Biosciences, 561664) and EOMES-eFluor660 (1:20; Thermo Scientific, 50-4877-41) antibodies were used. Cell sorting was carried out on a FACSAria Fusion (BD Biosciences; laser: 405 nm, 488 nm, 561 nm

and 640 nm) or a FACSAria III (BD Biosciences; laser: 405 nm, 488 nm, 561 nm and 633 nm) using a 100 µm nozzle. After sorting, cells were either directly used for 3DRAM-seq, MPRA library preparation or RNA was extracted using a Quick-RNA FFPE Miniprep kit (Zymo Research, R1008) with Zymo-Spin IC columns (Zymo Research, C1004-250). FACS plots were generated using FlowJo.

**Real-time quantitative PCR.** Reverse transcription was performed using Maxima H Minus Reverse Transcriptase (ThermoFisher, EP0751) with Oligo(dT)$_{18}$ primer (ThermoFisher, SO132) according to the manufacturer's instructions. Transcripts were quantified using Luna Universal quantitative PCR (qPCR) master mix (New England BioLabs, M3003X) with the appropriate primers (Supplementary Data 1) either on a Roche LightCycler 480 or on an Applied Biosystems QuantStudio 6 Flex Real-Time PCR system.

**3DRAM-seq**
**Generation of biotinylated methylation controls.** Methylation controls were generating by first mixing 10 µl of fully methylated pUC19 DNA (Zymo Research, D5017) with 10 µl of unmethylated lambda DNA (Promega, D1521). Control DNA was GpC methylated using the methyltransferase M.CviPI (New England BioLabs, M0227), purified with 1× AMPure XP beads (Agencourt, A63881) and sheared to ~550 bp using a Covaris S220 sonicator. Sticky ends were biotinylated by incubating the sheared DNA for 6 h at 37 °C with DNA polymerase I (New England BioLabs, M0210) and a nucleotide mix containing biotin-14-dATP (Life Technologies, 195245016) in DpnII buffer (New England BioLabs, R0543S) followed by 1× AMPure XP bead purification and quantification using a Qubit dsDNA HS Assay kit (ThermoFisher, Q32851).

**RNA isolation and library preparation.** To generate gene expression data, RNA from approximately $2.5 × 10^4$ fixed mES cells or $10^5$ fixed-sorted RGCs and IPCs was isolated using a Quick-RNA FFPE Miniprep kit (Zymo Research, R1008) in combination with Zymo-Spin IC columns (Zymo Research, C1004-250) according to the manufacturer's instructions starting from the tissue-dissociation step. Yield was quantified using a Qubit RNA HS Assay kit (ThermoFisher, Q32852) and a high RNA quality (RIN > 8) was verified using a Bioanalyzer High Sensitivity RNA 6000 Pico kit (Agilent, 5067-1513). Next, 100 ng of mES cell RNA and around 60 ng of RGC and IPC RNA was used for RNA library generation using a NEBNext Single Cell/Low Input RNA Library Prep kit (New England BioLabs, E6420) according to the manufacturer's instructions.

**Generation of 3DRAM-seq libraries.** To generate 3DRAM-seq libraries, which enables the simultaneous measurement of DNA methylation, accessibility and the 3D genome, approximately $2.5 × 10^5$ mES cells or between 1.5 and $2 × 10^5$ immunoFACS-sorted human RGCs and IPCs were used. Cells were first lysed with 0.2% Igepal-CA630 (Sigma-Aldrich, I3021) for 10 min at room temperature, washed once with 1× GpC buffer (New England BioLabs, M0227S) containing 1% BSA (Sigma-Aldrich, B6917) and subsequently incubated for 3 h at 37 °C in a reaction mix containing 60 U M.CviPI (New England BioLabs, M0227S) and 0.6 mM SAM (New England BioLabs, B9003). During the incubation period, the reaction was substitution with 8 U M.CviPI and 1 µl of 32 mM SAM every hour. Nuclei were washed, permeabilized with 0.5% SDS (Invitrogen, AM9823) quenched with 1.5% Triton-X-100 (Sigma-Aldrich, X100) and digested with 400 U DpnII (New England BioLabs, R0543) overnight at 37 °C. Subsequently, sticky ends were filled by incubating the nuclei for 4 h at room temperature with DNA polymerase I (New England BioLabs, M0210) and a nucleotide mix containing biotin-14-dATP (Life Technologies, 195245016) in DpnII buffer. Proximity ligation was performed for at least 6 h at 16 °C using T4 DNA ligase (New England BioLabs, M0202). Thereafter, nuclei and chromatin were digested using 200 µg proteinase K (New England BioLabs, P8107) with 1% SDS followed by reverse crosslinking overnight at 68 °C with 0.5 M NaCl, purification by ethanol

precipitation and shearing to ~550 bp DNA fragments using a Covaris S220 sonicator. To remove biotinylated ATPs and repair the sticky ends, the sheared DNA was incubated with T4 DNA polymerase (New England BioLabs, M0203) and non-biotinylated nucleotides for 4 h at 20 °C. Approximately 0.01% of biotinylated methylation controls were added to the sample, and bisulfite conversion was performed using an EZ DNA Methylation-Gold kit (Zymo Research, D5005) followed by construction of the sequencing library using a Accel-NGS Methyl-Seq DNA Library kit (Swift Bioscience, 30024, now xGen Methyl-Seq DNA Library Prep IDT, 10009860) according to the manufacturer's instructions until the adapter ligation step. After this step, biotin pulldown was performed using MyOne Streptavidin T1 beads (ThermoFisher, 65602) followed by 5 washes with washing buffer containing 0.05% Tween-20 (Sigma-Aldrich, P9416) and 2 additional washes with low-TE water. To increase library complexity, on-bead final library amplification was performed in five separate reactions using EpiMark Hot Start Taq (New England BioLabs, M0490) with Methyl-Seq Indexing primers (Swift Bioscience, 36024; now IDT, 10009965 or 10005975) with the following PCR program: 95 °C for 30 s; (95 °C for 15 s, 61 °C for 30 s, 68 °C for 80 s) ×10–11; 68 °C for 5 min; hold at 10 °C. The different reactions were pooled, streptavidin T1 beads were pelleted on a magnetic rack and the prepared libraries within the supernatant were purified using 0.65× AMPure XP beads (Agencourt, A63881) to reach an average fragment size of approximately 500 bp. A detailed version of the protocol can be found at protocols.io[63].

**Bisulfite amplicon sequencing of M.CviPI-treated DNA.** To optimize the M.CviPI incubation time, unfixed and fixed mES cells were lysed, washed as described above and incubated with 60 U M.CviPI (New England BioLabs, M0227S) and 0.6 mM SAM (New England BioLabs, B9003) at 37 °C for 10 min up to 4 h. The reactions were substituted with 8 U M.CviPI and 1 µl of 32 mM SAM every hour. Thereafter, nuclei were digested, reverse crosslinked, purified and bisulfite converted as described above. The bisulfite-converted DNA was amplified using EpiMark Hot Start Taq and target specific primers with the following PCR program: 95 °C for 30 s; (95 °C for 30 s, 58 or 52 °C for 30 s, 68 °C for 90 s) ×40; 68 °C for 5 min; hold at 10 °C. Different amplicons of one sample were pooled to equal molarity and purified using 1× AMPure XP beads (Agencourt, A63881). Sequencing libraries were generated from 50 µg purified DNA using a Nextera XT DNA Library Preparation kit (Illumina, FC-131-1024) with half of the recommend reaction volume and 5 min incubation at 55 °C. Final amplification was performed using NEBNext Ultra II Q5 master mix (New England BioLabs, M0544S) with sample-specific indexing primers using the following PCR conditions: 98 °C for 30 s; (98 °C for 10 s, 65 °C for 90 s) ×5; 65 °C for 5 min; hold at 10 °C. Subsequently, the generated libraries were purified using 1.2× AMPure XP beads.

Target-specific primers and indexing primers can be found in Supplementary Data 1.

**MPRA design and plasmid pool generation.** The designed MPRA plasmid pool included 500 scrambled control sequences, which had matched GC content and were pre-screened to minimize the presence of expressed TF motifs and 2,737 DARs that interact with a differentially expressed gene in at least one cell type in human organoids as well as 267 MER130 or UCON31 TEs. Additionally, we added 2,372 enhancer sequences for which only the corresponding motif sequence was iteratively mutated (100 permutations with similar GC content, lowest motif score selected). DARs were centred on the accessibility peak and resized to 266 bp, and nucleotide sequences were extracted using the 'BSgenome.Hsapiens.UCSC.hg38' R package. To facilitate barcode–CRE association, we added a 4 bp tag at the beginning of each WT/control (TCAG) or Mut (GTCA) sequence.

The MPRA plasmid pool was generated as previously described[3], and a detail protocol can be found at https://www.protocols.io/view/mpra-plasmid-pool-preparation-bxchpit6/. In brief, 300 bp

single-stranded oligonucleotides were synthesized (Twist Bioscience), and degenerate barcodes as well as KpnI/EcoRI restriction sites were added using two separate PCRs. PCR products were introduced into the pMPRA1 (ref. 64; Addgene, plasmid 49349) backbone through Gibson assembly and transformed into ElectroMAX Stbl4 competent cells (ThermoFisher, 11635018) using Gene Pulser/MicroPulser electroporation cuvettes with a 0.1 cm gap (parameters: 1.8 kV, 25 µF, 200 Ω). Transformed bacteria were immediately resuspended in 1 ml of warm SOC medium and a 1:10 dilution of the bacteria was distributed on 10 plates of LB agar containing 100 µg µl$^{-1}$ carbenicillin. Transformant number was estimated on a 1:100,000 diluted counting plate, and the required number of colonies to achieve the targeted library complexity was scraped for plasmid purification (Qiagen, 27104). The purified plasmids were digested using KpnI/EcoRI and ligated with an insert containing the minimal promoter and mScarlet-I. The purified ligation product was transformed into *Escherichia coli* as described above, scraped, and the final plasmid library was purified using a EndoFree Plasmid Maxi kit (Qiagen, 12362). Single CRE constructs were synthesized (Twist Bioscience) directly with the Gibson overhangs as well as KpnI/EcoRI restriction sites and cloned as described above. The resulting construct were transformed into NEB Turbo Competent *E. coli* (NEB, C2984I) and purified using an EndoFree Plasmid Maxi kit (Qiagen, 12362).

All primers and DNA blocks used are listed in Supplementary Data 1.

**MPRA and CRE barcode association library generation.** MPRA libraries were prepared as previously described[3] with minor modifications. In brief, RNA and DNA from fixed immunoFACS-sorted cells were extracted using a Quick-DNA/RNA Microprep Plus kit (Zymo Research, D7005) according to the manufacturer's instructions. Purified RNA was treated with TURBO DNase (ThermoFisher, AM1907) and reverse transcribed with Maxima H Minus RT (ThermoFisher, EP0753) using Oligo(dT)$_{18}$ Primer (ThermoFisher, SO132). cDNA was purified using 1.5× AMPure XP magnetic beads (Agencourt, A63881). For both the DNA and cDNA libraries, unique molecular identifiers (UMIs) were added by PCR (98 °C for 30 s; (98 °C for 10 s, 65 °C for 30 s, 72 °C for 1 min) × 3; 72 °C for 3 min, and hold at 4 °C) using the primers RV_univ_MPRA and FWD_mScar_Tn7_10UMI_3 (0.5 µM each). P7 and P5 dual indexing sequencing adaptors (0.1 µM each) were attached separately by first amplifying the library with Ad2.X (ref. 65) and P5NEXTPT5 primers using the following PCR program: 98 °C for 30 s; (98 °C for 10 s, 65 °C for 90 s) × (10× for DNA or 12× for cDNA); 72 °C 5 min, and hold at 4 °C. PCR products were purified and amplified using Ad2.X and P5NEXT_SX primers for additional PCR cycles (minimum 12) determined by qPCR and using one-tenth of the first PCR product as input. All PCRs were performed in 1× NEBNext Ultra II Q5 master mix (New England BioLabs, M0544), reactions were split into two separate reaction tubes to increase library complexity and PCR products were pooled and purified using 0.8× to 1.2× AMPure XP magnetic beads. Final libraries were quantified using Qubit (ThermoFisher) and Bioanalyzer 2100 (Agilent). For the CRE barcode association library, 5 ng of the plasmid pool without minimal promoter and mScarlet-I was used to attach P5 and P7 dual indexing sequencing adaptors in two separate PCRs. Both PCRs were performed using NEBNext Ultra II Q5 master mix (New England BioLabs, M0544) with RV_univ_MPRA + FWD_CRS_Tn7 (0.5 µM each; PCR conditions: 98 °C for 30 s; (98 °C for 10 s, 65 °C for 30 s and 72 °C for 3 min) × 3; 72 °C for 3 min, and hold at 4 °C) and P5NEXT_SX + Ad2.X (0.1 µM each; PCR conditions: 98 °C for 30 s; (98 °C for 10 s, 65 °C for 90 s) × 10; 72 °C for 5 min, and hold at 4 °C), respectively.

**Generation of hiPS cell line for MPRA.** The hiPS cell line CRTDi004-A (Human Pluripotent Stem Cell Registry) was generated from previously published foreskin fibroblasts (termed Theo) of a consenting healthy donor[66]. Isolation of cells and reprogramming to hiPS cells was approved by the ethics council of the Technische Universität

Dresden (EK169052010, EK386102017). Theo fibroblasts were reprogrammed at the CRTD Stem Cell Engineering Facility at Technische Universität Dresden using a CytoTune-iPS 2.0 Sendai Reprogramming kit (ThermoFisher, A16517) according to the supplier's recommendations for transduction. Following transduction with the Sendai virus, cells were cultured on irradiated CF1 mouse embryonic fibroblasts (ThermoFisher, A34180) in KOSR-based medium (80% DMEM/F12, 20% KnockOut Serum Replacement, 2 mM L-glutamine, 1% nonessential amino acids, 0,1 mM 2-mercaptoethanol, all from ThermoFisher, 11330-032, 10828028, 25030149, 11140050 and 31350010, respectively) supplemented with 10 ng ml$^{-1}$ human FGF2 (StemCell Technologies, 78003). Individual iPS cell colonies were mechanically picked, expanded as clonal lines and adapted to Matrigel (Corning, 354277), mTeSR1 and ReLeSR (both StemCell Technologies, 85850 and 05872, respectively) conditions after several passages. Master and working hiPS cell stocks were established from the clone with the best morphology.

To characterize the newly generated CRTDi004-A hiPS cell line, the following tests were performed: for flow cytometry analysis of pluripotency, Alexa Fluor 488 anti-Oct3/4, PE anti-Sox2, V450-SSEA-4, and Alexa Fluor 647 anti Tra-1-60 (all from BD Biosciences, 560253, 560291, 561156 and 560122, respectively) were used according to the manufacturer's recommendations. Three germ layer differentiation was performed as previously described[67], and resulting cells were stained using a 3-Germ Layer Immunocytochemistry kit (ThermoFisher, A25538) according to the manufacturer's instructions. For endoderm, SOX17 primary antibody (Abcam, ab84990) followed by Alexa Fluor 488 goat anti-mouse IgG (ThermoFisher, A32723) was used. qPCR with reverse transcription for pluripotency and tri-lineage spontaneous differentiation was performed according to the instruction manual of the human ES cell Primer Array (Takara Clontech). Standard G banding karyotyping was done in collaboration with the Institute of Human Genetics, Jena University Hospital, Germany, and 20 metaphases were analysed.

**Electroporation of human cortical organoids.** Human cortical organoids were generated following a previously reported protocol[68] using the hiPS cell line CRTDi004-A cultured in standard conditions (37 °C, 5% $CO_2$) on Matrigel-coated plates (Corning, 354277) and approved by the ethics council of the Technische Universität Dresden (SR-EK-456092021). Electroporation of the MPRA library was carried out 2 days after the first slicing on day 45 of organoid culture. Organoids were transferred to a 6 cm ultra-low-attachment dish (Eppendorf, 30701011) containing Tyrode's solution (Sigma-Aldrich, T2145). Using a glass microcapillary (Sutter Instrument, BF120-69-10), 0.2–0.5 µl of the plasmid DNA (either MPRA library or an equal molar mix of single CRS constructs with CAG-GFPnls control plasmids[3]) at a final concentration of 1 µg µl$^{-1}$ diluted in 0.1% Fast Green solution (in $dH_2O$) were injected into areas depicting ventricular morphology. Injections were carried out using a microinjector (World Precision Instruments, SYS-PV820) on continuous setting. Up to five ventricles were injected per organoid. A total of 35–40 organoids were processed per replicate depending on the size and number of ventricular structures. After injection, organoids were transferred into an electroporation chamber containing Tyrode's solution and electroporated with 5 pulses applied at 38 V for 50 ms each at intervals of 1 s (Harvard Bioscience, BTX ECM 830). Subsequently, the electroporated organoids were returned to culture medium and incubated for 72 h before further processing. Organoids were dissociated using a MACS neural tissue dissociation kit P (Miltenyi Biotec, 130-092-628) with a reduced incubation time of enzyme mix 1 (6 min at 37 °C) and omitting the incubation with enzyme mix 2.

**Library quality control and sequencing.** Libraries were quantified by qPCR using a NEBNext Library Quant kit (New England BioLabs, E7630), and the size distribution of the obtained libraries was assessed using an Agilent 2100 Bioanalyzer. Sequencing was performed on

a NextSeq550 or NovaSeq6000. Sequencing statistics are listed in Supplementary Table 2.

## Bioinformatics analysis

**Mapping and analysis of gene expression.** RNA sequencing libraries were mapped and deduplicated using STAR[69] with default settings. DESeq2 (ref. [70]) was used to calculate fragments per kilobase of transcript per million mapped read (FPKM) and differential expressed gene values (FDR < 0.05). Gene body coverage and transcriptomic distribution was computed using RSeQC[71].

**Mapping of 3DRAM-seq.** For mapping of 3DRAM-seq results, we used an adapted TAURUS-MH[9] pipeline, which includes read splitting based on the ligation junction, mapping with Bismark and improved quality control. Next, 100 bp paired-end reads were first trimmed using Trim Galore with the following parameters: --nextseq 30 --clip_R1 1 --clip_R2 15 --length 20. Subsequently, reads where aligned using Bismark[72] with Bowtie2 in single-end mode and the post-bisulfite adapter tagging option (--pbat) for the reverse read. To recover chimeric reads resulting from the proximity ligation step, and are therefore not aligned during the previous step, unmapped reads from the previous step were split at adjacent DpnII cutting sites (GATTGATT, GATTGATC for forward and AATCAATC, GATCAATC for reverse strand, including variations where endogenous C is not methylated and thus converted to T), separately aligned and subsequently merged with the non-chimeric reads. The unique read pairs were transformed into restriction fragment end (fend) coordinates, converted into 'misha' tracks and imported into the corresponding genomic database (mm10 or hg38). Methylation levels in both, CpG and GpC context, were calculated on uniquely mapped reads using the Bismark methylation extractor and coverage2cytosine function with the --nome-seq option on, ensuring that only cytosines in the correct context are considered. Only 5× for individual replicates or 10× for merged replicates covered cytosines were considered for further analysis. Data were mapped using the mm10 genome for mES cells and the hg38 genome for RGCs and IPCs.

**Mapping of external datasets.** To compare 3DRAM-seq results with comparable multiomics datasets (Methyl-3C, Methyl-HiC and WGBS), raw data were downloaded and processed using the adapted TAURUS-MH with dataset-specific modifications mainly during the trimming step. For Methyl-HiC[10], the parameters --clip_R1 1 and --clip_R2 1 were used to account for the pre-bisulfite adapter ligation step, which does not introduce low complexity tails. To account for the random primer amplification step and therefore template switch of Methyl-3C reads were trimmed with -a AGATCGGAAGAGCACACGTCTGAAC -a2 AGATCGGAAGAGCGTCGTGTAGGGA --clip_R1 16 --clip_R2 16 --three_prime_clip_R1 3 --three_prime_clip_R2 3 to remove the low-complexity 5' tail induced by the Adaptase and random primer sequence and adapter from the 3' prime end. Additionally, read 1 instead of read 2 was flagged using --pbat during the alignment steps. WGBS[27] data were trimmed and directly aligned using Bismarck with Bowtie2 in PE mode.

ChIP–seq and ATAC–seq datasets were uniformly processed using the ENCODE ChIP–seq or ATAC–seq pipeline, respectively, whereas the Hi-C data were processed as previously described[5]. DHS and MNase-seq data were directly downloaded from Encode.

**Estimation of bisulfite conversion efficiency.** The efficiency of bisulfite conversion was estimated through the CpG methylation of unmethylated lambda DNA using Bismark in paired-end mode with the --nome-seq option. The detection rate of methylated cytosines both in the CpG and GpC context was determined by fully methylated pUC19 DNA as well as in situ GpC methylated lambda DNA. In all cases, we observed methylation above 98%, indicating a false negative rate of less than 2%.

**Co-accessibility and co-methylation analysis at single-molecule resolution.** Sequencing reads were first split per chromosome into individual data frames containing only relevant fields such as read name, read pair identity and exact coordinates of methylation calls. We used the FST R package to reduce memory footprint of the full dataset and to facilitate downstream processing and to further improve read and write speed. To interrogate accessibility and/or methylation at multiple loci, bed files containing coordinates for the desired parameters were prepared (for example CTCF motifs overlapping ChIP–seq peaks). These regions were centred on, for example, transcription factor motifs (filter intervals), at which methylation and/or accessibility calls will be computed.

To search for reads containing methylation calls that fell within the region of interest, a binary search was executed to identify the closest filter interval existing in genomic space for each read. Next, the absolute distance (in base pairs) between each methylation call and the nearest filter interval was computed. Following that, a user-defined window (100 bp) was used as a threshold, at which methylation calls that lie within this distance were retained, whereas the rest were discarded. Overall, only reads that contained at least one methylation call were retained. The search and filter process was repeated separately and iteratively for read pairs 1 and read pairs 2. The final result was obtained by merging based on full read name to ensure only read pairs overlapping both filter intervals were kept.

To determine paired co-accessibility patterns, the average accessibility (based on GpC methylation) was calculated separately for read 1 and read 2 in a chosen window centred on the feature of interest, such as CTCF motif, and separated by a minimum distance. The resulting two-column matrix was then used as input for k-means clustering, clusters were reordered on the basis of their mean value for consistency and the matrix was plotted using the R package ComplexHeatmap. All subsequent analysis was performed using exactly the same cluster assignments.

To test whether there was a dependency between accessibility in read 1 and read 2, we used the Fisher exact test on the 2 × 2 contingency matrix, and we report the odds ratio and P values. This approach aims to test whether the null hypothesis (accessibility at read 1 and read 2 are independent events) can be rejected. The analysis for Fig. 5j,k was performed analogous to the CTCF-based analysis in Fig. 3b. First, we filtered LHX2–SOX2 or NEUROG2–EOMES motifs, retaining only those that overlapped with a GpC peak (based on bulk accessibility in RGCs or IPCs, respectively). Next, we identified all read pairs for which read 1 overlapped with one of the motifs (for example LHX2) and read 2 overlapped with the other motif (for example, SOX2). We then measured the average accessibility per read within a 50 bp window for reads that are separated by at least 100 bp but not more than 300 bp. This distance cut-off is different from our measurements of long-range interactions associated with CTCF loops because we wanted to determine whether these pairs of TFs interact directly or co-bind on chromatin synergistically at closer distances.

**Visualization of linear marks at genomic features and GO term enrichment.** Average enrichment plots and heatmaps of DNA methylation, chromatin accessibility or ChIP–seq in windows centred around the genomic feature were visualized using SeqPlots[73]. Functional enrichment analysis was performed using Cluster profiler and visualized with enrichPlot[74].

**Identification and characterization of DARs and differential methylated regions.** Accessible peaks based on GpC methylation were identified using the gNOMeHMM package[26] with default settings (q value ≤ 0.05), which resulted in 67,177 peaks for mES cells, 39,738 peaks for RGCs and 54,334 peaks for IPCs. Accessible peaks of RGCs and IPCs were merged to generate a common peak set (66,280 peaks). DARs and differential methylated regions (DMRs) were identified

in the common peak set using methylKit[75] with following settings: lo.count=10, hi.perc=99.9, overdispersion="MN", test="Chisq", qvalue=0.05. Peaks within promoter regions (±5 kb from the TSS) were associated with their nearest TSS, whereas distal peaks were associated with genes within the same TAD displaying the highest Hi-C score with a minimal and maximum distance of 5 kb and 2 Mb, respectively.

**TF motif analysis.** For motif-based analysis, we used the JASPAR2022 core vertebrate database and excluded all TFs that were not expressed in our data (FPKM < 1). TF factor motif enrichment was either calculated using the CreateMotifMatrix function from the Signac package[76] or using the monaLisa package[40]. Motifmatchr was then used to identify TF motifs within genomic regions (p.cutoff = 0.0005) and to centre the region around them.

**Repetitive element analysis.** Localization of repetitive elements for the hg38 genome were obtained from RepeatMasker and repeats classified as satellite, simple_repeat, tRNA, rRNA, snRNA, srpRNA or low_complexity were removed. Individual repeats were associated to genes as described for DARs.

**Hi-C data processing.** The filtered fend-transformed read pairs obtained from the TAURUS-MH pipeline were converted into tracks and imported into the genomic databases. Normalization was performed using the Shaman package (https://tanaylab.bitbucket.io/shaman/index.html), and Hi-C scores were calculated using a kNN strategy on the pooled replicates as previously described[5] with a kNN of 100. For visualization, fend-transformed read pairs were converted into .hic files using Juicer pre and displayed using Juicebox[77]. HiCRep[78] was used to calculate reproducibility between biological replicates and datasets.

**Contact probability, insulation, TAD boundary calling and average TAD contact enrichment.** Contact probability as a function of the genomic distance was calculated as previously described[5]. To define insulation based on observed contacts, we used the insulation score[5,79], which was calculated on the pooled contact map at 1 kb resolution within a region of ±250 kb and was multiplied by (−1). TAD boundaries were then defined as the local 2 kb maxima in regions where the insulation score was above the 90% quantile of the genome-wide distribution. Differential TAD boundaries were identified as previously described[5] using genome-wide normalized insulation scores. To calculate insulation and contact enrichment within TADs, their coordinates were extended upstream and downstream by the TAD length, and this distance was split into 100 equal bins. The observed versus expected enrichment ratio was calculated in each resulting 100 × 100 grid (per TAD) and the average enrichment was plotted per bin. Average DNA methylation and accessibility levels were calculated for each of these 100 bins per TAD and are represented as the mean ± 0.25 quantiles.

**Compartments and compartment strength.** The dominant eigenvector of the contact matrices (250 kb bins) were computed as previously described[80] using scripts available at https://github.com/dekkerlab/cworld-dekker/. Compartment strength was determined by the $\log_2$ ratio of observed versus expected contacts (intrachromosomal separated by at least 10 Mb) either between domains of the same (A–A, B–B) or different types (A–B), as previously described[5] and represents the ratio between the sum of observed contacts within the A and B compartments and the sum of intercompartment contacts (AA + BB)/(AB + BA).

**Aggregated and individual contact strength at pairs of genomic features.** Contact enrichment ratios between pairs of genomic features, such as motif-centred differential accessible regions, were calculated using two complementary approaches[3]. First, Hi-C maps were aggregated to calculate the $\log_2$ ratio of the observed versus expected contacts within a window centred on the pair of interest. In addition,

the average enrichment ratio of the contact strength in the centre of the window (central nine bins) versus each of the corners was calculated. Second, to analyse the heterogeneity of the data and the contribution of individual pairs, we extracted the kNN-based Hi-C score in a 10 kb window centred around each of the pairs separately and represented the data as a scatterplot or boxplot. Significance was then calculated using the Wilcoxon rank test.

**MPRA CRE–barcode association.** For CRE–barcode association, 75 bp pair-end reads were trimmed using cutadapt with the following parameters: -m 12 -a GAATTCATCTGGTA -G GACCGGATCAACT -u 1 --discard-untrimmed. Next, 150 bp paired-end reads were first filtered using cutadapt (-m 12 -a GAATTCATCTGGTACCTCGGTTCACG-CAATG -G ^CCAGGACCGGATCAACT -u 1 --discard-untrimmed --action=none --interleaved | cutadapt -g GAATTCATCTGGTACCTCG-GTTCACGCAATG -G ^CCAGGACCGGATCAACT --discard-untrimmed --action=none –interleaved). Subsequently, forward and reverse reads were individually trimmed using -l 12 for the barcode, -g ^CCAGGAC-CGGATCAACT–discard-untrimmed for the forward CRE reads or -g GAATTCATCTGGTACCTCGGTTCACGCAATG --discard-untrimmed -l 105 for the reverse CRE read. Trimmed fastq reads for both 75 bp and 150 bp pair-end reads were separated based on a 5′ 4 bp identifier (GTCA or TCAG) and CRE–barcode association was performed separately on wild-type, mutant sequences using MPRAflow[81]. The resulting pickle libraries were merged to increase the number of recovered CREs and filtered for promiscuous barcodes.

**MPRA data processing.** The 150 bp paired-end reads from the cell-type-specific DNA and RNA libraries were trimmed using cutadapt with the following parameters: -m 12:10 -e 0.4 -u 1 -a GAATTCTCATTAC -A TCGACCGCAAGTTGG --discard-untrimmed. Read 1 was additionally trimmed with -l 12. Count tables for RNA and DNA reads were generated using MPRAflow (--bc-length 12 and --mpranalyze). MPRAanalyse[82] was used to calculate the MPRA signal (mad.score) and to identify significant active enhancers (mad.score BH adjusted $P$ value ≤ 0.1). For comparison of replicates, the normalized DNA/RNA read counts and ratio of sums were calculated as previously described[83].

**Statistic and reproducibility.** No statistical methods were used to predetermine sample sizes, but our sample sizes are similar to those reported in previous publications[3,5–10]. No data were excluded from the analyses. Data collection and analysis were not performed blind to the conditions of the experiment.

**Reporting summary**
Further information on research design is available in the Nature Portfolio Reporting Summary linked to this article.

## Data availability

Sequencing data that support the findings of this study have been deposited into the GEO database under accession number GSE211736. Previously published data that were re-analysed here are available under the following accession codes from the GEO database: GSE196084 (ref. 25) and GSE96107 (ref. 5) for RNA sequencing; GSE119171 (ref. 10) for Methyl-HiC; GSE124391 (ref. 9) for Methyl-3C; GSE112520 (ref. 27) for WGBS; GSE96107 (ref. 5) for Hi-C; GSE113592 for ATAC–seq; GSE51336 (ref. 57) for DHS; GSE58101 (ref. 58) for MNase-seq; GSE96107 (ref. 5) and GSE116825 (ref. 33) for CTCF ChIP–seq; GSE63621 (ref. 37) for NEU-ROG2 ChIP–seq; GSE67539 (ref. 84) for NEUROD2 ChIP–seq; GSE130275 (ref. 55) for Micro-C; and GSE67867 (ref. 23) for NRF1 ChIP–seq. H3K27ac ChIP–seq data were from ENCODE (ENCSR000CGQ). SMC1 ChIP–seq data were from ref. 60, and mES cell ChromHMM data were from ref. 85. All other data supporting the findings of this study are available from the corresponding author on reasonable request. Source data are provided with this paper.

## Code availability

The code used for generating the data and all the figures are freely available at https://github.com/BonevLab/NoackVangelisti_2023. The R package to compute the expected tracks and the Hi-C scores is available at https://github.com/tanaylab/shaman.

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

## Acknowledgements

Sequencing was performed at the Helmholtz Zentrum München by staff at the NGS-Core Facility and cell sorting was done at the Flow Cytometry Core Facility at the Biomedical Center, Ludwig-Maximilians-Universität. We acknowledge M. A. Lee-Kirsch and staff at the CRTD Stem Cell Engineering Facility at TU Dresden for the generation of the CRTDi004-A hiPS cell line. M.A. was supported by the Center for Regenerative Therapies TU Dresden, the DFG (Emmy Noether, AL 2231/1-1), the Schram foundation, ERA-NET Neuron (MEPIcephaly) and by the Federal Ministry of Education and Research (01EW2208). Work in the group of B.B. was supported by the Helmholtz Center Munich, DFG priority programme SPP2202 (BO 5516/1-1), ERA-NET Neuron (MOSAIC) and European Research Council Consolidator grant to B.B. (EpiCortex, 101044469).

## Author contributions

F.N. and S.V. performed experiments and data analyses. F.N. and N.D. performed the MPRA. F.N., S.V., F.C. and B.B. analysed the data. M.A. and B.B supervised the project. F.N., S.V. and B.B. wrote the manuscript with input from all authors.

## Funding

## Competing interests

The authors declare no competing interests.

## Additional information

**Extended data** is available for this paper at https://doi.org/10.1038/s41556-023-01296-5.

**Correspondence and requests for materials** should be addressed to Boyan Bonev.

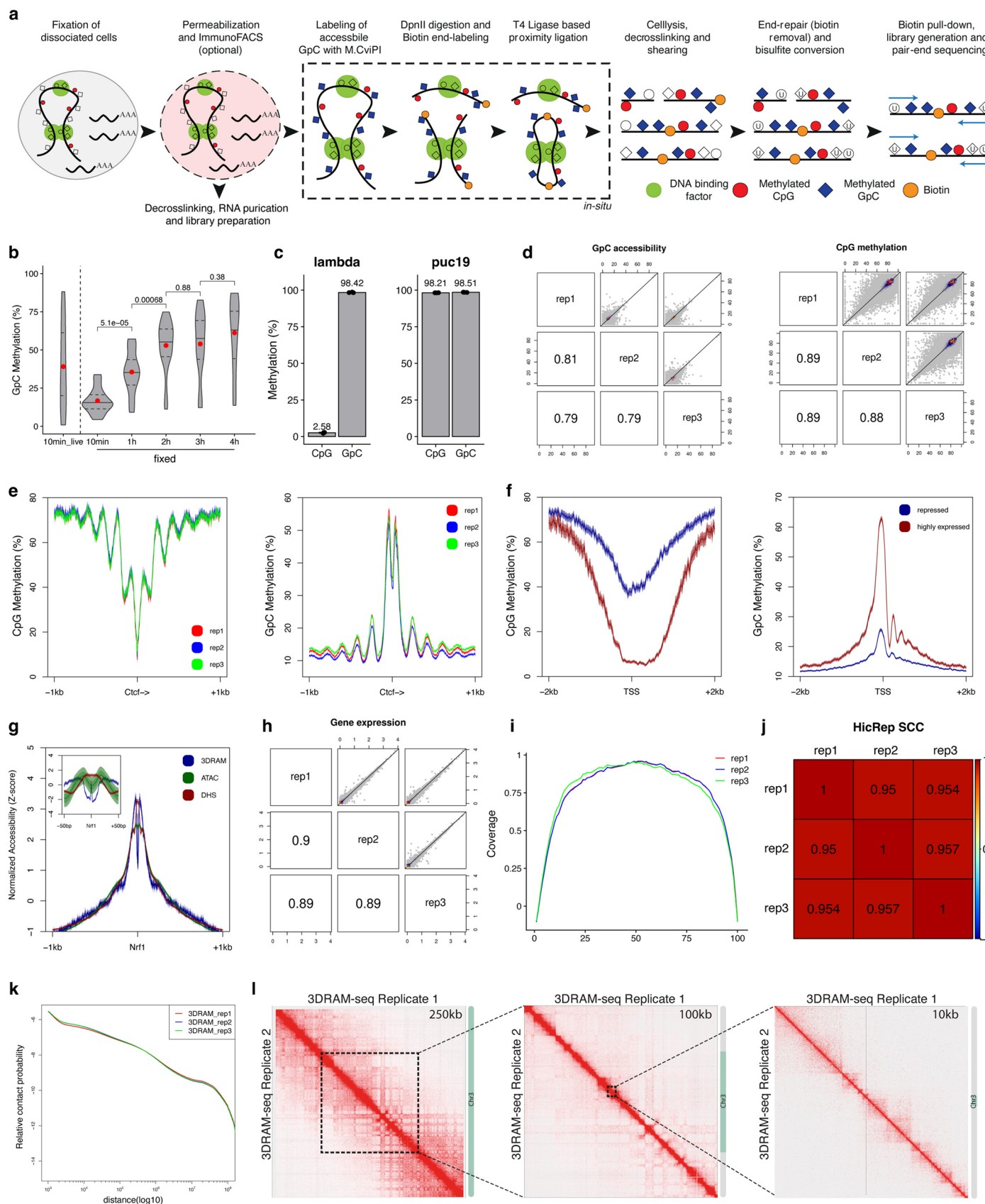

**Extended Data Fig. 1 | See next page for caption.**

**Extended Data Fig. 1 | 3DRAM-seq experimental overview and additional quality metrics. (A)** Experimental procedure of 3DRAM-seq **(B)** Violin plot depicting GpC methylation levels of live or fixed cells incubated for 10 min up to 4 h with M.CviPI. For each timepoint, five amplicons with a total of 17 to 29 GpC (coverage ≥50x) where analyzed. Statistical significance is calculated using wilcoxon rank-sum test between adjacent timepoints. Red dots: means; black lines: medians; black dotted lines: ±STD. **(C)** Bar plot depicting methylation levels of lambda (only GpC methylated) and fully methylated puc19 DNA spike-in controls. Dots: mean of individual replicates; Numbers indicate mean of all replicates (n = 3). **(D)** Pairwise correlation matrixes displaying Pearson's correlation coefficient as well as scatterplot of GpC accessibility and DNA methylation in 1 kb bins. **(E)** Average CpG methylation and GpC accessibility levels across motif centered Ctcf ChIP-seq peaks for individual replicates (5 bp bin size, at least 5x coverage). **(F)** Average CpG methylation and GpC accessibility levels of repressed (FPKM < 1) or highly expressed (top 25% expressed genes) genes centred at TSS (20 bp bin size, 10x coverage). **(G)** Average accessibility levels at motif centered Nrf1 ChIP-seq peaks (2 bp bins). **(H)** Pairwise correlation matrixes displaying correlation coefficient (Spearman), as well as scatterplot for gene expression. **(I)** Coverage of RNA-seq reads per replicate across gene bodies of housekeeping genes calculated by RSeQC. **(J)** Pairwise correlation matrixes displaying 3D genome correlation coefficient (stratum adjusted correlation coefficient, 10 kb bins, calculated by HiCRep). **(K)** Contact probability in logarithmic bins. **(L)** Contact maps (KR observed) of chromosome 3 for replicate 1 and 2. Source numerical data are available in source data.

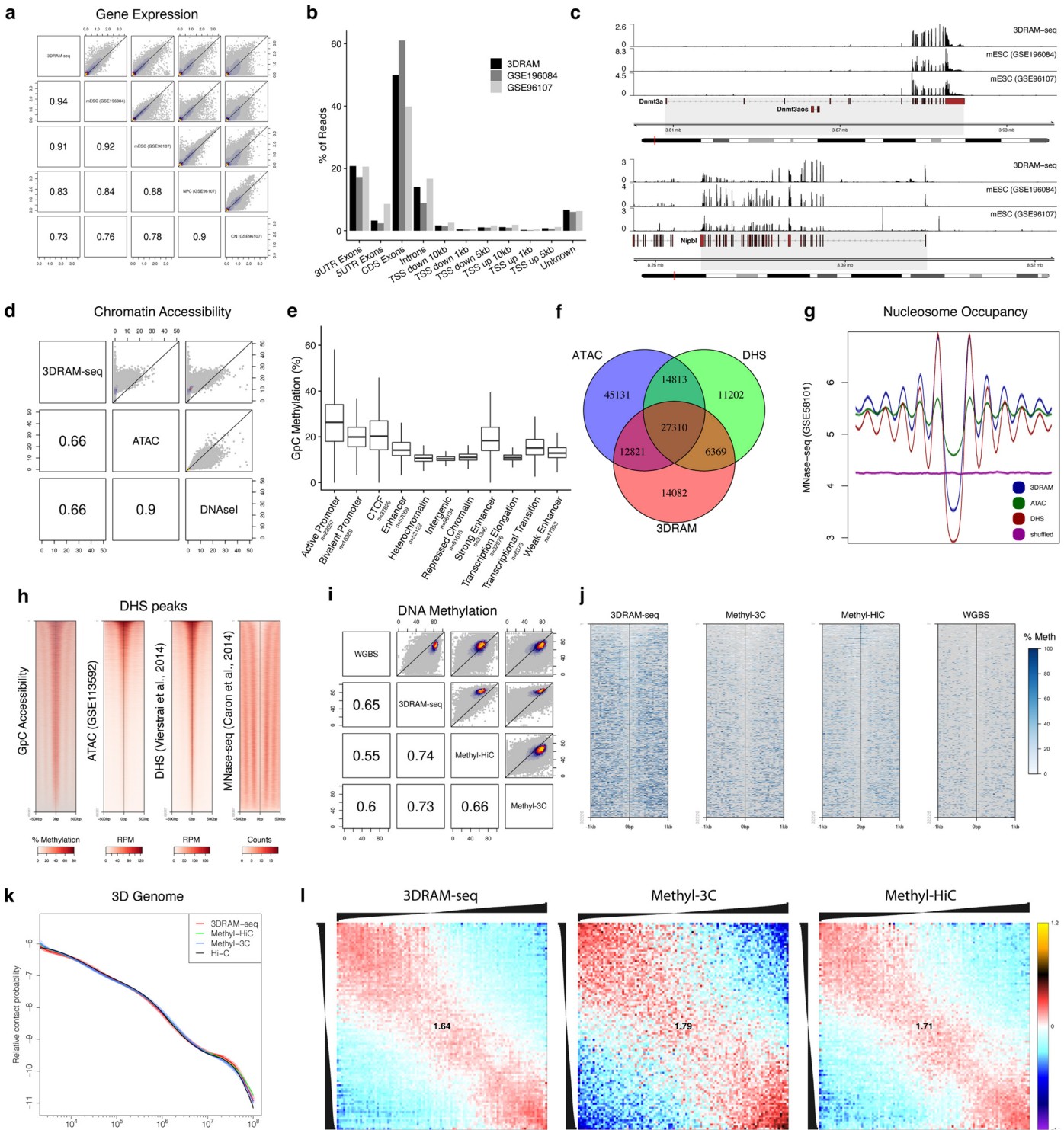

**Extended Data Fig. 2 | Comparison of 3DRAM-seq with other methods.**
(A) Pairwise correlation matrixes displaying Spearman's correlation coefficient as well as scatterplot between different RNA-seq datasets. (B) Distribution of reads across genomic features. (C) Representative examples of RNA coverage comparisons across two long, multi-exon genes. (D) Same as A but for DNA accessibility (bin size 10 kb) datasets. (E) GpC methylation levels across different ChromHMM features (n refers to the number of GpC sites analyzed per feature). (F) Venn diagram with numbers of overlapping ATAC-seq, DHS and 3DRAM-seq GpC peaks. (G) Nucleosome occupancy profiles centred at either

3DRAM-seq GpC, ATAC-seq or DHS peaks. Shown is also the profile at randomly shuffled regions as control. (H) Heatmaps showing accessibility measured by GpC methylation, ATAC-seq, DHS, as well as nucleosome occupancy (MNase-seq) across DHS peaks (I) Same as A but for DNA methylation (bin size 10 kb) (J) Heatmaps displaying methylation levels measured by different methods at individual motif centered CTCF peaks (50 bp bin size). (K) Contact probability in logarithmic bins. Lines: mean values from different methods; semitransparent ribbons: SEM. (L) Average contact enrichment at pairs of 250 kb loci arranged by their eigenvalue (shown on top). Numbers represent the compartment strength.

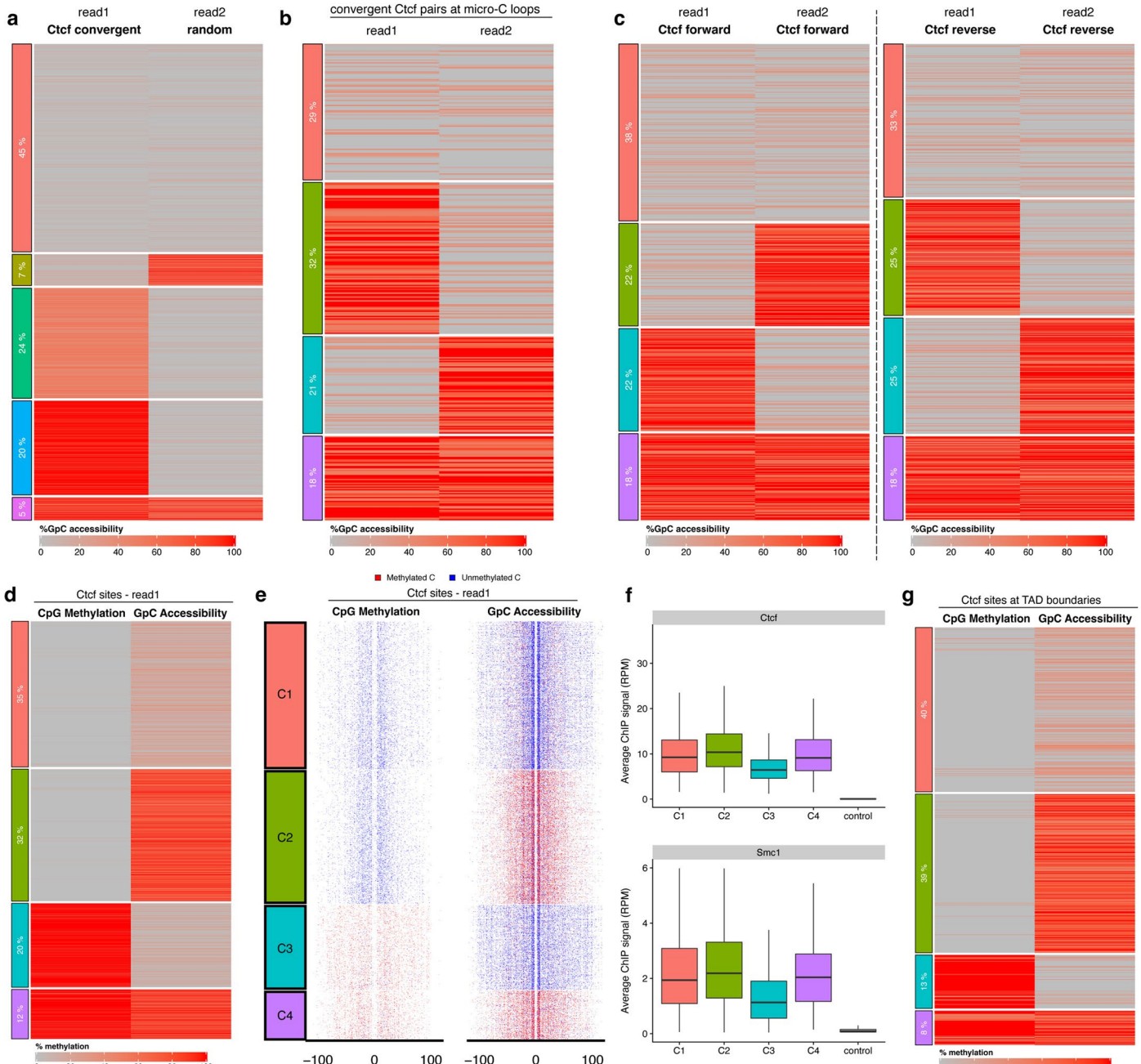

**Extended Data Fig. 3 | Paired single molecule accessibility across different genomic features. (A-C)** Clustered average co-accessibility between different sets of read pairs: a Ctcf motif and random region (A), pairs of convergent Ctcf pairs overlapping with chromatin loops identified by micro-C (B) as well as non-convergent Ctcf motifs (C). **(D)** Co-occurrence of CpG methylation and GpC accessibility at individual reads containing the Ctcf motif. **(E)** Same as (D) but with methylation levels of individual CpG or GpC dinucleotides of each read. **(F)** Boxplots showing ChIP-seq signal for Ctcf or Smc1 across the same clusters as in (D), as well as random shuffled regions (n = 1217, 1051, 727, 421 and 3400 regions respectively). Boxplots display median (line), 25th or 75th percentiles (box) as well as 10th or 90th percentiles (whisker). **(G)** Same as (D) but for Ctcf motifs overlapping a TAD boundary.

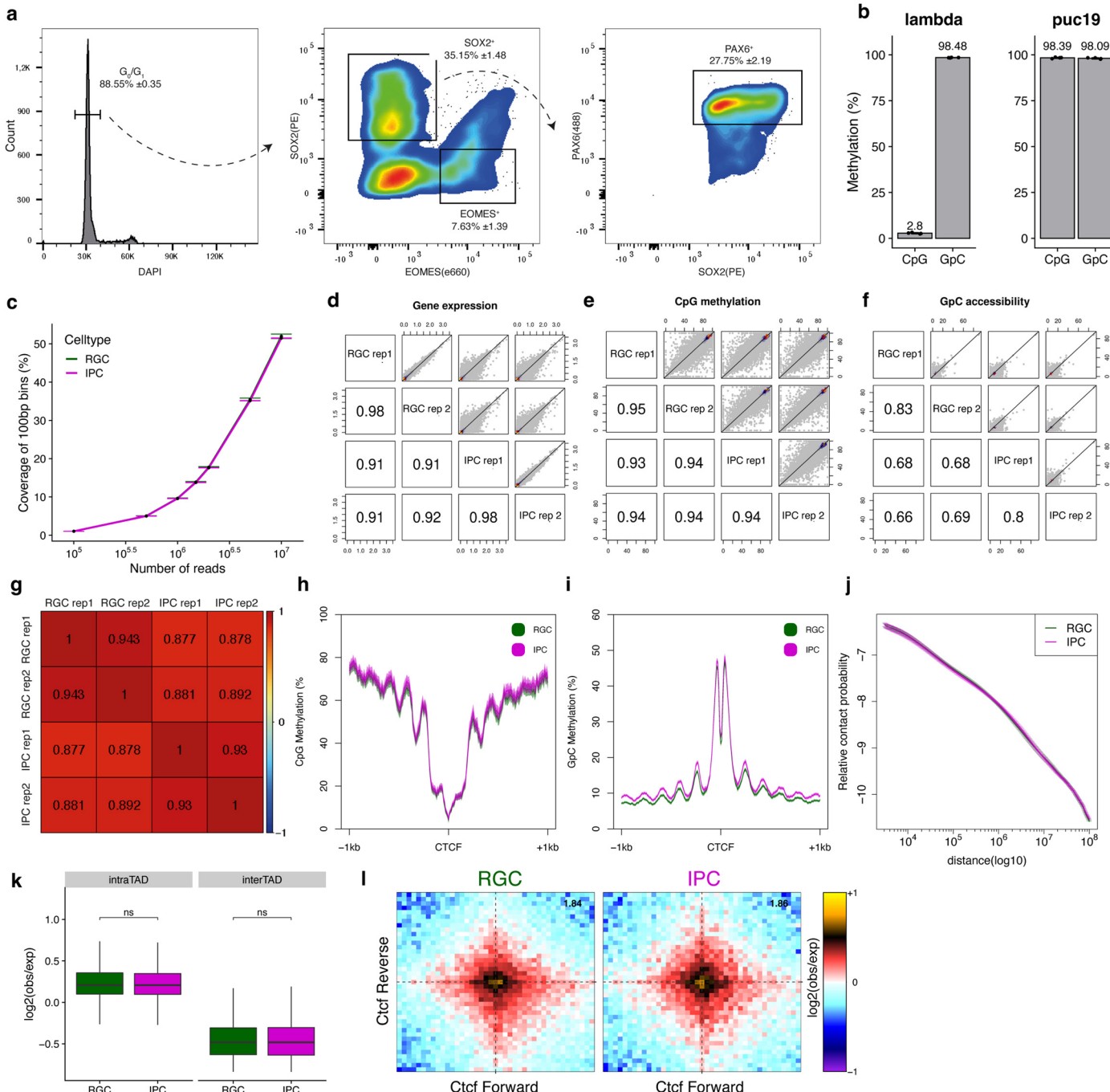

**Extended Data Fig. 4 | 3DRAM-seq in purified neural progenitors from human cortical organoids. (A)** Gating strategy for the immunoFACS of RGC and IPC from cortical organoids. From left to right: Singlets in $G_0/G_1$ based on their DNA content (DAPI) were selected first, followed by separation of nuclei positive for SOX2 or EOMES. SOX2 positive cells were further subdivided based on PAX6. Number represent mean ± SD from the parental singlet population. **(B)** Bar plot depicting measured methylation levels of lambda (only GpC methylated) and fully methylated puc19 DNA spike-in controls. Black dots: mean of individual biological replicates (n = 2 for each condition); Numbers: mean methylation levels. **(C)** DNA methylation coverage (100 bp bins) for RGC and IPC. Black dot and whiskers indicate mean ± SD (n = 2). **(D-G)** Pairwise correlation matrixes displaying correlation coefficient and/or scatterplot for gene expression

(D; Spearman's), DNA methylation (E; Pearson, 10 kb bins), GpC accessibility (F; Pearson, 10 kb bins) and 3D genome (G; stratum adjusted correlation coefficient, 10 kb bins). **(H-I)** CpG methylation (H) and GpC accessibility (I) levels at motif centered CTCF ChIP-seq peaks derived from the whole human cortex. **(J)** Contact probability in logarithmic bins for RGC and IPC. Lines: mean values from different methods; semi-transparent ribbons: SEM. **(K)** Boxplots displaying quantification of intraTAD and interTAD contact enrichment in RGC and IPC (n = 2939 TADs). Statistical significance is calculated using a two-sided paired *t*-test. Boxplots display median (line), 25th or 75th percentiles (box) as well as 10th or 90th percentiles (whisker). **(L)** Average contact strength between intraTAD pairs of GpC peaks containing convergent orientated CTCF motifs. Source numerical data are available in source data.

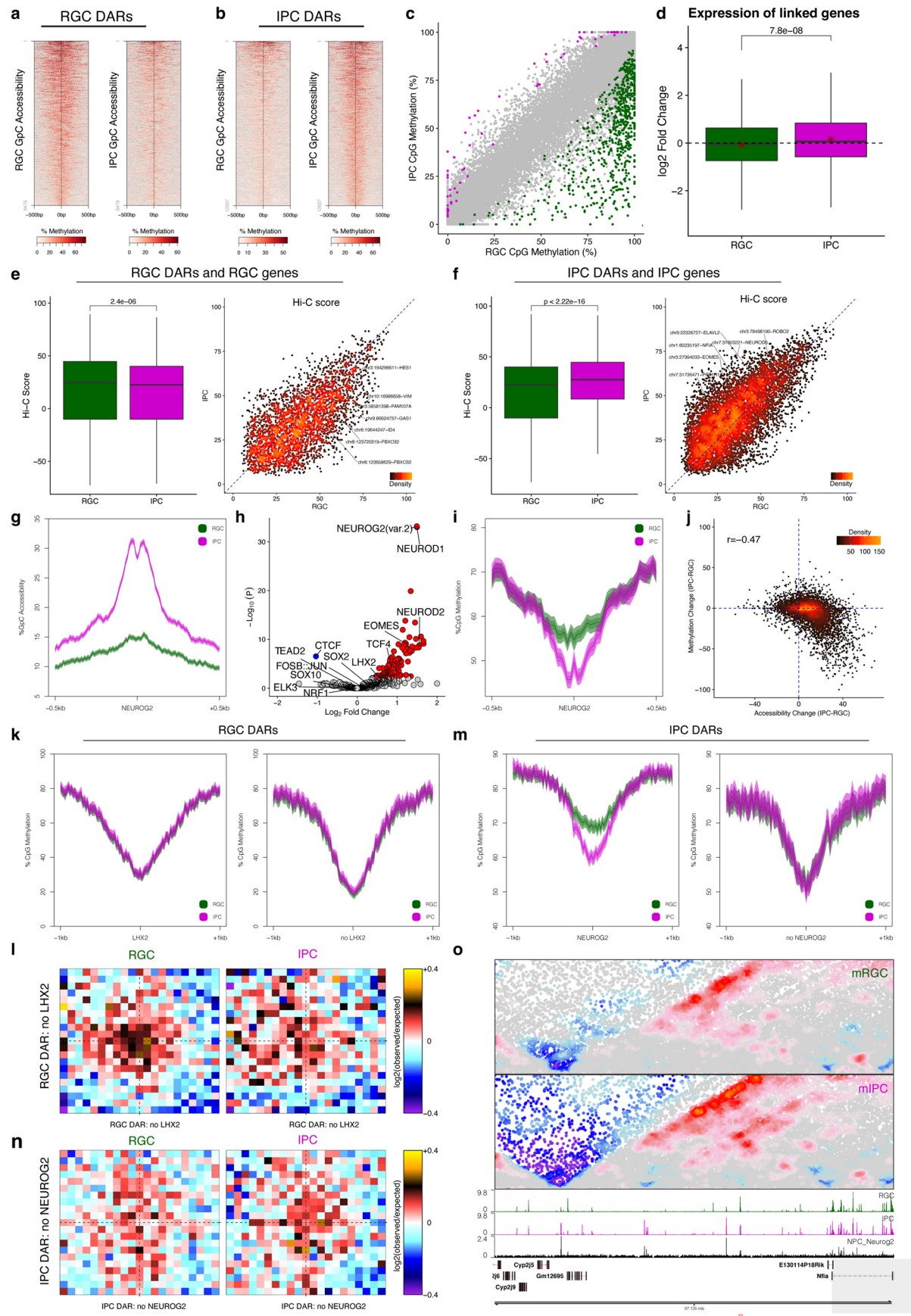

**Extended Data Fig. 5 | See next page for caption.**

**Extended Data Fig. 5 | Further characterization of epigenome dynamics in human cortical organoids. (A-B)** Heatmaps displaying accessibility levels. **(C)** Scatterplot depicting CpG methylation levels of individual GpC peaks. Peaks that significantly (FDR ≤ 0.05) lose (RGC DMR; n = 663) or gain methylation (IPC DMR; n = 66) are colored. Grey dots indicate peaks that do not change significantly CpG methylation (n = 65551). **(D)** Boxplots displaying gene expression fold changes for genes associated with either RGC or IPC DARs. Statistical significance was calculated using an unpaired two-sided Wilcoxon rank-sum test (n = 3391 and 5445 linked genes for RGC and IPC respectively). **(E)** Boxplots depicting contact strength (Hi-C score) of pairs of distal RGC DARs and their associated gene based on highest Hi-C score in RGC (n = 6358 pairs). Scatterplot colored by density represents the same individual RGC DAR – gene pairs and their Hi-C score. Statistical significance was calculated using a two-sided Wilcoxon rank-sum test. **(F)** Same as (E) but for IPC DAR and IPCs (n = 13845 pairs). **(G)** Average GpC accessibility levels of RGC and IPC at GpC peaks centered on the NEUROG2 motif (n = 10662, 10 bp bins). **(H)** Scatter plot showing the enrichment of TF motifs within NPC DMR. Red and blue dots indicate significantly (p < 0.01; abs(logFC) ≥ 0.25) enriched or depleted motifs, respectively. **(I)** Average methylation levels of RGC or IPC at GpC peaks centered on the NEUROG2 motif (n = 10662) (20 bp bin size, coverage 10x). **(J)** Scatterplot colored by density depicting changes of methylation and GpC accessibility between RGC and IPC for individual GpC peaks containing the NEUROG2 motif (n = 10662). Spearman's correlation is indicated in the left top corner. **(K)** Average CpG methylation levels of RGC or IPC across RGC DARs with (right; n = 3459) or without (left; n = 3020) LHX2 motif (20 bp bins). **(L)** Aggregated contact enrichment for RGC and IPC between RGC DARs without LHX2 motifs. **(M)** Same as (K) but for IPC DARs with (n = 9,116) and without (n = 3721) NEUROG2 motif. **(N)** Same as (L) but for IPC DARs without NEUROG2 motif. **(O)** Genomic tracks depicting the Nfia locus in the developing mouse cortex. All boxplots display the median (line), 25th and 75th percentiles (box limits), 10th and 90th percentiles (whiskers).

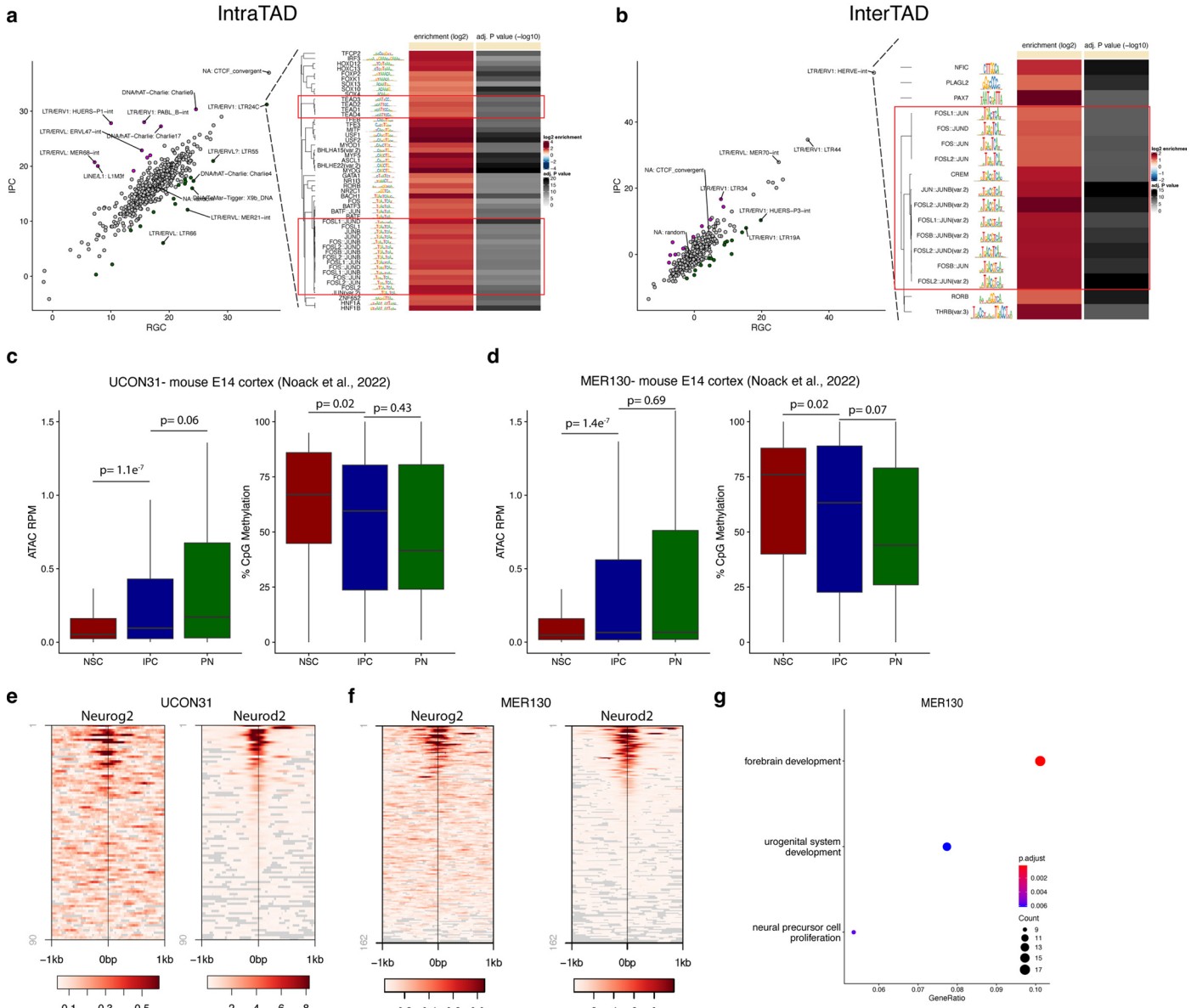

**Extended Data Fig. 6 | Epigenome dynamics at TE in human and mouse cortical development. (A)** Scatterplot depicting IntraTAD contact strength between repetitive elements of different classes. Heatmap displays motif enrichment for the LTR24C retrotransposons. **(B)** Same as (A) but with InterTAD contact strength and motif enrichment for HERV repetitive elements. **(C)** Boxplots depicting accessibility (left) and DNA methylation levels for the UCON31 TE in the mouse cortex. Statistical significance was calculated using a paired two-sided Wilcoxon rank-sum test (n = 90). **(D)** Same as (C) but for MER130

(n = 162). **(E-F)** Heatmaps displaying ChIP-seq signal of Neurog2 (left) or Neurod2 (right) at individual mouse UCON31 (E) or MER130 (F) retrotransposons. Each line represents an individual genomic region (50 bp bin size, 10x coverage). **(G)** GO term enrichment analysis of genes associated with MER130 repetitive elements. Colour and size of circles indicate adj. p-value (hypergeometric) and number of genes, respectively. All boxplots show the median (line), 25th and 75th percentiles (box limits), 10th and 90th percentiles (whiskers).

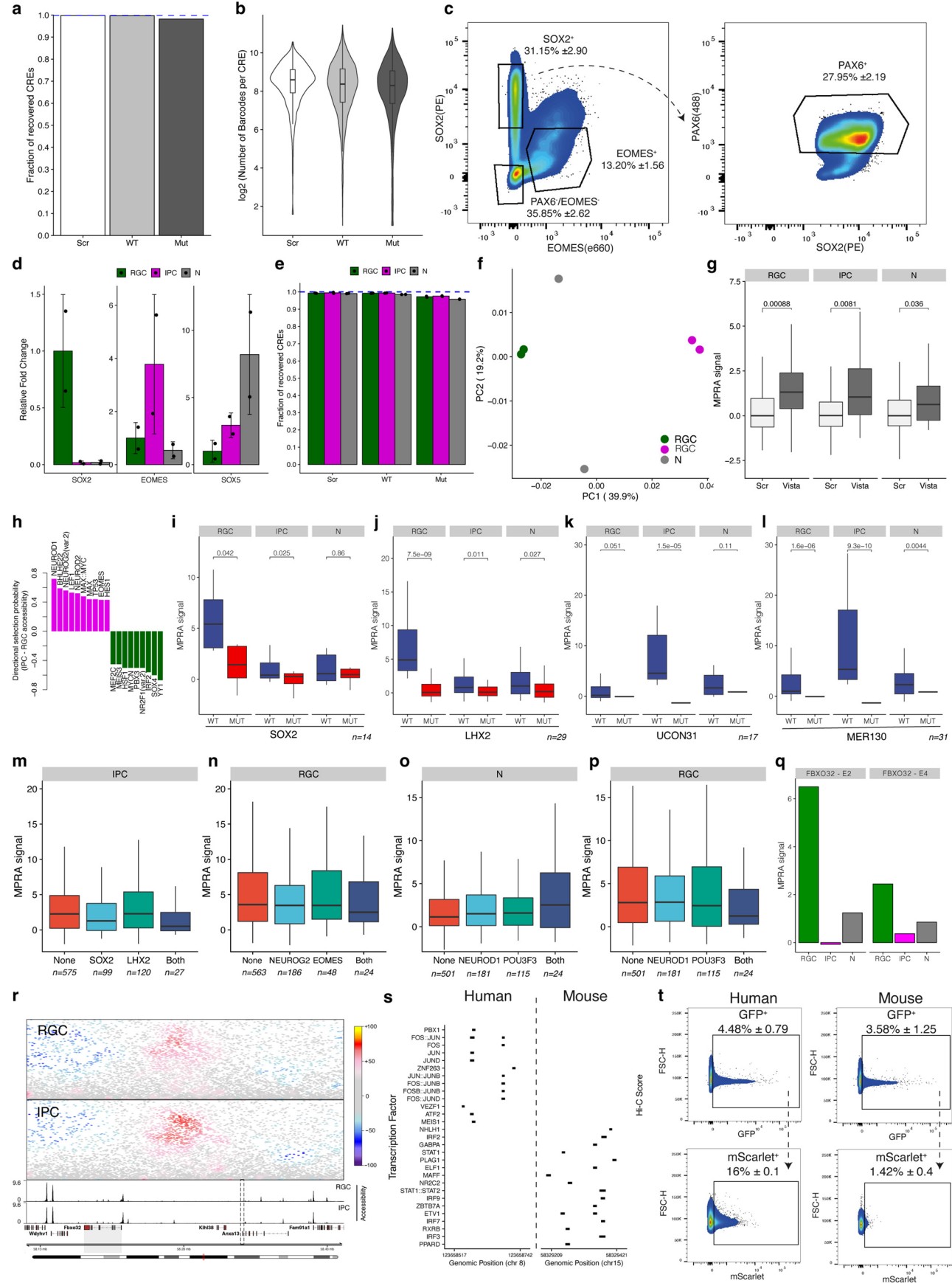

**Extended Data Fig. 7 | See next page for caption.**

**Extended Data Fig. 7 | In organoid MPRA identifies cell-type-specific enhancer activity in human neural development. (A)** Barplot displaying the fraction of recovered CREs. **(B)** Violin and boxplots displaying the number of unique barcodes (n = 499, 2995 and 2330 for Scr, WT and Mut respectively). **(C)** Gating strategy for the immunoMPRA. Number represent mean ± SD from the parental population. **(D)** Bar plots depicting cell expression levels of marker genes determined by qPCR. Dots represent individual biological replicates (n = 2). **(E)** Fraction of recovered CREs in the DNA MPRA library represented as bar graph. Dots represent individual biological replicates. **(F)** Principal component analysis performed on the ratio of sums. **(G)** Boxplots depicting MPRA signal for control regions (n = 492) and CREs overlapping with VISTA enhancers (n = 20). Statistical significance was calculated using unpaired two-sided Wilcoxon rank-sum test. **(H)** Bar plot depicting the probability of a TF motif to contribute to change in MPRA activity between IPC and RGC. **(I-J)** Boxplots depicting MPRA signal for significantly active CREs (FDR < 0.1 in RGC) containing either WT or mutated SOX2 (I) or LHX2 (J) motif. Statistical significance was calculated using a paired two-sided Wilcoxon rank-sum test. **(K-L)** Same as (I-J) but for significantly active UCON31 (K) or MER130 (L) retroelements containing a NEUROG2 motif. **(M)** Boxplots showing MPRA signal in IPC for significantly active CREs that contain SOX2 or LHX2 motifs. **(N)** Same as (M) but in RGC for NEUROG2 and EOMES containing CREs. **(O-P)** Same as (M) but for significantly active CREs for NEUROD1 and POU3F3 in N (O) or RGC (P). **(Q)** Cell-type specific MPRA signal across the E2 and E4 FBXO32 enhancers. **(R)** Contact maps and accessibility levels for RGC and IPC across the mouse Fbxo32 locus. Dotted rectangle indicates the orthologue genomic region of the human FBXO32 enhancer. **(S)** TF motifs for either the human or mouse FBXO32 enhancer sequence. **(T)** Enhancer activity of the human or mouse version of the FBXO32 – E2. Numbers represent the percentage and standard deviation (n = 3) from either singlets (top) or GFP+ (bottom) cells. Box plots in all panels display the median (line), 25th and 75th percentiles (box limits), 10th and 90th percentiles (whiskers). Source numerical data are available in source data.

# Reporting Summary

## Statistics

For all statistical analyses, confirm that the following items are present in the figure legend, table legend, main text, or Methods section.

| n/a | Confirmed | |
|---|---|---|
| ☐ | ☒ | The exact sample size (*n*) for each experimental group/condition, given as a discrete number and unit of measurement |
| ☐ | ☒ | A statement on whether measurements were taken from distinct samples or whether the same sample was measured repeatedly |
| ☐ | ☒ | The statistical test(s) used AND whether they are one- or two-sided *Only common tests should be described solely by name; describe more complex techniques in the Methods section.* |
| ☐ | ☒ | A description of all covariates tested |
| ☐ | ☒ | A description of any assumptions or corrections, such as tests of normality and adjustment for multiple comparisons |
| ☐ | ☒ | A full description of the statistical parameters including central tendency (e.g. means) or other basic estimates (e.g. regression coefficient) AND variation (e.g. standard deviation) or associated estimates of uncertainty (e.g. confidence intervals) |
| ☐ | ☒ | For null hypothesis testing, the test statistic (e.g. *F*, *t*, *r*) with confidence intervals, effect sizes, degrees of freedom and *P* value noted *Give P values as exact values whenever suitable.* |
| ☒ | ☐ | For Bayesian analysis, information on the choice of priors and Markov chain Monte Carlo settings |
| ☒ | ☐ | For hierarchical and complex designs, identification of the appropriate level for tests and full reporting of outcomes |
| ☐ | ☒ | Estimates of effect sizes (e.g. Cohen's *d*, Pearson's *r*), indicating how they were calculated |

*Our web collection on statistics for biologists contains articles on many of the points above.*

## Software and code

Policy information about availability of computer code

| Data collection | No software was used for data collection |
|---|---|
| Data analysis | imageJ (2.1.0), FlowJo (10.8.1), FACSDiva (8.0.1), MPRAflow (2.2),  MPRAanalyze (1.11), HiCRep (1.12.2), Juicebox (1.19), chromvar (1.6), JuiceMe (1.5.6), Methylkit (1.10), gNOMeHMM (0.2.1), Shaman (2.0), SeqPlots (1.22.2), modified TAURUS-MH pipeline , R, ENCODE ChIP–seq and ATAC-seq pipeline (2.3), monaLisa (0.1.5). All code as well as a list of all used packages including version numbers is available at: https://github.com/BonevLab/NoackVangelisti_2023. |

For manuscripts utilizing custom algorithms or software that are central to the research but not yet described in published literature, software must be made available to editors and reviewers. We strongly encourage code deposition in a community repository (e.g. GitHub). See the Nature Portfolio guidelines for submitting code & software for further information.

## Data

Policy information about availability of data

All manuscripts must include a data availability statement. This statement should provide the following information, where applicable:
- Accession codes, unique identifiers, or web links for publicly available datasets
- A description of any restrictions on data availability
- For clinical datasets or third party data, please ensure that the statement adheres to our policy

All raw and processed sequencing data are available in the Gene Expression Omnibus (GEO) repository: GSE211736.Previously published data that were re-analysed

here are available under accession codes: RNA-seq (GSE19608425, GSE961075), Methyl-HiC (GSE119171), Methyl-3C (GSE124391), WGBS (GSE112520), Hi-C (GSE96107), ATAC-seq (GSE113592), H3K27ac ChIP-seq (ENCSR000CGQ),  DHS  (GSE51336), MNase-seq (GSE58101), Ctcf ChIP-seq (GSE96107)5, CTCF ChIP-seq (GSE116825), Smc1 ChIP-seq (GSE22557), Neurog2 ChIP-seq (GSE63621), Neurod2 ChIP-seq (GSE67539), Micro-C (GSE130275), Nrf1 ChIP-seq (GSE67867).

## Human research participants

Policy information about studies involving human research participants and Sex and Gender in Research.

| | |
|---|---|
| Reporting on sex and gender | N/A |
| Population characteristics | N/A |
| Recruitment | N/A |
| Ethics oversight | N/A |

Note that full information on the approval of the study protocol must also be provided in the manuscript.

# Field-specific reporting

Please select the one below that is the best fit for your research. If you are not sure, read the appropriate sections before making your selection.

☒ Life sciences        ☐ Behavioural & social sciences        ☐ Ecological, evolutionary & environmental sciences

For a reference copy of the document with all sections, see nature.com/documents/nr-reporting-summary-flat.pdf

# Life sciences study design

All studies must disclose on these points even when the disclosure is negative.

| | |
|---|---|
| Sample size | Sample sizes for all data types are provided in the Supplementary Data Table 2. Sample sizes for 3DRAM-seq and MPRA were chosen based upon the ability to get representative data described based upon analogous studies in the field and to ensure replication of the results with affordable cost. |
| Data exclusions | No data was excluded from the analysis. |
| Replication | 3DRAM-seq in mESCs was performed in three biological replicates. 3DRAM-seq in cortical organoids was performed in biological duplicates. Cell-type specific MPRA in organoids was performed in two biological replicates. All attempts of replications were successful. |
| Randomization | For the 3DRAM-seq and MPRA, there was no randomization performed as they do not involve multiple study groups. For the mouse vs human FBXO32 enhancer electroporation experiments, organoids were assigned to the mouse (mE2-Fbxo32) or human (E2-FBXO32) group based on the construct used during the procedure. |
| Blinding | The investigators were not blinded to the group as no human subjects were involved and no subjective measurements were taken. |

# Reporting for specific materials, systems and methods

We require information from authors about some types of materials, experimental systems and methods used in many studies. Here, indicate whether each material, system or method listed is relevant to your study. If you are not sure if a list item applies to your research, read the appropriate section before selecting a response.

## Materials & experimental systems

| n/a | Involved in the study |
|---|---|
| ☐ | ☒ Antibodies |
| ☐ | ☒ Eukaryotic cell lines |
| ☒ | ☐ Palaeontology and archaeology |
| ☒ | ☐ Animals and other organisms |
| ☒ | ☐ Clinical data |
| ☒ | ☐ Dual use research of concern |

## Methods

| n/a | Involved in the study |
|---|---|
| ☒ | ☐ ChIP-seq |
| ☐ | ☒ Flow cytometry |
| ☒ | ☐ MRI-based neuroimaging |

# Antibodies

| | |
|---|---|
| Antibodies used | For flow-cytometry following antibodies were used: SOX2-PE (BD Biosciences, Cat. N.: 562195, clone O30-678, dilution 1:20), PAX6-AlexaFluor488 (BD Biosciences, Cat. N.: 561664, clone O18-1330, dilution 1:40) and EOMES-eFluor660 (ThermoFisher, Cat. N.: 50-4877-41, clone WD1928, dilution 1:20). For immunohistochemistry the following antibodies were used: anti-PAX6 (Biolegend, Cat. N.: 901301, clone Poly19013, dilution 1:100), anti-EOMES (R&D Systems, Cat. N.: AF6166, Polyclonal Sheep IgG, dilution 1:150), donkey anti-rabbit-A555 (Thermo Scientific, Cat. N.: A32794,dilution 1:1000), donkey anti-sheep-A488(Thermo Scientific, Cat. N.: A11015, dilution 1:1000), goat anti-Chicken A488 (Thermo Scientific, Cat. N.: A11039, Polyclonal Goat IgG, dilution 1:1000), donkey anti-goat APlus 555 (Thermo Scientific, Cat. N.: A32816, dilution 1:1000), anti-RFP (Rockland, Cat. N.: 200-101-379, Polyclonal Goat IgG, dilution 1:1000), anti-GFP (Abcam, Cat. N.: ab13970, Chicken polyclonal IgY, dilution 1:1000), anti-Oct3/4 A488 (BD Biosciences, Cat. N.: 560253, clone 40/Oct-3, dilution 1:33), PE anti-Sox2 (BD Biosciences, Cat. N.: 560291, clone 245610, dilution 1:50), V450-SSEA-4 (BD Biosciences, Cat. N.: 561156 clone MC813-70, dilution 1:67), Alexa Fluor 647 anti Tra-1-60 (BD Biosciences, Cat. N.: 560122, clone TRA-1-60, dilution 1:67), SOX17 primary antibody (Abcam, Cat, N.: ab84990, clone OTI3B10, dilution 1:200), Alexa Fluor 488 goat anti-mouse IgG (ThermoFisher, Cat. N.: A11001, dilution 1:200). |
| Validation | Antibodies were validated by the respective supplier:<br>SOX2-PE (https://www.bdbiosciences.com/ko-kr/products/reagents/flow-cytometry-reagents/research-reagents/single-color-antibodies-ruo/pe-mouse-anti-sox2.562195)<br>PAX6-AlexaFluor488: (https://www.bdbiosciences.com/en-nz/products/reagents/flow-cytometry-reagents/research-reagents/single-color-antibodies-ruo/alexa-fluor-488-mouse-anti-human-pax-6.561664)<br>EOMES-eFluor660:  (https://www.thermofisher.com/antibody/product/EOMES-Antibody-clone-WD1928-Monoclonal/50-4877-42)<br>anti-PAX6 (https://www.biolegend.com/fr-fr/products/purified-anti-pax-6-antibody-11511)<br>anti-EOMES (https://www.rndsystems.com/products/human-eomes-antibody_af6166)<br>Additionally, RNA-seq (Figure 4C) or qPCR (Extended Date Figure 7D) of FAC-sorted cell populations was performed. |

# Eukaryotic cell lines

Policy information about cell lines and Sex and Gender in Research

| | |
|---|---|
| Cell line source(s) | mESC E14TG2a was obtained from ATCC (Cat. N.: CRL-1821). The hiPS cells human cortical organoid generation were kindly provided by the Helmholtz Zentrum München iPSC core facility (HMGU1 hiPSC line). For electroporation of human organoids the hIPS cell line CRTDi004-A was used. |
| Authentication | E14TG2a mESC or HMGU1 hIPS line were authenticated by ATCC (https://www.atcc.org/products/crl-1821) or the Helmholtz Zentrum München iPSC core facility (https://www.helmholtz-muenchen.de/fileadmin/IPSC/PDF/HMGU1_datasheet_AP_ER_18052018.pdf), respectively. CRTDi004-A were authenticated by the CRTD Stem Cell Engineering Facility at Technische Universität Dresden as described in the manuscript (https://hpscreg.eu/cell-line/CRTDi004-A). |
| Mycoplasma contamination | All cell lines were tested negative for mycoplasma contamination. |
| Commonly misidentified lines<br>(See ICLAC register) | No misidentified lines were used. |

# Flow Cytometry

## Plots

Confirm that:

☒ The axis labels state the marker and fluorochrome used (e.g. CD4-FITC).

☒ The axis scales are clearly visible. Include numbers along axes only for bottom left plot of group (a 'group' is an analysis of identical markers).

☒ All plots are contour plots with outliers or pseudocolor plots.

☒ A numerical value for number of cells or percentage (with statistics) is provided.

## Methodology

| | |
|---|---|
| Sample preparation | Organoids were  dissociated using a papain-based neural dissociation kit (Miltenyi Biotec, Cat. N: 130-092-628) according to the manufacturer protocol with minor modifications. Dissociated cells were fixed in 1% Formaldehyde, quenched with 0.2M Glycine followed by permeabilization using 0.1% Saponin (Sigma-Aldrich, Cat. N: SAE0073). Cells were stained for 1h at 4C for SOX2-PE (1:20; BD Biosciences, Cat. N.: 562195), PAX6-AlexaFluor488 (1:40; BD Biosciences, Cat. N.: 561664) and EOMES-eFluor660 (ThermoFisher, Cat. N.: 50-4877-41) in staining buffer containing 0.1 % Saponin, 0.5x cOmplete, EDTA-free Protease Inhibitor Cocktail (Roche, Cat.N.: 11873580001) and 1:25 RNAse Inhibitor (Promega, Cat. N.: N261A). Stained cells were washed 4 times including one wash with washing buffer containing DAPI (1:1000; ThermoFisher, Cat. N: 62248). Cells were passed through a 40μM cell strainer and immediately FAC-sorted. |
| Instrument | Cell sorting was carried out on a FACSAria Fusion (BD Biosciences; laser: 405nm, 488nm, 561nm, 640nm) or a FACSAria III (BD Biosciences; laser: 405nm, 488nm, 561nm, 633nm) using a 100μm nozzle. |
| Software | BD FACSDiva |

Cell population abundance

Abundance of relevant cell populations are shown in Extended Data Figure 4A and 7C. Purity of FAC-sorted cells were determined by RNA-seq (Figure 4C) or qPCR of relevant marker genes (Extended Date Figure 7D).

Gating strategy

Singlets were selected using forward and side scatter followed by the identification of cells in G0/G1 by genomic content based on DAPI staining (only for 3DRAM-seq). These cells where further divided into SOX2-/EOMES+ for hIPC and SOX2+/PAX6+/EOMES- for hRGC as well as triple negative cells (only MPRA).

☒ Tick this box to confirm that a figure exemplifying the gating strategy is provided in the Supplementary Information.

