## [Peer review file · Nature Cell Biology]

Peer Review Information

Journal: Nature Cell Biology

Manuscript Title: Joint epigenome profiling reveals cell-type-specific gene regulatory programs in human cortical organoids

Corresponding author name(s): Dr Boyan Bonev

Editorial Notes:

Reviewer Comments & Decisions:

Decision Letter, initial version:

Dear Dr Bonev,

Your manuscript "Joint epigenome profiling reveals cell-type-specific gene regulatory programs in human cortical organoids", has now been seen by 3 referees, who are experts in 3D genome, chromatin, molecular/computational methods (referee 1); DNA methylation, chromatin accessibility, multiomic approaches, neurobiology (referee 2); and DNA methylation, enhancers, genomic technologies/sequencing, brain organoids (referee 3), and whose comments are pasted below. In light of their advice, we regret that we cannot offer to publish the study in Nature Cell Biology.

As you will see, although reviewer #2 offers a more positive feedback, reviewers #1 and #3 raise

serious concerns that question, among other things, the conceptual advance and impact of the method presented in the manuscript.

We are very sorry that we could not be more positive on this occasion, but we thank you for the opportunity to consider this work. Please, let me know if you would like me to consult with my colleagues at Nature Communications and ask them if they would be interested in taking the manuscript forward with the current review reports.

With kind regards,
Stelios

Stylios Lefkopoulos, PhD
He/him/his
Associate Editor
Nature Cell Biology
Springer Nature
Heidelberger Platz 3, 14197 Berlin, Germany

E-mail: stylios.lefkopoulos@springernature.com
Twitter: @s_lefkopoulos

Reviewers' comments:

Reviewer #1 (Remarks to the Author):

Noack, Vangelisti et al. present a manuscript describing a novel method, 3DRAM-seq, and its application in human cortical organoids. The method is a multi-omic approach that aims to profile 3D genome structure, DNA methylation, chromatin accessibility, and RNA expression simultaneously in the same sample. Previous studies have included versions of such multi-omic approaches but never capturing these 4 modalities together. In this sense, this would be an achievement. They demonstrate this method in mESCs and then apply it in human cortical organoids, focusing on radial glia and IPC cells. They also apply MPRAs in this system as well. While I like the general idea of such multi-omic methods, the quality of data obtained appears to be considerably worse than using gold standard measures, particularly for chromatin accessibility and RNA-seq. Further, it is unclear what utility the multi-omic method they have developed achieves beyond performing these assays in parallel, as the features they now include (ATAC-seq and RNA) relative to existing methods (Methyl-HiC) do not require extensive numbers of cells or sequencing. They do identify some interesting observations when the assays are applied in organoids, such as the changes in accessibility in UCON31 and MER130 or the potential synergy between TFs in the MPRAs, but these remain under explored. As a result, I remain unconvinced of the methods they present in this study and the new information revealed by this study in cortical organoids. I have divided my comments below into major and minor points.

Major points:

It is unclear, other than modest cost savings, what is the value of performing these assays simultaneously as opposed to in parallel. I think this is particularly true of the open chromatin aspect, as this requires fairly small numbers of cells (<50k) compared to typical protocols for bulk Hi-C or bisulfite sequencing experiments. The authors could demonstrate this by possibly performing such assays where the sample numbers are extremely limited, but instead used cortical organoids which is a system they can generate abundant numbers of cells in culture.

I am not convinced of the utility of the method from the standpoint of RNA-seq or open chromatin. From extended data Fig. 2a and b, they see similar correlations between 3DRAM-seq and RNA-seq in mESCs (0.91) as they do in RNA-seq experiments between mESCs and NPCs (0.88) and NPCs and CN (0.9). It is hard for me to accept that this is producing high-quality RNA-seq profiles that are consistent with gold standards. For ATAC-seq and DNaseI, the correlations between 3DRAM-seq and ATAC/DNase are much worse (0.66) in the same cell type than the same comparison between ATAC and DNase (0.9).

One could make the argument that the diminished correlation in open chromatin signal between 3DRAM-seq and ATAC/DNase is that 3DRAM-seq is picking up on slightly different signals. In this case, I think it is worthwhile to compare in addition with NOME-seq experiments in addition.

The authors state that they have developed a novel variant of MPRA, but it is unclear to me how this method is different than previously published MPRA other than the method of delivery of the MPRA pool. It seems more appropriate to say that they have applied MPRA in organoids.

Minor points:

Fig. 1B and C should show a negative control where the sample has not been treated with the GpC methyltransferase.

For extended data fig. 1B, are the authors using all GpC's in the genome or just those over open chromatin sites? If they are using all GpCs, having 50% GpC methylation seems like it has a lot of non-specific methylation occurring. They should make this plot showing the saturation of methylation over known open chromatin regions and outside of known open chromatin regions to give the reader an idea of the specificity of this approach. This is an important issue as high levels of non-specific GpC labelling may alter the "normal" CpG methylation profiles and make the method less useful.

Reviewer #2 (Remarks to the Author):

Multiple epigenetic mechanisms such as histone marks, DNA methylation, and chromatin accessibility may simultaneously regulate gene expression and TF binding. Current multi-omic methods are yet to simultaneously profile all three epigenetic layers (3D genome, chromatin accessibility and DNA methylation). Authors developed a method called 3DRAM-seq that will simultaneously measure (3D genome, RNA, Accessibility and Methylation sequencing) at a single cell resolution and describe the protocol by which performing these modalities are possible. Authors used three biological replicates in mouse ESCs to perform and developed the 3DRAM-seq method. The replicates unanimously revealed lower DNA methylation levels and higher accessibility for transcription start sites compared to repressed genes which is inline with previous studies. The protocol also showed high reproducibility for gene expression. They looked at Sox2 locus for confirming the protocol can capture all three epigenetic modalities. The authors benchmarked the protocols to other multiomic approaches including Methyl-3C, Methyl-HiC. Reportedly, 3DRAM-seq measurement of DNA-methylation was highly correlated with the other methods. Moreover, 3DRAM-seq had the highest proportion of uniquely mapped reads and total Hi-C contacts.

Next, the authors coupled 3DRAM-seq with immunoFACS and purified glia cells (RGC) and intermediate progenitor cells (IPC) from human cortical organoids. Authors reported upregulation of genes involved in neuronal differentiation as opposed to downregulation of RGC specific genes which is expected. Radial glial cells specialize in the developing nervous system in vertebrates and intermediate progenitor cells are transient amplifying cells in the developing cerebral cortex.

They report DAR (differentially accessible regions) in each organoid, and that there is an uncoupling between DNA methylation and 3D DNA looping for accessibility that may be dependent on TF binding regulation. Authors translate this phenomena as the fact that high methylation levels does not always result in low accessibility and vice versa.

Next, authors investigated the roles of two different classes of transposable elements (TE); MER130 and UCON31 that are associated with epigenetic differences across modalities. Despite an increase in accessibility, only MER130 showed a decrease in DNA methylation which further supports their claim of uncoupling between the two modalities.

Authors developed a novel variant of MPRA by coupling electroporation in organoids with FACS to investigate cell-type-specific regulation. Then they performed 3DRAM-seq in each cell type to investigate the role of specific TF in regulating enhancers in human neurogenesis. The authors found that enhancer activity is correlated with accessibility, but not all accessible regions drive gene expression.

Major comments:

1. Authors develop the protocol on mouse embryonic stem cells

- Although they perform the protocol in human cortical organoids later to detect cell type specific TF binding, can authors mention if the protocol takes into account the regulatory differences between mouse stem cells vs. human brain organoids

2. Authors provide sox2 locus for confirming the protocol can capture all three epigenetic modalities as well as gene expression levels, emphasizing sox2 exhibits short distances at enhancer-promoter loops: "the resolution provided by 3DRAM-seq allowed to observe all three epigenetic modalities as well as gene expression levels even at very short distances at enhancer-promoter loops, such as the Sox2 locus"

- It would be helpful for the authors elaborate on how enhancer-promoter proximity and distance matters in capturing epigenetic status as well as gene expression levels

- What is the minimum and maximum distance between enhancer and promoter over which 3DRAM-seq can successfully capture cell-type regulatory landscape?

3. Combining 3DRAM-seq with immunoFACS and RNA-Seq in RGC and IPC to look at epigenome landscape

- Authors reasoned that to profile specific cell types in complex systems such as brain development, 3DRAM-seq can be coupled with immunoFAC-sorting of RGCs and IPCs from human cortical organoids
- Would be interesting to see if the protocol is also able to identify regulatory TFs associated with mature neuronal cell types (e.g. Purkinje cells) in addition to stem cells involved in a developing brain.

4. Authors compare the performance of 3DRAM-seq coupled with ImmunoFACS purified RGC and IPC from human cortical organoids to their earlier results in mouse ESC, where replicates demonstrated a high bisulfite conversion rate, uniformly high coverage, and a high pairwise correlation across modalities; "Reminiscent of the previous results in mESC, all replicates showed a high context independent bisulfite conversion rate,"

- Can authors reason that this result is mainly due to the biological reasons and their protocol is not over-reporting the conversion efficiency and coverage? Can they provide an example of a gene where the coverage was low and that was consistent with previously established protocols such as Methyl-HiC?

- Also, often when comparing 3DRAM-seq results in human cortical organoids, authors refer back to findings in mouse ESCs. Can authors comment on how comparable the gene regulatory network is between human cortical organoids and mouse ESCs?

5. Why did the authors choose UCON31 and MER130 as two specific classes of TE to study chromatin accessibility?

6. Authors performed MPRA to show the utility of their protocol works not only in bulk nuclei but can also be extended to cell-type specific, can they perform a single-cell experiment (rna-seq, methylation, etc) to profile cell-type-specific enhancer activity?

7. Some figures are missing scale bar, eg. x axis y axis annotation (1.6 in the top right corner in fig 1E is hard to see except mention in legend)

8. How was the protocol optimized for fixed cells? It is often difficult to separate clusters on FACS sort. The paper acknowledged their success as usually difficult to achieve but changes were not apparent. Protocol says cells can go directly into 3DRAM-seq before FACS sorting. It seems it would be beneficial to sort before to enrich for specific cell types for such a high resolution multi-omic approach. Is this an option or was it tried during adaptation of the protocol?

Reviewer #3 (Remarks to the Author):

This manuscript applied robust epigenomic/3D-genome profiling approaches and functional validation methods previously described in Noack et al., 2022 (from the same group) Nature Neuroscience to human brain organoids. Conceptually and also methodologically, this manuscript is quite similar to the published Noack et al., 2022 paper. The addition of M.CviPI-based chromatin accessibility profiling to Methyl-HiC is a relatively modest improvement that has been demonstrated multiple times by various groups. Although cerebral organoids have been shown as a sound model for brain development, there have also been compelling arguments that organoids fall short in modeling the specification of cell subtypes (for example, Bhaduri et al., 2020, PMID: 31996853). So I would view the current manuscript as a bit less significant than Noack et al., 2022, especially from a data resource perspective), since Noack et al, 2022 profiled primary mouse brain tissues, whereas the present study only included data generated from organoids. Therefore although I think the presented works are solid, the manuscript might be a better fit for a more specialized journal.

More specific comments

1. Technically the RNA profile was not generated from the same nuclei sample as those used by the Methyl-HiC assay. It might be helpful to clarify that in Fig. 1A schematics.

2. Could the authors further clarify the low overlapping between DMRs and DARs (1% for decreased DNA methylation, or 0.1% for increased DNA methylation). The degree of overlap seems to be much lower than what the meta-analysis in Fig. 4B suggests.

3. The observation that LHX2-associated binding does not associate with DNA demethylation could be explained by that these regions are already lowly-methylation (<25%) to start with, whereas regions bound by Neurog2 show a methylation level of 50-60%. Therefore it is unclear whether the result can be generalized to an uncoupling of methylation and other data modalities. The genomic contexts (promoters vs. enhancers, GpC contents, etc) of Lhx2 and Neurog2 binding sites should be further dissected.

4. How many copies of MER130 and UCON31 TEs were found in the human genome? This will help to distinguish whether the p-value difference shown in Fig. 5C-D (significant for MER130, not significant

for UCON31) was driven by a difference in the effect size or sample size.

**Although we cannot publish your paper, it may be appropriate for another journal in the Nature Portfolio. If you wish to explore the journals and transfer your manuscript please use our manuscript transfer portal. You will not have to re-supply manuscript metadata and files, but please note that this link can only be used once and remains active until used. For more information, please see our manuscript transfer FAQ page.

Note that any decision to opt in to In Review at the original journal is not sent to the receiving journal on transfer. You can opt in to In Review at receiving journals that support this service by choosing to modify your manuscript on transfer. In Review is available for primary research manuscript types only.

**For Nature Portfolio general information and news for authors, see <http://npg.nature.com/authors>.

Author Rebuttal to Initial comments

We would like to thank the reviewers for considering our manuscript and providing constructive and thoughtful feedback. We have now thoroughly revised the manuscript and have included completely new experimental data and computational analysis. We believe that these additions significantly expand the 3DRAM-seq toolbox to include paired co-accessibility measurements at single molecule resolution and lead to novel conceptual advances in how enhancer evolution could have contributed to human specific gene expression in the developing cortex. Below we summarize the main changes in the revision, followed by a point-by-point response to each of the issues raised by the reviewers.

1. We have developed a new computational pipeline which enables co-accessibility/co-methylation measurement across pairs of regions separated by large distances with single molecule resolution. This approach leverages the property of the proximity ligation which joins together genomic regions located proximally in the 3D genome space but potentially spanning hundreds or thousands of bps in the linear genome. We then couple this property with co-accessibility or co-methylation measurements across the two reads from the same read pair, essentially enabling single-molecule resolution. We employ this novel approach to measure co-accessibility at convergent Ctf binding motifs and enhancer-promoter pairs for the first time. Importantly, we observe no evidence for synergistic co-accessibility at Ctf sites, a novel finding which is consistent with the predictions from the loop extrusion model and with recent microscopy findings about Ctf residence time and the duration of chromatin loops. These results are presented in a completely new main and supplementary figure (Fig. 3 and Extended Data Fig. 3) and highlight the unique ability of the 3DRAM-seq method to leverage the simultaneous mapping of multiple epigenetic modalities. Furthermore, we have also employed this approach to interrogate the synergistic effects between TF associated with cell-type specific changes in chromatin accessibility in our organoid system, confirming our results from the MPRA.
2. We have identified a novel set of enhancers for the radial glia marker gene *FBXO32*, which is highly expressed in the developing human but not mouse cortex. We show that these putative enhancers are active, accessible and form chromatin loops with the *FBXO32* promoter specifically in RGC but not in IPC. Furthermore, we show that they are not present in the mouse, despite overall high levels of synteny at the *FBXO32* locus. Finally, we have added new experiments to directly test the activity of one of the predicted human enhancers and its mouse orthologous sequence in human cortical organoids. Consistent with our predictions, only the human but not the mouse sequence is able to drive gene expression. These novel results highlight the ability of 3DRAM-seq coupled with in organoid MPRA to uncover new regulatory elements associated with brain evolution.
3. We have developed an interactive platform for visualizing and exploring all the data generated by 3DRAM-seq in this manuscript: shiny.bonevlab.com/ram. This enable people interested in gene regulatory networks associated with human brain development to browse the data and visualize the different epigenetic modalities uncovered by 3DRAM-seq at their desired loci.

Reviewers' comments:

Reviewer #1 (Remarks to the Author):

Noack, Vangelisti et al. present a manuscript describing a novel method, 3DRAM-seq, and its application in human cortical organoids. The method is a multi-omic approach that aims to profile 3D genome structure, DNA methylation, chromatin accessibility, and RNA expression simultaneously in the same sample. Previous studies have included versions of such multi-omic approaches but never capturing these 4 modalities together. In this sense, this would be an achievement. They demonstrate this method in mESCs and then apply it in human cortical organoids, focusing on radial glia and IPC cells. They also apply MPRA in this system as well. While I like the general idea of such multi-omic methods, the quality of data obtained appears to be considerably worse than using gold standard measures, particularly for chromatin accessibility and RNA-seq.

Further, it is unclear what utility the multi-omic method they have developed achieves beyond performing these assays in parallel, as the features they now include (ATAC-seq and RNA) relative to existing methods (Methyl-HiC) do not require extensive numbers of cells or sequencing.

They do identify some interesting observations when the assays are applied in organoids, such as the changes in accessibility in UCON31 and MER130 or the potential synergy between TFs in the MPRA, but these remain under explored.

As a result, I remain unconvinced of the methods they present in this study and the new information revealed by this study in cortical organoids. I have divided my comments below into major and minor points.

We would like to thank the reviewer for acknowledging the ability of 3DRAM-seq to simultaneously capture multiple different epigenetic modalities for the first time. In the revised manuscript, we have performed additional quality controls and emphasized the conceptual advances through new experimental data and computational analysis. In particular, we would like to highlight three particular advances:

- We have now performed additional detailed comparisons with gold standard methods for chromatin accessibility and gene expression such as ATAC-seq and RNA-seq. This analysis confirms the high quality of the 3DRAM-seq data and highlights some of the advantages of GpC-based accessibility compared to conventional ATAC-seq such as single-nucleotide resolution and motif footprinting. In addition, a major advantage for 3DRAM-seq compared to conventional RNA/ATAC is its compatibility with formaldehyde fixation, enabling coupling with intracellular immunoFACS to study specific cell types in a complex tissue. At the same time, we do acknowledge that (sc)ATAC-seq is still the method of choice, if one is interested in chromatin accessibility alone, due to its ease of use and lower sequencing requirements.
- We have also developed a new computational pipeline which leverages 3DRAM-seq to enable paired co-accessibility measurements with single-molecule resolution as detailed below. Importantly, this approach is only possible using 3DRAM-seq and cannot be inferred using other methods such as ATAC-seq + Methyl-HiC. Applying this novel method to convergently-oriented Ctf motifs at chromatin loop anchors suggests that Ctf binding is dynamic and occurs independently, consistent with loop extrusion and recent findings based on microscopy (Gabriele et al., Science 2022; Mach et al., Nat Genet 2022).
- We have also expanded our observations on TF-based synergy in the MPRA to single-molecule co-accessibility, which confirms our previous results. Finally, we leverage our 3DRAM-seq and MPRA data in human organoids to identify a novel, human-specific enhancer element for the ventricular radial glia marker gene *FBXO32*, which we validate experimentally using in organoid electroporation.

Major points:

It is unclear, other than modest cost savings, what is the value of performing these assays simultaneously as opposed to in parallel. I think this is particularly true of the open chromatin aspect, as this requires fairly small numbers of cells (<50k) compared to typical protocols for bulk Hi-C or bisulfite sequencing experiments. The authors could demonstrate this by possibly performing such assays where the sample numbers are extremely limited, but instead used cortical organoids which is a system they can generate abundant numbers of cells in culture.

We thank the reviewer for this opportunity to better highlight what are the advantages of performing 3DRAM-seq instead of parallel measurements: 1) compatibility with formaldehyde fixation (discussed above); 2) single-nucleotide resolution for more precise TF motif footprinting and 3) the newly added ability to measure co-accessibility and co-methylation at single-molecule resolution. In addition, the M.CviPI-based in vitro methylation assay for chromatin accessibility used in 3DRAM-seq has been shown to have several advantages over Tn5-based ATAC-seq, such as the lack of a

characteristic sequence bias (Nordström et al., 2019; Zhang et al., NAR Genomics 2021), its ability to resolve better nucleosome occupancy (Nordström et al., 2019; Figure 2B-C and S2G) and single molecule TF occupancy measurements (Sönmezer et al., Molecular Cell 2021). However, we would like to clarify that 3DRAM-seq is not meant as a replacement for individual assays where only a single epigenetic layer is of interest (which we have explicitly now added to the discussion - L437-439).

In particular, we would like to highlight here the new computational analysis we have now developed, which is unique to 3DRAM-seq and allows for the first time co-accessibility/methylation measurements over long distances at single molecule resolution. Specifically, this approach allows us to measure the epigenetic state at pair of genomic elements (such as TF binding sites or enhancer-promoters) which can be separated by large genomic distance, as long as they are in close physical proximity and can be cross linked together. Previously, similar approaches were limited to single-reads (i.e. <150nt) or had very limited throughput using long-read sequencing (Shipony et al., Nat Methods 2020; Abdulhay et al., eLife 2020). We employ this novel approach to measure co-accessibility at convergent Ctf binding motifs and enhancer-promoter pairs for the first time. Importantly, we observe no evidence for synergistic co-accessibility at Ctf sites, a novel finding which is consistent with the predictions from the loop extrusion model and with recent microscopy findings about Ctf residence time and the duration of chromatin loops. These results are presented in a completely new main and supplementary figure (Fig. 3 and Extended Data Fig. 3) and highlight the unique ability of the 3DRAM-seq method to leverage the simultaneous mapping of multiple epigenetic modalities. Furthermore, we have also employed this approach to interrogate the synergistic effects between TF associated with cell-type specific changes in chromatin accessibility in our organoid system (Fig 5J-K), generating further support for our MPRA-based findings.

I am not convinced of the utility of the method from the standpoint of RNA-seq or open chromatin. From extended data Fig. 2a and b, they see similar correlations between 3DRAM-seq and RNA-seq in mESCs (0.91) as they do in RNA-seq experiments between mESCs and NPCs (0.88) and NPCs and CN (0.9). It is hard for me to accept that this is producing high-quality RNA-seq profiles that are consistent with gold standards.

We thank the reviewer for raising this important point. We have now performed additional analysis to directly compare the quality of our RNA data to stand-alone gold standard methods. First, we want to clarify that the ES/NPC/CN RNA-seq data we have previously used (GSE96107) is based on a different library preparation method compared to our 3DRAM-seq data, namely, stranded total RNA (for GSE96107) vs unstranded polyA enrichment (for 3DRAM-seq). This difference in library preparation likely explains the higher correlations across cell types observed previously. To directly test this, we have now included additional comparisons with a recently published mES RNA-seq polyA-based dataset (Galle et al., 2022; GSE196084). We now show that the correlation between 3DRAM-seq and this new mES dataset is significantly higher compared to other cell types as expected (Extended Data Figure 2A; 0.94 for mES vs 0.83/0.73 for NPC/CN). Furthermore, we have performed additional comparisons between the different mES RNA-seq datasets showing that the genome-wide read distribution across genome features such exons and introns (Extended Data Figure 2B) as well as at individual genomic loci (Extended Data Figure 2C) is comparable, confirming the high quality of the RNA.

For ATAC-seq and DNaseI, the correlations between 3DRAM-seq and ATAC/DNase are much worse (0.66) in the same cell type than the same comparison between ATAC and DNase (0.9). One could make the argument that the diminished correlation in open chromatin signal between 3DRAM-seq and ATAC/DNase is that 3DRAM-seq is picking up on slightly different signals.

We agree with the reviewer that the M.CviPI-based methylation assay for chromatin accessibility that is used in 3DRAM-seq picks up slightly different signals than ATAC-seq and DNA-seq. In fact, these differences (including between ATAC-seq and DHS) have been extensively characterized previously (Nordström et al., NAR 2019), where they provide evidence that “assay specific nucleosome-depleted regions are indeed genuine open chromatin sites and contribute important information for accurate gene expression prediction”. This is indeed consistent with our own analysis of the chromatin

accessibility modality in 3DRAM-seq, where we observe substantial overlap between both peaks (Fig. 2A-B) and signal intensity (Fig. 2C). The differences in the genome-wide correlations that the reviewer rightly points out in Extended Data Figure 3D likely results from: 1) the slightly higher dynamic range of ATAC-seq and 2) the difference in resolution and aggregation across genomic bins.

We now include additional comparisons with ATAC-seq and DHS which provide further support that the chromatin accessibility aspect of 3DRAM-seq is on par with the gold-standard methods. First, we show that GpC accessibility at DHS-peaks is almost identical to the DHS signal itself (Extended Data Figure 2H) and it is also prominently present at ATAC-seq peaks, although those are typically less sharp and with fuzzier nucleosome patterns (Figure R1). Next, we show that GpC-based chromatin accessibility is highest at regions annotated as active/bivalent promoters, CTCF sites and enhancers as expected (Extended Data Figure 2E) and is comparable to both the DHS and the ATAC-seq data (Figure R2). Finally, we highlight the higher resolution of 3DRAM-seq for motif footprinting using the TF Nrfl1, which is known to bind as homodimer. This pattern is clearly visible using GpC-based methylation (Extended Data Figure 1G) as it provides a single-nucleotide resolution of accessibility, but is either fuzzy or absent using other assays (ATAC-seq and DHS respectively).

Figure R1. Heatmaps showing accessibility measured by GpC methylation, ATAC-seq, DHS and MNase across ATAC peaks

Figure R2. Accessibility levels measured using 3DRAM-seq (left), ATAC-seq (middle) and DHS (right) across different ChromHMM features (Pintacuda G et al. 2017).

In this case, I think it is worthwhile to compare in addition with NOME-seq experiments in addition.

We thank the reviewer for this comment which allowed us to further highlight the advantages of 3DRAM-seq compared to individual methods. Since the cost for generating a high-quality NOME-seq are substantial, we opted to instead use a recently published dataset (Keniry et al. 2022) from a similar mouse embryonic cell line despite a shallower sequencing depth. The accessibility pattern at CTCF bound motifs and active TSSs were highly comparable between the two datasets (Figure R3), indicating that no additional bias was introduced in 3DRAM-seq. Importantly, the sensitivity of 3DRAM-seq was higher than conventional NOME-seq, likely due to the additional fixation which stabilizes the nucleosome pattern and allows for longer treatment

Figure R3. Average accessibility levels measured by 3DRAM-seq and NOME-seq across CTCF bound loci and active TSSs.

with M.CviPI without a significant increase in background (Sönmezer et al. 2021, Oberbeckmann et al., 2019). This was not simply due to higher sequencing depth, as subsampled 3DRAM-seq sample with sequencing depth matching the NOME-seq (~67 million reads) was almost indistinguishable.

The authors state that they have developed a novel variant of MPRA, but it is unclear to me how this method is different than previously published MPRA other than the method of delivery of the MPRA pool. It seems more appropriate to say that they have applied MPRA in organoids.

We acknowledge that the reviewer is correct and adapted the manuscript accordingly (L28-29 and L469-470). Nevertheless, we would like to point out that we report for the first time the feasibility of performing a MPRA in cortical organoids, which is not trivial due to the limited number of cells that can be obtained. Furthermore, we show that despite this input limitations, MPRA can be coupled with immunoFACS to obtain cell-type specific resolution. We think this represents an important proof of principle and can be now expand to other organoid systems or tissue explants (such as the human fetal brain).

Minor points:

Fig. 1B and C should show a negative control where the sample has not been treated with the GpC methyltransferase.

We would like to point out that the original NOME-seq manuscript addressed precisely this question and found that M.CviPI treatment does not affect CpG methylation levels, except in the C^mCG context (Kelly et al., Genome Research 2012), which we also exclude computationally. Furthermore, recent publication from the Krebs lab directly addressed the concern of the reviewer by showing that treatment with the GpC methyltransferase does not influence endogenous DNA methylation levels (Kreibich et al., Mol Cell 2023). Finally, previous studies have also shown that the enzymatic steps required for measuring 3D genome architecture (e.g. DpnII digestion, A-tailing, T4 ligation) do not influence DNA methylation pattern (MethylHiC - Li et al., 2019 and Methyl-3C - Lee et al., 2019), which is also evident by the highly characteristic pattern of DNA methylation at Ctfc sites and TSSs in our 3DRAM-seq data.

For extended data fig. 1B, are the authors using all GpC's in the genome or just those over open chromatin sites? If they are using all GpCs, having 50% GpC methylation seems like it has a lot of non-specific methylation occurring. They should make this plot showing the saturation of methylation over known open chromatin regions and outside of known open chromatin regions to give the reader an idea of the specificity of this approach. This is an important issue as high levels of non-specific GpC labelling may alter the "normal" CpG methylation profiles and make the method less useful.

GpC methylation levels for Extended Data Figure 1B were calculated based on bisulfite amplicon sequencing of 5 regions covering in total 29 individual GpCs. We intentionally selected open chromatin regions based on the available ATAC and DHS data to use as positive controls for our initial pilot tests. Indeed, we find that these regions are associated with open chromatin with up to

90% GpC methylation (e.g.: chr11:98345052-98345883, Figure R4) and are not representative for the whole genome. We used these regions solely to estimate the required M.CviPI incubation times for fixed cells which is comparable to what was used previously for fixed yeast nuclei (Oberbeckmann et al., 2019). Following the suggestion of the reviewer we provide now the measured GpC methylation levels across various chromatin states define by ENCODE Chip-seq data (Extended Data Figure 2F; Pintacuda G et al. 2017). This demonstrates up to 5-fold increase in average methylation (~25% vs. ~5%) between open and closed regions which is in line or even better than previous measurements (Nordström et al., 2019).

Finally, one of our spike-in controls consists of non-methylated lambda DNA incubated in vitro with M.CviPI, which even shows GpC labelling to saturation (since the lambda DNA is not chromatinized and thus fully accessible) does not lead to unspecific CpG signal (Extended Data Figure 1C).

Reviewer #2 (Remarks to the Author):

Multiple epigenetic mechanisms such as histone marks, DNA methylation, and chromatin accessibility may simultaneously regulate gene expression and TF binding. Current multi-omic methods are yet to simultaneously profile all three epigenetic layers (3D genome, chromatin accessibility and DNA methylation). Authors developed a method called 3DRAM-seq that will simultaneously measure (3D genome, RNA, Accessibility and Methylation sequencing) at a single cell resolution and describe the protocol by which performing these modalities are possible. Authors used three biological replicates in mouse ESCs to perform and developed the 3DRAM-seq method. The replicates unanimously revealed lower DNA methylation levels and higher accessibility for transcription start sites compared to repressed genes which is inline with previous studies. The protocol also showed high reproducibility for gene expression. They looked at Sox2 locus for confirming the protocol can capture all three epigenetic modalities. The authors benchmarked the protocols to other multiomic approaches including Methyl-3C, Methyl-HiC. Reportedly, 3DRAM-seq measurement of DNA-methylation was highly correlated with the other methods. Moreover, 3DRAM-seq had the highest proportion of uniquely mapped reads and total Hi-C contacts.

Next, the authors coupled 3DRAM-seq with immunoFACS and purified glia cells (RGC) and intermediate progenitor cells (IPC) from human cortical organoids. Authors reported upregulation of genes involved in neuronal differentiation as opposed to downregulation of RGC specific genes which is expected. Radial glial cells specialize in the developing nervous system in vertebrates and intermediate progenitor cells are transient amplifying cells in the developing cerebral cortex.

They report DAR (differentially accessible regions) in each organoid, and that there is an uncoupling between DNA methylation and 3D DNA looping for accessibility that may be dependent on TF binding regulation. Authors translate this phenomena as the fact that high methylation levels does not always result in low accessibility and vice versa.

Next, authors investigated the roles of two different classes of transposable elements (TE); MER130 and UCON31 that are associated with epigenetic differences across modalities. Despite an increase in accessibility, only MER130 showed a decrease in DNA methylation which further supports their claim of uncoupling between the two modalities.

Authors developed a novel variant of MPRA by coupling electroporation in organoids with FACS to investigate cell-type-specific regulation. Then they performed 3DRAM-seq in each cell type to investigate the role of specific TF in regulating enhancers in human neurogenesis. The authors found that enhancer activity is correlated with accessibility, but not all accessible regions drive gene expression.

We thank the reviewer for the positive comments and for highlighting the major strengths of 3DRAM-seq and our manuscript.

Major comments:

1. Authors develop the protocol on mouse embryonic stem cells

- Although they perform the protocol in human cortical organoids later to detect cell type specific TF binding, can authors mention if the protocol takes into account the regulatory differences between mouse stem cells vs. human brain organoids

3DRAM-seq was developed to be broadly applicable to many cell-types and species. Indeed, the protocol is identical when applied to mouse ES or human brain organoids, i.e. it does not require cell-type or species-specific optimisation steps (in comparison to other single modality methods such as ATAC or MNase). To ensure that we can also capture regulatory differences between mouse ES and human RGC/IPC, we further examined enrichment of TF binding sites and chromatin accessibility / DNA Methylation associated with Neurog2. We observed that GpC accessible regions in mESC display a strong enrichment for Sp-like and Krüppel-like transcription factors (Figure R5) which have been shown to act together to regulate gene expression (Kaczynski et al., 2003) in order to maintain pluripotency and self-renewal potential (Yamane et al., 2018). On the other hand mESC are strongly depleted for transcription factor motifs associated with cortical radial glia (Lhx2, Hes1) or intermediate progenitor fate (Eomes, Neurog2). Similarly, both species and cell-type specific differences are seen the level of 3D genome organisation and DNA Methylation levels at the Sox2 locus (compare Figure 2G and 4I), where cell-type specificity in enhancers and chromatin loops have been previously shown (Bonev et al., Cell 2017).

Figure R5. Heatmap displaying TF motif enrichment for mouse ESC, human RGC and human IPC.

2. Authors provide sox2 locus for confirming the protocol can capture all three epigenetic modalities as well as gene expression levels, emphasizing sox2 exhibits short distances at enhancer-promoter loops: “the resolution provided by 3DRAM-seq allowed to observe all three epigenetic modalities as well as gene expression levels even at very short distances at enhancer-promoter loops, such as the Sox2 locus”

- It would be helpful for the authors elaborate on how enhancer-promoter proximity and distance matters in capturing epigenetic status as well as gene expression levels

We used the Sox2 locus because of its very well characterized enhancer (also called Sox2 superenhancer or LCR) which interacts with Sox2 promoter (Li et al, Plos One 2014 and others). This interaction has been observed with high-resolution Hi-C (Bonev et al., Cell 2017) and microscopy, but it is not as strong as chromatin loops associated with architectural proteins such as Ctf. Therefore, to benchmark an assay designed to measure 3D genome organization, it is useful to determine its ability to resolve weaker chromatin loops, such as those between enhancers and promoters.

The relationship between enhancer-promoter proximity (on the linear genome and in 3D) and how it affects gene expression is one of the major topics of the whole field of gene regulation. While this question is not the major focus of the manuscript, we point the reviewer to recent relevant publications and reviews (Schoenfelder et al., Nat Rev Genet 2019; Zuin et al., Nature 2022; Bonev et al., Cell 2017).

To address the second question, whether the epigenetic status of two loci is affected if they are in close physical proximity, we have now developed a new computational pipeline as part of our 3DRAM-seq method which, for the first time, allows co-accessibility/methylation measurements over long distances at single molecule resolution. Specifically, this approach allows us to measure the

epigenetic state at pair of genomic elements (such as TF binding sites or enhancer-promoters) which can be separated by large genomic distance, as long as they are in close physical proximity and can be cross linked together. Previously, similar approaches were limited to single-reads (i.e. <150nt) or had limited throughput using long-read sequencing (Shipony et al., Nat Methods 2020; Abdulhay et al., eLife 2020). We employ this novel approach to measure co-accessibility at convergent Ctf binding motifs and enhancer-promoter pairs for the first time. Importantly, we observe no evidence for synergistic co-accessibility at Ctf sites, a novel finding which is consistent with the predictions from the loop extrusion model and with recent microscopy findings about Ctf residence time and the duration of chromatin loops. These results are presented in a completely new main and supplementary figure (Fig. 3 and Extended Data Fig. 3) and highlight the unique ability of the 3DRAM-seq method to leverage the simultaneous mapping of multiple epigenetic modalities.

- What is the minimum and maximum distance between enhancer and promoter over which 3DRAM-seq can successfully capture cell-type regulatory landscape?

It is important to state that the chromatin accessibility and DNA methylation modality of 3DRAM-seq are entirely local, i.e. the ability to fully capture the accessibility/methylation status of a locus is independent of its 3D genome conformation. Since 3DRAM-seq is a variant of the chromatin conformation capture assays like Hi-C, its ability to resolve enhancer-promoter interactions depends on the resolution of the assay (which scales with sequencing depth), the polymer nature of chromatin (exponential decay of interaction strength as a function of the genomic distance) and the background normalization model. Our current estimate is that with sequencing depth of ~300 million uniquely mapped reads we can resolve interactions involving elements separated by at least 5Kb, with exponentially decreasing power after 5Mb.

3. Combining 3DRAM-seq with immunoFACS and RNA-Seq in RGC and IPC to look at epigenome landscape

- Authors reasoned that to profile specific cell types in complex systems such as brain development, 3DRAM-seq can be coupled with immunoFAC-sorting of RGCs and IPCs from human cortical organoids

- Would be interesting to see if the protocol is also able to identify regulatory TFs associated with mature neuronal cell types (e.g. Purkinje cells) in addition to stem cells involved in a developing brain.

We would like to point out that 3DRAM-seq is broadly applicable across cell-types and species. Therefore 3DRAM-seq could be in principle performed on selected population of neurons such as Purkinje cells and is only limited by the availability of suitable antibodies for the purification via immuno-FACS. However, as far as we aware, current protocols for brain organoids do not produce Purkinje cells, which will make a direct comparison with progenitor cells such as RGC and IPC impossible. Therefore, the only option would be to perform these experiments in mice, where we hope the reviewer agrees would be out of scope of this manuscript.

4. Authors compare the performance of 3DRAM-seq coupled with ImmunoFACS purified RGC and IPC from human cortical organoids to their earlier results in mouse ESC, where replicates demonstrated a high bisulfite conversion rate, uniformly high coverage, and a high pairwise correlation across modalities; “Reminiscent of the previous results in mESC, all replicates showed a high context independent bisulfite conversion rate, ...”

- Can authors reason that this result is mainly due to the biological reasons and their protocol is not over-reporting the conversion efficiency and coverage?

We thank the reviewer for this important comment. We used two independent ways to calculate bisulfite conversion rate. First, we calculated direct bisulfite conversion rates using spike-in controls: 1) in vitro M.CviPI treated unmethylated lambda DNA (lack endogenous DNA methylation, accessible) and fully methylated puc19 DNA (CpG/GpC methylated). Such spike-in controls report bisulfite conversion rates independently from the biological samples and are commonly used (e.g. Foox et al., 2021). The results are shown in Extended Figure 1C and they indicate very high conversion efficiency based on the lambda DNA (100% conversion efficiency will result in 0% CpG methylation, so measured conversion rate is $100 - 2.58 = 97.42\%$), as well as very high detection level of

actual methylated CpG/GpC (98.21 and 98.51 respectively). In addition to these controls, we also evaluated methylation levels genome-wide. At active promoters (which are known to be hypomethylated) our measured CpG methylation levels reach almost zero (Extended Data Figure 1F), indicating a high conversion rates of unmethylated cytosine to thymine during the bisulfite treatment. These results, coupled with the high reproducibility of the 3DRAM-seq biological replicates (Extended Data Figure 1D-E), suggest that 3DRAM-seq is able to faithfully recapitulate endogenous DNA methylation and accessibility levels.

To test for coverage in comparison to other methods we subsampled the number of reads and plotted the resulting coverage in genomic bins of 100bp (Fig 2D). This analysis shows that 3DRAM-seq has comparable coverage to Methyl-3C (which can only detect DNA methylation and 3D genome organization, but has less power to resolve chromatin loops - Fig 3E) and actually performs better than Methyl-HiC or WGBS. To allow a robust estimation of methylation levels of individual cytosines in all contexts, all results reported are filtered for a sequencing coverage of at least 5x (individual replicates mESC; S1D-E) or 10x (all other analysis), which are with range of the recommended coverage for bisulfite sequencing (Ziller et al., 2015).

Can they provide an example of a gene where the coverage was low and that was consistent with previously established protocols such as Methyl-HiC?

To compare the distribution of reads (coverage) between the different multiomic methods (3DRAM-seq, MethylHiC, Methyl3C) we first down sampled the data to the same number of unique reads (1x10⁷) and plotted the coverage density across chromosome 10 (Figure R6A). On the chromosome wide scale 3DRAM-seq and Methyl3C have a fairly even distribution of sequencing reads, while pile-up of sequencing reads are observed in MethylHiC, indicating potential amplification biases. Zooming closer into a smaller region (chr10:22156141-22385360) we observed comparable sequencing coverages between the datasets indicating that no biases are introduced by 3DRAM-seq (Figure R6B).

Figure R6: Comparison of genomic coverage archived by 3DRAM-seq, Methyl3C and MethylHiC across chr 10 (10kb bins; A) and a randomly chosen smaller region (chr10:22156141-22385360; 100bp; B).

- Also, often when comparing 3DRAM-seq results in human cortical organoids, authors refer back to findings in mouse ESCs. Can authors comment on how comparable the gene regulatory network is between human cortical organoids and mouse ESCs?

We thank the reviewer for this comment but would like to mention the main purpose of using mESC as a starting point was to establish 3DRAM-seq in a biological system, where there are many published datasets available for comparison. Therefore, we draw mainly technical comparison (conversion rates, coverage etc.) between the mESC and human datasets to show that 3DRAM-seq can be performed across species and cell-types. Nevertheless, as already discussed above (Figure R5) a biological comparison between the cell types is possible and display distinct cell-type specific features. We would also like to mention that in the original version of the manuscript we perform several comparisons to our recently published multiomic data from the mouse developing cortex (Noack et al., 2022).

5. Why did the authors choose UCON31 and MER130 as two specific classes of TE to study chromatin accessibility?

We apologize for not making the rationale for focusing on UCON31 and MER130 clearer. We examined the changes in accessibility and DNA methylation across all repeat classes as defined by RepeatMasker and a randomized control set of genomic regions. For each TE family we calculated mean accessibility/methylation per cell-type as well as if the difference observed is significant or not using wilcoxon test (Fig. 5A-B). Based on this analysis, only UCON31 and MER130 were identified as significantly different for chromatin accessibility, and only MER130 also for DNA methylation. We have elaborated on this in the figure legend and the methods.

6. Authors performed MPRA to show the utility of their protocol works not only in bulk nuclei but can also be extended to cell-type specific, can they perform a single-cell experiment (rna-seq, methylation, etc) to profile cell-type-specific enhancer activity?

It is important to note that the MPRA assay is not a part of the 3DRAM-seq method, but was used to further characterize the ability of the putative enhancer sequences (identified using 3DRAM-seq) to drive gene expression in organoids. In addition, we would like to point out that the combination of MPRA with a FACS-based purification already enables cell-type-specific enhancer activity as we demonstrate for RGC and IPC. Although it would be highly informative to examine enhancer activity at the single-cell level as the reviewer suggest, using scRNA-seq (presumably via the presence of eRNAs) would not be feasible, as these class of RNAs are extremely low abundant and not polyadenylated, so they will not be captured with commercially available single-cell methods such as 10x. scATAC-seq or, alternatively single-cell methylation, can be used to identify the heterogeneity within our putative enhancers but these modalities cannot be used to directly predict activity (for an excellent review on the topic, please see Gasperini et al., Nat Rev Genet 2020).

7. Some figures are missing scale bar, eg. x axis y axis annotation (1.6 in the top right corner in fig 1E is hard to see except mention in legend).

We apologize for this oversight and have corrected all the instances where we could identify missing or hard to see labels.

8. How was the protocol optimized for fixed cells? It is often difficult to separate clusters on FACS sort. The paper acknowledged their success as usually difficult to achieve but changes were not apparent.

Since the detection of chromatin folding requires fixation, 3DRAM-seq can be only performed on fixed samples. This, however, has several benefits including the ability to use antibodies against intracellular targets such as transcription factors (TFs), in comparison to live cells where only cell-surface markers are possible. To identify the correct combination of factors required to purify RGC and IPC, we used a combination of previously published methods (Song et al., Nature 2020) and our own expertise with immunoFACS in the developing mouse cortex (Noack et al., Nat Neurosci 2022).

To optimize our immunostaining protocol we tested different permeabilization reagents, different antibodies as well as concentration, optimized reagents to prevent degradation of DNA/RNA as well as proteins. We have added the relevant details and the missing citations to the text. We have also included a link to a protocols.io version of the immunoFACS protocol with details about each step (L745).

Protocol says cells can go directly into 3DRAM-seq before FACS sorting. It seems it would be beneficial to sort before to enrich for specific cell types for such a high resolution multi-omic approach. Is this an option or was it tried during adaptation of the protocol?

The Reviewer 2 is correct that an enrichment of cell-types is highly beneficial for such a multiomic approach and thus the compatibility with FACS was planned from the beginning as an optional step. Therefore, all the experiments in cortical organoids have been performed using FACS-sorting to enrich for RGC and IPC as shown in Fig. 4A-C. The results and the quality control comparison to the mouse ES cells (which were not sorted) indicates that introducing the sorting to purify desired cell types does not result in any loss of quality of complexity in 3DRAM-seq.

Reviewer #3 (Remarks to the Author):

This manuscript applied robust epigenomic/3D-genome profiling approaches and functional validation methods previously described in Noack et al., 2022 (from the same group) Nature Neuroscience to human brain organoids. Conceptually and also methodologically, this manuscript is quite similar to the published Noack et al., 2022 paper.

We thank the reviewer for pointing out the robustness in both our current manuscript but also our previous work. Although there is some conceptual overlap in that they both investigate the multi-layered epigenetic regulation in neural development, we would like to highlight some of the major differences between the two studies. First, 3DRAM-seq can be used to measure all 3 epigenetic modalities (accessibility, methylation and 3D genome) + gene expression simultaneously, while previously we were limited to only 2 (HiC + methylation or accessibility + methylation). Second, our previous work was entirely based on the mouse cortex, while this manuscript uses cortical organoids for identification/validation of putative human enhancers. Finally, we also examine the epigenetic changes associated with transposable elements and their ability to act as enhancers, which is not something we had studied previously.

In addition to these differences already present in the original version of the manuscript, we now also make several new additions inspired by the reviewers' comments, which we elaborate below. Specifically, we have developed a new computational pipeline which leverages 3DRAM-seq to enable paired co-accessibility measurements with single-molecule resolution as detailed below. Importantly, this approach is only possible using 3DRAM-seq and cannot be inferred using other methods such as ATAC-seq + Methyl-HiC. Applying this novel method to convergently-oriented Ctf motifs at chromatin loop anchors suggests that Ctf binding is dynamic and occurs independently, consistent with loop extrusion and recent findings based on microscopy (Gabriele et al., Science 2022; Mach et al., Nat Genet 2022) but arguing against previous models based on dimerization (Saldana-Meyer et al., Genes&Dev 2014). These results are described in an entirely new main and supplementary figure (Fig. 3 and S3). We have also expanded our observations on TF-based synergy in the MPRA to single-molecule co-accessibility, which confirms our previous results (Fig. 5J-K). Finally, we leverage our 3DRAM-seq and MPRA data in human organoids to identify a novel, human-specific enhancer element for the ventricular radial glia marker gene FBXO32, which we validate experimentally using in organoid electroporation (Fig. 7L-N).

The addition of M.CviPI-based chromatin accessibility profiling to Methyl-HiC is a relatively modest improvement that has been demonstrated multiple times by various groups.

Respectfully, we disagree with this statement, since, to the best of our knowledge, our study describes for the first time the combination of the M.CviPI based accessibility measurement with the interrogation of 3D genome architecture and DNA methylation. Perhaps the reviewer refers to the NOME-seq method which combines M.CviPI-based chromatin accessibility with DNA methylation, which indeed has been implemented by several groups including us (Kelly et al., 2012; Sönmezer et al. 2021; Noack et al., 2022). However, the addition of 3D genome architecture required non-trivial changes and significant optimizations due to the necessary formaldehyde fixation and streptavidin-biotin based enrichment of the resulting library. We also refer the reviewer to the direct comparison between our method and published NOME-seq dataset, which highlights the higher sensitivity of 3DRAM-seq (Figure R3, p4).

Although cerebral organoids have been shown as a sound model for brain development, there have also been compelling arguments that organoids fall short in modeling the specification of cell subtypes (for example, Bhaduri et al., 2020, PMID: 31996853).

We thank the reviewer for this comment as it indeed highlights the importance of carefully evaluating the advantages and disadvantages of using organoids as a model system. Our goal here was to choose a relatively early stage of organoid maturation (week 7) and to prioritize the two broad progenitor cell types such as RGC and IPC which we can purify reliably using our antibody strategy. Therefore, we do not make any claims about more specialized cell types such as oRG, cortical area or neuronal subtypes which, indeed, have been shown to vary between human fetal cortex and cerebral organoids. Nevertheless, we now include a specific comparison between the RGC/IPC accessible regions we identify in this manuscript and single-cell accessibility data from a recent study in the human fetal cortex (Ziffra et al., Nature 2021). We show that the majority of the regions identified in our organoid RGC/IPC are also accessible in the corresponding cell type in the human fetal cortex (Figure R7), which we also include in the text (L218-220). This suggests that the overall epigenome landscape within these two cell types is comparable, at least at the level of chromatin accessibility.

Cell Type	Total peaks (3DRAM-seq)	Overlapping with human fetal brain scATAC peaks	% overlap
RGC	39738	36111	90.9
IPC	54334	40826	75.1

Table R1. Overlap between RGC/IPC accessible peaks in organoids (this study) and human fetal brain (Ziffra et al., 2021)

So I would view the current manuscript as a bit less significant than Noack et al., 2022, especially from a data resource perspective), since Noack et al., 2022 profiled primary mouse brain tissues, whereas the present study only included data generated from organoids. Therefore although I think the presented works are solid, the manuscript might be a better fit for a more specialized journal.

To further strengthen the conceptual novelty of the current study we have now performed additional experiments to identify a novel human enhancer for the ventricular radial glia marker gene FBXO32, which we validate experimentally using in organoid electroporation (Fig. 7L-N). We focus on FBXO32 as we had previously shown that this gene is expressed in the human but not in the mouse cortex (Trevino et al., 2021; Noack et al., 2022). We show that this gene is also expressed specifically in our organoid-derived RGC and identify 4 putative enhancer elements which interact specifically with the FBXO32 promoter only in RGC (Fig. 7L). Importantly, we show that these enhancers and the cell-type-specific chromatin loops are not present in the mouse locus despite the overall high synteny (Fig. S7R). Finally, we show that these human enhancers exhibit high cell-type specificity based on the MPRA (Fig. S7Q) and experimentally validate that only the human, but not the mouse version is able to drive gene expression in organoids (Fig. 7M-N).

In addition to these novel experiments, we also now include a free and fully interactive platform which allows for the visualization of the 3DRAM-seq data we have generated in human organoids, which we hope will be of value to the community: shiny.bonevlab.com/ram .

More specific comments

1. Technically the RNA profile was not generated from the same nuclei sample as those used by the Methyl-HiC assay. It might be helpful to clarify that in Fig. 1A schematics.

We apologize that the origin of the RNA profile was misleading in Fig. 1A. Although, we had previously described this clearly in the text (L93-95), we now also make this distinction clear in the schematic. However, we do want to point out that the high quality of the RNA-seq data obtained from formaldehyde fixed cells is not a standard method and it really enables combining 3DRAM-seq with immunoFACS to profile rare cell types.

2. Could the authors further clarify the low overlapping between DMRs and DARs (1% for decreased DNA methylation, or 0.1% for increased DNA methylation). The degree of overlap seems to be much lower than what the meta-analysis in Fig. 4B suggests.

We thank the reviewer for this comment and apologize for the misleading text. The percentages stated were based on all GpC peaks and not just the DMRs (663 NSC and 66 IPC DMRs represent approximately 1% or 0.1% of all 67177 GpC peaks), respectively. As the reviewer rightfully expects, the overlap between NSC DMRs and IPC DARs is indeed very high - 89.3% (592/663) in agreement with the results shown in Fig. 4B (now 5B). We clarify this now in the text (Line 244-249).

3. The observation that LHX2-associated binding does not associate with DNA demethylation could be explained by that these regions are already lowly-methylation (<25%) to start with, whereas regions bound by Neurog2 show a methylation level of 50-60%. Therefore it is unclear whether the result can be generalized to an uncoupling of methylation and other data modalities.

We thank the reviewer for this comment which prompted us to examine the changes in LHX2-associated DNA methylation depending on the initial state of the regions. To this we stratified the LHX2 regions used for Extended Data Fig. 5K based on their methylation state in RGC into hypomethylated (less than 25%), average (between 25% and 75%) and hypermethylated (>75%). Importantly, while the methylation levels in RGC were anticorrelated with chromatin accessibility as expected, we did not observe any changes in DNA methylation levels irrespective of their initial state (Figure R7). This results suggest that the **changes** in chromatin accessibility associated with LHX2 are not always coupled with changes in DNA methylation. We include this clarification in the text (L277-280).

Figure R7. DNA methylation and chromatin accessibility levels at LHX2 RGC DARs stratified by the DNA methylation levels in RGC. Shown are also the number of sites per category.

The genomic contexts (promoters vs. enhancers, GpC contents, etc) of Lhx2 and Neurog2 binding sites should be further dissected.

We thank the reviewer for this comments which prompted us to examine the genomic context of the LHX2/NEUROG2 associated DARs more closely. We found that the LHX2 regions (and the RGC DARs in general) are typically associated with slightly lower GC content compared to NEUROG2 regions (Figure R8). This could be due to the differences in the motifs themselves or due to the preferred genomic context. However, we did not observe any differences when we compared LHX2 and NEUROG2 sites to genomic features such as promoters and genes.

Figure R8. GC content and genomic features enrichment at RGC and IPC DARs, containing either LHX2 or NEUROG2 respectively

4. How many copies of MER130 and UCON31 TEs were found in the human genome? This will help to distinguish whether the p-value difference shown in Fig. 5C-D (significant for MER130, not significant for UCON31) was driven by a difference in the effect size or sample size.

The number of MER130 and UCON31 transposable elements found in the human genome are comparable (181 and 117 respectively). Similar results were observed when the numbers for MER130 were downsampled to match UCON31. Therefore, we think it is unlikely that the observed differences are due to an unbalanced sample size. Nevertheless, we appreciate the comment and now indicate the number for each TE class in the text (L255-256).

Decision Letter, first revision:

: *Please delete the link to your author homepage if you wish to forward this email to co-authors.

Dear Dr Bonev,

Your manuscript, "Joint epigenome profiling reveals cell-type-specific gene regulatory programs in human cortical organoids", has now been seen by all of our original referees, who are experts in 3D-genome (referee 1); Chromatin accessibility (referee 2); and epigenetics (referee 3). As you will see from their comments (attached below) they find this work of interest, but have raised some important points. Although we are also very interested in this study, we believe that their concerns should be addressed before we can consider publication in Nature Cell Biology.

Nature Cell Biology editors discuss the referee reports in detail within the editorial team, including the chief editor, to identify key referee points that should be addressed with priority, and requests that are overruled as being beyond the scope of the current study. To guide the scope of the revisions, I have listed these points below. We are committed to providing a fair and constructive peer-review process, so please feel free to contact me if you would like to discuss any of the referee comments further.

In particular, it would be essential to:

- A) Include all necessary controls for all experiments as outlined by reviewer#1
- B) Clarify data and analysis throughout the manuscript, as recommended by reviewer#1
- C) Tone down claims around developing a novel variant of a massively parallel reporter assay (MPRA)
- D) All other referee concerns pertaining to strengthening existing data, providing controls, methodological details, clarifications and textual changes, should also be addressed.
- E) Finally please pay close attention to our guidelines on statistical and methodological reporting (listed below) as failure to do so may delay the reconsideration of the revised manuscript. In particular please provide:
 - a Supplementary Figure including unprocessed images of all gels/blots in the form of a multi-page pdf file. Please ensure that blots/gels are labeled and the sections presented in the figures are clearly indicated.
 - a Supplementary Table including all numerical source data in Excel format, with data for different figures provided as different sheets within a single Excel file. The file should include source data giving rise to graphical representations and statistical descriptions in the paper and for all instances where the figures present representative experiments of multiple independent repeats, the source data of all repeats should be provided.

In contrast, although we agree with referee 3 that physiological relevance (primary cells) would provide valuable insights, we consider this point to be beyond the scope of the present study. Thus, addressing it experimentally will not be necessary for reconsideration of the manuscript at this journal.

We therefore invite you to take these points into account when revising the manuscript. In addition, when preparing the revision please:

- ensure that it conforms to our format instructions and publication policies (see below and www.nature.com/nature/authors/).
- provide a point-by-point rebuttal to the full referee reports verbatim, as provided at the end of this letter.
- provide the completed Editorial Policy Checklist (found here <https://www.nature.com/authors/policies/Policy.pdf>), and Reporting Summary (found here <https://www.nature.com/authors/policies/ReportingSummary.pdf>). This is essential for reconsideration of the manuscript and these documents will be available to editors and referees in the event of peer review. For more information see <http://www.nature.com/authors/policies/availability.html> or contact me.

Nature Cell Biology is committed to improving transparency in authorship. As part of our efforts in this direction, we are now requesting that all authors identified as 'corresponding author' on published papers create and link their Open Researcher and Contributor Identifier (ORCID) with their account on the Manuscript Tracking System (MTS), prior to acceptance. ORCID helps the scientific community achieve unambiguous attribution of all scholarly contributions. You can create and link your ORCID from the home page of the MTS by clicking on 'Modify my Springer Nature account'. For more information please visit www.springernature.com/orcid.

[Redacted]

We would like to receive the revision within four weeks. If submitted within this time period, reconsideration of the revised manuscript will not be affected by related studies published elsewhere, or accepted for publication in Nature Cell Biology in the meantime. We would be happy to consider a revision even after this timeframe, but in that case we will consider the published literature at the time of resubmission when assessing the file.

We hope that you will find our referees' comments, and editorial guidance helpful. Please do not hesitate to contact me if there is anything you would like to discuss.

Best wishes,
Sabrya Carim

Sabrya Carim, PhD
(she/her/hers)
Associate Editor, Nature Cell Biology
Nature Portfolio

Springer Nature
The Campus, 4 Crinan Street, London N1 9XW, UK
sabrya.carim@springernature.com
<https://orcid.org/0000-0001-9485-1938>

Reviewers' Comments:

Reviewer #1:

Remarks to the Author:

Noack, Vangelisti et al. present a revised version of their manuscript describing 3DRAM-seq and its application in brain organoids. I find the revised manuscript to be a substantial improvement over the original submission. The additional analysis comparing different RNA-seq and open chromatin modalities goes a long way to convincing me of the quality of the data. Further, the new analysis of co-accessible motifs greatly improves the insights from the manuscript and indeed helps illustrate the utility of the 3DRAM-seq assay. The results of this analysis are pretty interesting, though somewhat unexpected (I would have assumed you would see greater than expected co-accessibility, I'm still trying to wrap my head around what this means). I still believe that there are some substantial issues that remain to be addressed, most of which surrounds the new analysis of co-accessibility, but I believe that with adequate revisions this study would ultimately be appropriate for publication in Nature Cell Biology. I have divided my comments into major and minor points below:

Major Comments:

I have several points related to the co-accessibility and single molecule interaction analysis. First, the methods for this section ("Co-accessibility and co-methylation analysis at single-molecule resolution") do not provide sufficient depth to either understand or recapitulate the analyses performed in the manuscript. They mostly describe some of the bioinformatic approaches used to identify reads, but they don't go into how any of the clustering was performed or any statistical tests presented.

Related to above, for the "co-accessibility pattern" testing (line 170-171), it isn't clear at all how this analysis was performed. For example, if their randomized control is 5% and their observed is 19%, how do they perform the testing and get an odds ratio of 0.94? They need more details here on what was done.

I think the authors should comment in the discussion on the fact they don't see greater than expected co-accessibility of CTCF or CRE-TSS during interactions. It is a somewhat unexpected result (at least in my mind) and I think providing some level of interpretation or speculation about why this is observed would be helpful. I think that this is an important result more broadly for the single-cell community, as methods for single-cell ATAC-seq for linking CRE to genes rely on such "co-accessibility" measurements (PMID 30078726), but this makes it clear that co-accessibility (at least within cell types) doesn't really match with other molecular features that relate to enhancer-

promoter connectivity.

Can the authors further explain the analysis in Figure 5J and K? This is a two part question, 1) Are they first just calling reads as co-accessible (based on what criteria) and then overlapping with motifs? And 2) Similar to the analysis with CTCF/Tss, is the co-accessibility here occurring more likely than by chance (at least within cell type)?

In the abstract and discussion (line 378), they say "3DRAM-seq can be applied to any tissue" but they can't really claim this, they have profiled still a limited number of cell types.

In both their prior submission and the current one, the authors state that they have developed a novel variant of MPRAs. But they haven't really done this. I think it would be more appropriate to say something like that they have "developed novel systems for applying MPRAs in organoids." Saying they have a novel variant of MPRAs is misleading to readers. They have this in the abstract and at line 276-277 and line 367.

Minor comments:

There is a competing preprint that has been online for quite some time now describing a somewhat similar method that I think the authors should cite: (<https://www.biorxiv.org/content/10.1101/2022.03.29.486102v1>).

Lhx2 in olfactory neurons is involved in the formation of long range and interchromosomal hubs. Are they seeing something similar or are these more local? Further Lhx2 is associated with Ldb proteins as a bridging molecule (and not CTCF/cohesin), so do the Lhx2 interactions show similar enrichment for CTCF/cohesin as non-Lhx2 interactions? This is an interesting observation.

In the prior submission, I suggested that the authors should show a sample that has not been treated with the GpC methyltransferase in Figure 1B and C. I still think they should do this. They point out that prior work showed that the GpC affects CpG methylation only in certain base contexts and they exclude these, and that is fine, but there is also an issue of the levels of non-CpG methylation in these cells and whether this is more likely to occur over accessible regions or not. I don't think they really need to do any additional experiments here, even just reanalyzing previously published bisulfite sequencing data from the same cell line would be sufficient.

For Fig. 2G, I think the authors should plot the total number of contacts in the datasets they use to make the figure here. The lack of signal in Methyl-HiC and enhanced signal in Hi-C could easily be due to read depth. Not accounting for read depth here could artificially lead readers to believe that certain methods are higher or lower quality when instead it could just be how much sequencing was performed.

For Fig. 2E, they say they define Hi-C contacts as paired end reads spanning a ligation junction, but I think that this isn't a great way to defined contacts, as if reads were simply undigested they would be counted as a contact according to this metric. I think that it is better to use distance cut offs that are larger than the fragment sized used for sequencing, in particular for defining "intra" contacts in order to exclude possibly uncut fragments from being counted as "contacts".

This is a minor language point, but the authors write: "We then performed 3DRAM-seq in mESC in three biological replicates (Supplementary Data 2), which were characterized by high bisulfite conversion efficiency at both CpG and GpC dinucleotides (>98%, Extended Data Fig. 1C)". The data in Extended Data Fig. 1C are from lambda phage DNA, but the way the sentence is written it makes

it seem like they have a method for calculating conversion efficiency from the mESC genome. I think the authors should point out that the conversion efficiencies are from lambda DNA spike ins.

At line 139 the authors have a call out for Extended Data Fig. 2H, but it looks like this should be Extended Data Figure 2i. The same goes for line 142, they call out Ex. Fig. 2i but it looks like that should be Ex. Data Fig. 2J. Related to this, they don't call out the real Ext. Data Fig. 2H in the main text, but they do discuss this in the reviewer response letter (so as a reviewer I get it but I don't think a reader would).

Reviewer #2:

Remarks to the Author:

The study introduces a novel and the revision significantly expands the 3DRAM-seq approach, which simultaneously profiles three epigenetic layers (3D genome, chromatin accessibility, and DNA methylation) at a single-cell resolution. The authors benchmarked the method against other multiomic approaches and found that it had the highest proportion of uniquely mapped reads and total Hi-C contacts. They also demonstrated the method's applicability to human cortical organoids and investigated the roles of transposable elements and specific transcription factors in regulating enhancers in human neurogenesis. Overall, the study's findings provide new insights into the complex interplay between epigenetic mechanisms in gene regulation and highlight the importance of simultaneous profiling of multiple layers of epigenetic information. The 3DRAM-seq method represents a significant step forward in multi-omic analysis and has the potential to advance our understanding of epigenetic regulation in diverse biological systems.

The revision process has improved the manuscript and it should be accepted.

Reviewer #3:

Remarks to the Author:

I have carefully considered the rebuttal provided by the authors, and I am afraid the revision did not significantly change my assessment of the manuscript: the works are solid, but the novelty and impact do not reach NCB.

The more fundamental concerns are -

1. the technological novelty is moderate. The profiling and validation methods are similar to that presented in Noack et al., 2022, published in a journal with a similar impact (Nature Neuroscience).
2. The novelty of biological samples (cerebral organoids) is modest. The culture of cerebral organoids is a relatively standard technique. It is unclear how much the findings represent endogenous biology without a rigorous comparison with primary human brain tissue.

GUIDELINES FOR SUBMISSION OF NATURE CELL BIOLOGY TECHNICAL REPORTS

TECHNICAL REPORT FORMAT

TITLE – should be no more than 100 characters including spaces, without punctuation and avoiding technical terms, abbreviations, and active verbs.

ABSTRACT – should not exceed 150 words and should be unreferenced. This paragraph is the most visible part of the paper and should briefly outline the background and rationale for the work, and accurately summarize the main results and conclusions. Key genes, proteins and organisms should be specified to ensure discoverability of the paper in online searches.

TEXT – the main text consists of the Introduction, Results, and Discussion sections and must not exceed 3000 words including the abstract. The Introduction should expand on the background relating to the work. The Results should be divided in subsections with subheadings, and should provide a concise and accurate description of the experimental findings. The Discussion should expand on the findings and their implications. All relevant primary literature should be cited, in particular when discussing the background and specific findings.

REFERENCES – are limited to a total of 40 in the main text and Methods combined (although they could be extended at the discretion of the editor). They must be numbered sequentially as they

appear in the main text, tables and figure legends and Methods and must follow the precise style of Nature Cell Biology references. References only cited in the Methods should be numbered consecutively following the last reference cited in the main text. References only associated with Supplementary Information (e.g. in supplementary legends) do not count toward the total reference limit and do not need to be cited in numerical continuity with references in the main text. Only published papers can be cited, and each publication cited should be included in the numbered reference list, which should include the manuscript titles. Footnotes are not permitted.

Methods should be written concisely, but should contain all elements necessary to allow interpretation and replication of the results. As a guideline, Methods sections typically do not exceed 3,000 words. The Methods should be divided into subsections listing reagents and techniques. When citing previous methods, accurate references should be provided and any alterations should be noted. Information must be provided about: antibody dilutions, company names, catalogue numbers and clone numbers for monoclonal antibodies; sequences of RNAi and cDNA probes/primers or company names and catalogue numbers if reagents are commercial; cell line names, sources and information on cell line identity and authentication. Animal studies and experiments involving human subjects must be reported in detail, identifying the committees approving the protocols. For studies involving human subjects/samples, a statement must be included confirming that informed consent was obtained. Statistical analyses and information on the reproducibility of experimental results should be provided in a section titled "Statistics and Reproducibility".

All Nature Cell Biology manuscripts submitted on or after March 21 2016, must include a Data availability statement at the end of the Methods section. For Springer Nature policies on data availability see <http://www.nature.com/authors/policies/availability.html>; for more information on this particular policy see <http://www.nature.com/authors/policies/data/data-availability-statements-data-citations.pdf>. The Data availability statement should include:

- Accession codes for primary datasets (generated during the study under consideration and designated as "primary accessions") and secondary datasets (published datasets reanalysed during the study under consideration, designated as "referenced accessions"). For primary accessions data should be made public to coincide with publication of the manuscript. A list of data types for which submission to community-endorsed public repositories is mandated (including sequence, structure, microarray, deep sequencing data) can be found here <http://www.nature.com/authors/policies/availability.html#data>.
- Unique identifiers (accession codes, DOIs or other unique persistent identifier) and hyperlinks for datasets deposited in an approved repository, but for which data deposition is not mandated (see here for details <http://www.nature.com/sdata/data-policies/repositories>).
- At a minimum, please include a statement confirming that all relevant data are available from the authors, and/or are included with the manuscript (e.g. as source data or supplementary information), listing which data are included (e.g. by figure panels and data types) and mentioning any restrictions on availability.
- If a dataset has a Digital Object Identifier (DOI) as its unique identifier, we strongly encourage including this in the Reference list and citing the dataset in the Methods.

We recommend that you upload the step-by-step protocols used in this manuscript to the Protocol Exchange. More details can be found at www.nature.com/protocolexchange/about.

DISPLAY ITEMS – main display items are limited to 6-8 main figures and/or main tables. For Supplementary Information see below.

FIGURES – Colour figure publication costs \$620 for the first, and \$310 for each subsequent colour figure. All panels of a multi-panel figure must be logically connected and arranged as they would appear in the final version. Unnecessary figures and figure panels should be avoided (e.g. data presented in small tables could be stated briefly in the text instead).

All imaging data should be accompanied by scale bars, which should be defined in the legend. Cropped images of gels/blots are acceptable, but need to be accompanied by size markers, and to retain visible background signal within the linear range (i.e. should not be saturated). The boundaries of panels with low background have to be demarked with black lines. Splicing of panels should only be considered if unavoidable, and must be clearly marked on the figure, and noted in the legend with a statement on whether the samples were obtained and processed simultaneously. Quantitative comparisons between samples on different gels/blots are discouraged; if this is unavoidable, it should only be performed for samples derived from the same experiment with gels/blots were processed in parallel, which needs to be stated in the legend.

- For line art, graphs, charts and schematics we prefer Adobe Illustrator (.AI), Encapsulated PostScript (.EPS) or Portable Document Format (.PDF). Files should be saved or exported as such directly from the application in which they were made, to allow us to restyle them according to our journal house style.
- We accept PowerPoint (.PPT) files if they are fully editable. However, please refrain from adding PowerPoint graphical effects to objects, as this results in them outputting poor quality raster art. Text used for PowerPoint figures should be Helvetica (preferred) or Arial.
- We do not recommend using Adobe Photoshop for designing figures, but we can accept Photoshop generated (.PSD or .TIFF) files only if each element included in the figure (text, labels, pictures, graphs, arrows and scale bars) are on separate layers. All text should be editable in 'type layers' and line-art such as graphs and other simple schematics should be preserved and embedded within 'vector smart objects' - not flattened raster/bitmap graphics.
- Some programs can generate Postscript by 'printing to file' (found in the Print dialogue). If using

an application not listed above, save the file in PostScript format or email our Art Editor, Allen Beattie for advice (a.beattie@nature.com).

Regardless of format, all figures must be vector graphic compatible files, not supplied in a flattened raster/bitmap graphics format, but should be fully editable, allowing us to highlight/copy/paste all text and move individual parts of the figures (i.e. arrows, lines, x and y axes, graphs, tick marks, scale bars etc). The only parts of the figure that should be in pixel raster/bitmap format are photographic images or 3D rendered graphics/complex technical illustrations.

SUPPLEMENTARY INFORMATION – Supplementary information is material directly relevant to the conclusion of a paper, but which cannot be included in the printed version in order to keep the manuscript concise and accessible to the general reader. Supplementary information is an integral part of a Nature Cell Biology publication, and should be prepared and presented with as much care as the main display item, but it must not include non-essential data or text, which may be removed at the editor's discretion. All supplementary material is fully peer-reviewed and published online as part of the HTML version of the manuscript. Supplementary Figures and Supplementary Notes are appended at the end of the main PDF of the published manuscript.

Unprocessed scans of all key data generated through electrophoretic separation techniques need to be presented in a supplementary figure that should be labeled and numbered as the final supplementary figure, and should be mentioned in every relevant figure legend. This figure does not count towards the total number of figures and is the only figure that can be displayed over multiple pages, but should be provided as a single file, in PDF or TIFF format. Data in this figure can be displayed in a relatively informal style, but size markers and the figures panels corresponding to the presented data must be indicated.

The total number of Supplementary Figures (not including the “unprocessed scans” Supplementary Figure) should not exceed the number of main display items (figures and/or tables (see our Guide to Authors and March 2012 editorial <http://www.nature.com/ncb/authors/submit/index.html#suppinfo>; <http://www.nature.com/ncb/journal/v14/n3/index.html#ed>). No restrictions apply to Supplementary Tables or Videos, but we advise authors to be selective in including supplemental

data.

GUIDELINES FOR EXPERIMENTAL AND STATISTICAL REPORTING

REPORTING REQUIREMENTS – To improve the quality of methods and statistics reporting in our papers we have recently revised the reporting checklist we introduced in 2013. We are now asking all life sciences authors to complete two items: an Editorial Policy Checklist (found here <https://www.nature.com/authors/policies/Policy.pdf>) that verifies compliance with all required editorial policies and a Reporting Summary (found here <https://www.nature.com/authors/policies/ReportingSummary.pdf>) that collects information on experimental design and reagents. These documents are available to referees to aid the evaluation of the manuscript. Please note that these forms are dynamic 'smart pdfs' and must therefore be downloaded and completed in Adobe Reader. We will then flatten them for ease of use by the reviewers. If you would like to reference the guidance text as you complete the template, please access these flattened versions at <http://www.nature.com/authors/policies/availability.html>.

Author Rebuttal, first revision:

Reviewer #1:

Remarks to the Author:

Noack, Vangelisti et al. present a revised version of their manuscript describing 3DRAM-seq and its application in brain organoids. I find the revised manuscript to be a substantial improvement over the original submission. The additional analysis comparing different RNA-seq and open chromatin modalities goes a long way to convincing me of the quality of the data. Further, the new analysis of co-accessibly motifs greatly improves the insights from the manuscript and indeed helps illustrate the utility of the 3DRAM-seq assay. The results of this analysis are pretty interesting, though somewhat unexpected (I would have assumed you would see greater than expected co-accessibility, I'm still trying to wrap my head around what this means). I still believe that there are some substantial issues that remain to be address, most of which surrounds the new analysis of co-accessibility, but I believe that with adequate revisions this study would ultimately be appropriate for publication in Nature Cell Biology. I have divided my comments into major and minor points below:

We thank the reviewer for the helpful comments and insights, which have substantially improved the manuscript.

Major Comments:

I have several points related to the co-accessibility and single molecule interaction analysis. First, the methods for this section ("Co-accessibility and co-methylation analysis at single-molecule resolution") do not provide sufficient depth to either understand or recapitulate the analyses performed in the manuscript. They mostly describe of some of the bioinformatic approaches used to identify reads, but they don't go into how any of the clustering was performed or any statistical tests presented.

We apologise for the insufficient details to fully describe the clustering approach used and the exact statistical tests. We now include this information in the figure legend as well as the methods section (L813-822). For convenience we briefly summarize it here:

- The average accessibility (based on GpC methylation) is calculated separately for read1 and read2 in a chosen window (100bp) centered at the motif of interest. Additionally, we required that the motifs are separated by at least 1kb on the linear genome.
- The resulting 2 columns matrix is then used as input for k-means clustering with $k=4$, $\text{iter.max}=10000$, $\text{nstart}=100$. Clusters are then reordered based on their mean value for consistency and the matrix is plotted using the R package complex heatmap. All subsequent analysis is performed using the exact same cluster assignments.
- To test if there is a dependency between accessibility in read1 and read2 we used fisher exact test on the 2x2 contingency matrix and we reported the odds ratio as well as the p-value (0.94 and 0.52 respectively for Ctfc pairs in Fig 3B). This approach is analogous to what was used in Sönmezer et al., Mol Cell 2021 to test for co-occupancy of TF using single-molecule GpC methylation data and aims to test whether the null hypothesis (accessibility at read1 and read2 are independent events) can be rejected. In the case of Ctfc and CRE-TSS the null hypothesis could not be rejected, therefore we concluded that the accessibility pattern at read1 and read2 is a result of independent events. Same results were obtained if we used a Chi-squared test instead ($X\text{-squared}=0.38$) or pearson's correlation ($r = 0.0024$)

Related to above, for the “co-accessibility pattern” testing (line 170-171), it isn't clear at all how this analysis was performed. For example, if their randomized control is 5% and their observed is 19%, how do they perform the testing and get an odds ratio of 0.94? They need more details here on what was done.

We apologies for the confusion here. The randomized control is generated by including read pairs where read2 does not have to overlap with a Ctfc motif (in comparison to Fig 3B, where both reads have to overlap a Ctfc motif). This is useful to estimate the “genomic background” probability for a read to be accessible and to show that if a read overlaps a Ctfc motif it has a high probability of being accessible (~43% for read2 in Fig3B vs only 12% of read2 spanning a random position in Fig S3A). However, this randomized control cannot be used to test if co-accessibility at CTCF sites results from independent or synergistic events, because there is no requirement for read2 to overlap a CTCF site. This is why we use the fisher exact test as described above.

I think the authors should comment in the discussion on the fact they don't see greater than expected co-accessibility of CTCF or CRE-TSS during interactions. It is a somewhat unexpected result (at least in my mind) and I think providing some level of interpretation or speculation about why this is observed would be helpful. I think that this is an important result more broadly for the single-cell community, as methods for single-cell ATAC-seq for linking CRE to genes rely on such “co-accessibility” measurements (PMID 30078726), but this makes it clear that co-accessibility (at least within cell types) doesn't really match with other molecular features that relate to enhancer-promoter connectivity.

We agree that these results are important and elaborate more in the discussion (L277-283). We believe that these results are consistent with recent findings based on microscopy which suggest that Ctf binding is highly dynamic and loops occur only between 3-6% of the time (Gabriele et al., Science 2022). Similarly, live imaging of enhancer-promoter contacts at the Sox2 locus (Alexander et al, eLife 2019) suggest that such loops are also relatively rare and are not directly correlated with transcription (although enhancer-promoter proximity influence gene expression in a non-linear way – Zuin et al., Nature 2022). Therefore, one hypothesis is that the time these regions spend in proximity may not be enough for synergistic (or antagonistic) effects on accessibility. A testable prediction from this hypothesis (however outside of the scope of this manuscript) would be that a forced chromatin loop between two Ctf sites (or CRE-TSS) would increase the co-accessibility levels.

Regarding using co-accessibility to link CREs and genes in single-cell data, it is important to note that current algorithms (such as Cicero) aggregate accessibility across multiple (typically 50) cells to gain power, thus they do not represent truly single-cell measurements. Second, as the reviewer correctly points out, such co-accessibility links are usually performed in tissues containing different cell types. This is similar to our analysis in the organoids where we do indeed observe changes in co-accessibility across two cell types (Fig 5J-K).

Can the authors further explain the analysis in Figure 5J and K? This is a two part question, 1) Are they first just calling reads as co-accessible (based on what criteria) and then overlapping with motifs? And 2) Similar to the analysis with CTCF/Tss, is the co-accessibility here occurring more likely than by chance (at least within cell type)?

The analysis for Figure 5J and 5K was performed very similar to the CTCF-based analysis in Figure 3B. First, we filtered LHX2/SOX2 or NEUROG2/EOMES motifs retaining only those that overlapped with a GpC peak (based on bulk accessibility in RGC or IPC respectively). Next, we identified all read pairs where read1 overlapped with one of the motifs (for example LHX2) and read2 overlapped with the other motif (SOX2 respectively). We then measured the average accessibility per read within a 50bp window for reads that are separated by at least 100bp but not more than 300bp. This distance cutoff is different from our measurements of long-range interactions associated with Ctf loops, because we wanted to determine if these pairs of TFs interact directly or co-bind on chromatin synergistically at closer distances. We now include also the co-accessibility statistics for both LHX2/SOX2 and NEUROG2/EOMES (L188-191). Fisher's exact test confirms that the two interact synergistically (odds ratios for SOX2-LHX2: 3.23 in RGC 9.3 in IPC; odds ratios for NEUROG2-EOMES: 11.08 in RGC and 4.1 in IPC). Interestingly, these results suggest that the low relative proportion of co-accessible reads in the opposite cell type (9 and 8% respectively) are

likely also a result of synergistic interactions between TFs, probably expressed at low levels in either neurogenic RGC or not fully mature IPCs.

In the abstract and discussion (line 378), they say “3DRAM-seq can be applied to any tissue” but they can’t really claim this, they have profiled still a limited number of cell types.

We agree with the reviewer and apologies with the overstatement. We have now removed this statement from the abstract and the discussion.

In both their prior submission and the current one, the authors state that they have developed a novel variant of MPRA. But they haven’t really done this. I think it would be more appropriate to say something like that they have “developed novel systems for applying MPRA in organoids.” Saying they have a novel variant of MPRA is misleading to readers. They have this in the abstract and at line 276-277 and line 367.

We agree with the reviewer and had already adjusted the text in the revised version to replace “novel” with “modified”. We believe that the reviewer was looking at the original version of the manuscript, as there is no mention of MPRA in the respective lines in the resubmitted version. Nevertheless, we have now adopted the reviewer’s suggestion and have either removed this altogether (in the abstract) or referred to it as “developed a novel system for applying an MPRA” in the discussion (L27-28, L215-216, L299).

Minor comments:

There is a competing preprint that has been online for quite some time now describing a somewhat similar method that I think the authors should cite:

(<https://www.biorxiv.org/content/10.1101/2022.03.29.486102v1>).

We now cite and discuss this manuscript and comment on the differences and similarities between NOME-HIC and 3DRAM-seq (L302-309).

Lhx2 in olfactory neurons is involved in the formation of long range and interchromosomal hubs. Are they seeing something similar or are these more local? Further Lhx2 is associated with Ldb proteins as a

bridging molecule (and not CTCF/cohesin), so do the Lhx2 interactions show similar enrichment for CTCF/cohesin as non-Lhx2 interactions? This is an interesting observation.

The analysis in Figure 5 focuses on intraTAD interactions between pairs of regions overlapping an accessible LHX2 motif. In that sense, these interactions are local and are different than the long-range and trans contacts reported by the Lomvardas lab in olfactory neurons. Indeed, we don't observe such long-range trans-interactions in our data, but this is not surprising as they have been reported in mouse and only form during the maturation of olfactory neurons. Instead, our results point to a role of LHX2 in mediating close-range interactions, potentially also via LDB1 (now included in the discussion – L288-290). How mechanistically these contacts are established and if this is dependent of Ctfc/cohesin is a very interesting questions, which is however difficult to resolve currently, as there is no CHIP/Cut&Run data available in these cell types or tissues.

In the prior submission, I suggested that the authors should show a sample that has not been treated with the GpC methyltransferase in Figure 1B and C. I still think they should do this. They point out that prior work showed that the GpC affects CpG methylation only in certain base contexts and they exclude these, and that is fine, but there is also an issue of the levels of non-CpG methylation in these cells and whether this is more likely to occur over accessible regions or not. I don't think they really need to do any additional experiments here, even just reanalyzing previously published bisulfite sequencing data from the same cell line would be sufficient.

We agree with the reviewer and have now included a plot comparing the levels of GpC methylation between 3DRAM-seq and WGBS at Ctfc and Nrf1 sites (as in Fig 1B and 1C). The results clearly show that there is almost no detectable GpC methylation and no preferential enrichment at accessible regions unless treated with the M.CviPI GpC methyltransferase.

Figure R1. GpC methylation levels across Ctfc or Nrf1 bound motifs for either 3DRAM-seq or WGBS data from the same cell line (E14).

For Fig. 2G, I think the authors should plot the total number of contacts in the datasets

they use to make the figure here. The lack of signal in Methyl-HiC and enhanced signal in Hi-C could easily be due to read depth. Not accounting for read depth here could artificially lead readers to believe that

certain methods are higher or lower quality when instead it could just be how much sequencing was performed.

We agree with the reviewer and have included this information in Figure 2F legend. Indeed, the reviewer is correct, and the resolution is highly correlated with the sequencing depth, which we now specifically state in the figure legend. Our intention was simply to show that 3DRAM-seq is capable of identifying close-range interactions similarly to the other methods.

For Fig. 2E, they say they define Hi-C contacts as paired end reads spanning a ligation junction, but I think that this isn't a great way to defined contacts, as if reads were simply undigested they would be counted as a contact according to this metric. I think that it is better to use distance cut offs that are larger than the fragment sized used for sequencing, in particular for defining "intra" contacts in order to exclude possibly uncut fragments from being counted as "contacts".

Indeed, the reviewer is correct and this is very important to consider in chromatin conformation capture data. All of our analysis has been done using contacts separated by at least 1kb on the linear genome to avoid precisely these potential "false positive" contacts. Previously we have included information about "short" and "long range" contacts in Supplementary Table 1, but we now explicitly add this also in Fig 2E to compare across all different methods. As expected, methods which utilize streptavidin-biotin to enrich for ligation events (3DRAM-seq, Methyl-HiC) lead to a higher percentage of informative, long-range interactions.

This is a minor language point, but the authors write: "We then performed 3DRAM-seq in mESC in three biological replicates (Supplementary Data 2), which were characterized by high bisulfite conversion efficiency at both CpG and GpC dinucleotides (>98%, Extended Data Fig. 1C)". The data in Extended Data Fig. 1C are from lambda phage DNA, but the way the sentence is written it makes it seem like they have a method for calculating conversion efficiency from the mESC genome. I think the authors should point out that the conversion efficiencies are from lambda DNA spike ins.

We thank the reviewer for pointing out this ambiguity and have amended it accordingly "**...efficiency based on spike-in controls**" (L70-71).

At line 139 the authors have a call out for Extended Data Fig. 2H, but it looks like this should be Extended Data Figure 2i. The same goes for line 142, they call out Ex. Fig. 2i but it looks like that should be Ex. Data

Fig. 2J. Related to this, they don't call out the real Ext. Data Fig. 2H in the main text, but they do discuss this in the reviewer response letter (so as a reviewer I get it but I don't think a reader would).

We apologies for this oversight and have amended the text accordingly, including also a reference to Extended Data Fig. 2H.

Reviewer#2:

Remarks to the Author:

The study introduces a novel and the revision significantly expands the 3DRAM-seq approach, which simultaneously profiles three epigenetic layers (3D genome, chromatin accessibility, and DNA methylation) at a single-cell resolution. The authors benchmarked the method against other multiomic approaches and found that it had the highest proportion of uniquely mapped reads and total Hi-C contacts. They also demonstrated the method's applicability to human cortical organoids and investigated the roles of transposable elements and specific transcription factors in regulating enhancers in human neurogenesis. Overall, the study's findings provide new insights into the complex interplay between epigenetic mechanisms in gene regulation and highlight the importance of simultaneous profiling of multiple layers of epigenetic information. The 3DRAM-seq method represents a significant step forward in multi-omic analysis and has the potential to advance our understanding of epigenetic regulation in diverse biological systems.

The revision process has improved the manuscript and it should be accepted.

We thank the reviewer for the helpful comments and insights, which have substantially improved the manuscript.

Reviewer #3:

Remarks to the Author:

I have carefully considered the rebuttal provided by the authors, and I am afraid the revision did not significantly change my assessment of the manuscript: the works are solid, but the novelty and impact do not reach NCB.

The more fundamental concerns are –

1. the technological novelty is moderate. The profiling and validation methods are similar to that presented in Noack et al., 2022, published in a journal with a similar impact (Nature Neuroscience).

2. The novelty of biological samples (cerebral organoids) is modest. The culture of cerebral organoids is a relatively standard technique. It is unclear how much the findings represent endogenous biology without a rigorous comparison with primary human brain tissue.

We apologies for not making the conceptual novelty of our study clearer. We have now gone thoroughly through the text to ensure we highlight the technical novelty of the study and how it differs from previously published multiomic methods. Fruthermore, we believe that the comparison with the human fetal brain epigenetic landscape (L146-148) indicates that our cortical organoids are highly representative of the human brain tissue, at least at the level of chromatin accessibility.

Decision Letter, second revision:

Our ref: NCB-TR49535B

14th July 2023

Dear Dr. Bonev,

Thank you for submitting your revised manuscript "Joint epigenome profiling reveals cell-type-specific gene regulatory programs in human cortical organoids" (NCB-TR49535B). It has now been seen by one of the original referees and their comments are below. The reviewers find that the paper has improved in revision, and therefore we'll be happy in principle to publish it in Nature Cell Biology, pending minor revisions to satisfy the referees' final requests and to comply with our editorial and formatting guidelines.

- Please supply Source Data files for all data presented in graphs within the Figures and Extended Data. Please provide numerical source data in Excel format, with data for different figures provided as different sheets within a single Excel file. The file should include source data giving rise to graphical representations and statistical descriptions in the paper and for all instances where the figures present representative experiments of multiple independent repeats, the source data of all repeats should be provided.

- Please note that all key data should be presented in the main figures, with Extended Data only presenting supportive information. There is a limit of 6-8 display items for [Technical Reports] and all key data should be presented in these figures. Please keep the current structure of the figures, but ensure that you are using the full A4 space and increase panel size and font size (at least 7 pt) for improved readability. More data can be moved from Extended Data Figures into the main figures. No peer-reviewed data should be removed.

Please ensure that all figures fit into a single standard page and adhere to a maximum page size of roughly 180mm wide x 200mm high, but also please use the full page space to fill the figure. At

present several figures are too tiny to be legible once re-sized during the production process. To ensure legibility once figures are re-sized, please use a font size of no smaller than 6pt Arial or Helvetica throughout the figures.

Thank you again for your interest in Nature Cell Biology Please do not hesitate to contact me if you have any questions.

Sincerely,

Sabrya Carim, PhD
(she/her/hers)
Associate Editor, Nature Cell Biology
Nature Portfolio

Springer Nature
The Campus, 4 Crinan Street, London N1 9XW, UK
sabrya.carim@springernature.com
<https://orcid.org/0000-0001-9485-1938>

Reviewer #1 (Remarks to the Author):

Noack et al present another revised version of their study describing 3DRAM-seq and its application in cortical organoids. I find this to be another continued improvement on their prior revision. They have addressed my concerns from the prior round of review and I believe this study is now appropriate for publication.

Decision Letter, final checks:

Our ref: NCB-TR49535B

11th August 2023

Dear Dr. Bonev,

Thank you for your patience as we've prepared the guidelines for final submission of your Nature Cell Biology manuscript, "Joint epigenome profiling reveals cell-type-specific gene regulatory programs in human cortical organoids" (NCB-TR49535B). Please carefully follow the step-by-step instructions provided in the attached file, and add a response in each row of the table to indicate the changes that you have made. Please also check and comment on any additional marked-up edits we have proposed within the text. Ensuring that each point is addressed will help to ensure that your revised manuscript

can be swiftly handed over to our production team.

In recognition of the time and expertise our reviewers provide to Nature Cell Biology's editorial process, we would like to formally acknowledge their contribution to the external peer review of your manuscript entitled "Joint epigenome profiling reveals cell-type-specific gene regulatory programs in human cortical organoids". For those reviewers who give their assent, we will be publishing their names alongside the published article.

Nature Cell Biology offers a Transparent Peer Review option for new original research manuscripts submitted after December 1st, 2019. As part of this initiative, we encourage our authors to support increased transparency into the peer review process by agreeing to have the reviewer comments, author rebuttal letters, and editorial decision letters published as a Supplementary item. When you submit your final files please clearly state in your cover letter whether or not you would like to participate in this initiative. Please note that failure to state your preference will result in delays in accepting your manuscript for publication.

Cover suggestions

As you prepare your final files we encourage you to consider whether you have any images or illustrations that may be appropriate for use on the cover of Nature Cell Biology.

Nature Cell Biology has now transitioned to a unified Rights Collection system which will allow our Author Services team to quickly and easily collect the rights and permissions required to publish your

work. Approximately 10 days after your paper is formally accepted, you will receive an email in providing you with a link to complete the grant of rights. If your paper is eligible for Open Access, our Author Services team will also be in touch regarding any additional information that may be required to arrange payment for your article.

Please note that *Nature Cell Biology* is a Transformative Journal (TJ). Authors may publish their research with us through the traditional subscription access route or make their paper immediately open access through payment of an article-processing charge (APC). Authors will not be required to make a final decision about access to their article until it has been accepted. Find out more about Transformative Journals

Please use the following link for uploading these materials:
[Redacted]

Best regards,

Adam Lipkin
Staff
Nature Cell Biology

On behalf of

Sabrya Carim, PhD
(she/her/hers)
Associate Editor, Nature Cell Biology
Nature Portfolio

Springer Nature
The Campus, 4 Crinan Street, London N1 9XW, UK
sabrya.carim@springernature.com
<https://orcid.org/0000-0001-9485-1938>

Reviewer #1:

Remarks to the Author:

Noack et al present another revised version of their study describing 3DRAM-seq and its application in cortical organoids. I find this to be another continued improvement on their prior revision. They have addressed my concerns from the prior round of review and I believe this study is now appropriate for publication.

Author Rebuttal, second revision:

Reviewer #1:

Remarks to the Author:

Noack et al present another revised version of their study describing 3DRAM-seq and its application in cortical organoids. I find this to be another continued improvement on their prior revision. They have addressed my concerns from the prior round of review and I believe this study is now appropriate for publication.

We thank the reviewer for the constructive feedback which has allowed us to significantly improve the manuscript.

Final Decision Letter:

Dear Dr Bonev,

I am pleased to inform you that your manuscript, "Joint epigenome profiling reveals cell-type-specific gene regulatory programs in human cortical organoids", has now been accepted for publication in Nature Cell Biology- Congratulations!

Over the next few weeks, your paper will be copyedited to ensure that it conforms to Nature Cell

Biology style. Once your paper is typeset, you will receive an email with a link to choose the appropriate publishing options for your paper and our Author Services team will be in touch regarding any additional information that may be required.

Please note that *Nature Cell Biology* is a Transformative Journal (TJ). Authors may publish their research with us through the traditional subscription access route or make their paper immediately open access through payment of an article-processing charge (APC). Authors will not be required to make a final decision about access to their article until it has been accepted. Find out more about Transformative Journals

If you have not already done so, we strongly recommend that you upload the step-by-step protocols used in this manuscript to the Protocol Exchange (www.nature.com/protocolexchange), an open online resource established by Nature Protocols that allows researchers to share their detailed experimental know-how. All uploaded protocols are made freely available, assigned DOIs for ease of citation and are fully searchable through nature.com. Protocols and Nature Portfolio journal papers in which they are used can be linked to one another, and this link is clearly and prominently visible in the online versions of both papers. Authors who performed the specific experiments can act as primary authors for the Protocol as they will be best placed to share the methodology details, but the Corresponding Author of the present research paper should be included as one of the authors. By uploading your Protocols to Protocol Exchange, you are enabling researchers to more readily reproduce or adapt the methodology you use, as well as increasing the visibility of your protocols and papers. You can also establish a dedicated page to collect your lab Protocols. Further information can be found at www.nature.com/protocolexchange/about

Please feel free to contact us if you have any questions. Congratulations again.

With kind regards,

Sabrya Carim, PhD
(she/her/hers)
Associate Editor, Nature Cell Biology
Nature Portfolio

Springer Nature
The Campus, 4 Crinan Street, London N1 9XW, UK
sabrya.carim@springernature.com
<https://orcid.org/0000-0001-9485-1938>

** Visit the Springer Nature Editorial and Publishing website at www.springernature.com/editorial-and-publishing-jobs for more information about our career opportunities. If you have any questions please click here.**